# Demystifying integrable QFTs in AdS:
# No-go theorems for higher-spin charges

**António Antunes,**[d] **Nat Levine,**[p] **Marco Meineri**[k]

[d]*Deutsches Elektronen-Synchrotron DESY, Notkestr. 85, 22607 Hamburg, Germany*

[d,p]*Laboratoire de Physique de l'École Normale Supérieure, Université PSL, CNRS, Sorbonne Université, Université Paris Cité, 24 rue Lhomond, F-75005 Paris, France*

[p]*Institute for Theoretical Physics, University of Amsterdam, PO Box 94485, 1090 GL Amsterdam, The Netherlands*

[k]*Istituto Nazionale di Fisica Nucleare, Torino Section, and Department of Physics, University of Turin, Via P. Giuria 1, 10125, Turin, Italy*

*E-mail:*
antonio.antunes@phys.ens.fr, n.j.levine@uva.nl, marco.meineri@unito.it

ABSTRACT: Higher-spin conserved currents and charges feature prominently in integrable 2d QFTs in flat space. Motivated by the question of integrable field theories in AdS space, we consider the consequences of higher-spin currents for QFTs in $AdS_2$, and find that their effect is much more constraining than in flat space. Specifically, it is impossible to preserve: (a) any higher-spin charges when deforming a massive free field by interactions, or (b) any spin-4 charges when deforming a CFT by a Virasoro primary. Therefore, in these settings, there are no integrable theories in AdS with higher-spin conserved charges. Along the way, we explain how higher-spin charges lead to integer spacing in the spectrum of primaries, sum rules on the OPE data, and constraints on correlation functions. We also explain a key difference between AdS and flat space: in AdS one cannot 'partially' conserve a higher-spin current along particular directions, since the AdS isometries imply full conservation.

# 1  Introduction

While quantum field theories (QFTs) are typically expected to behave chaotically, there are notable exceptions in flat 2d space: the **integrable** QFTs. These special theories— examples including the sine-Gordon model, $O(n)$ sigma-model, and affine Toda models— are associated with exact solvability, and often feature large extended symmetry algebras. One definition of integrability in flat 2d space is stated in terms of the S-matrix: there should be no particle production (i.e. conserved particle number); the set of individual momenta should be preserved; and the $n \to n$ S-matrix should factorise as a product of $2 \to 2$ S-matrices.

Given the existence of integrable QFTs in flat 2d space and their beautiful properties, it is natural to ask if some analogue of integrability is also possible in **curved 2d spacetimes**. One particular motivation to ask this question is the recent activity surrounding QFTs in Anti de Sitter (AdS) space [1–13]. This 'rigid holography' setup is amenable to many of the tools of conventional holography, and describes a conformal theory at the boundary of AdS. The conformal boundary correlators are analogous to the flat space S-matrix and, indeed, when the AdS radius becomes large compared to characteristic scales of the theory, they have been shown to map directly onto flat-space S-matrix elements [4, 7, 14–17]. For this reason, they may be the natural observables to make integrability manifest in AdS$_2$, if it exists.

However, we do not know precisely which constraints integrability would place on the boundary correlators in AdS$_2$. Since **higher-spin conserved charges** are a ubiquitous feature of integrable field theories—and they imply S-matrix factorisation in flat space [18– 20]—*we will demand the existence of conserved, local, higher-spin currents in AdS$_2$.*[1] We will find that this assumption is drastically more constraining for QFTs in AdS:

1. Unlike flat space, it is impossible to 'partially' conserve a higher-spin current in AdS$_2$.

2. Higher-spin symmetries lead to integer spacings in the spectrum, and are found in (a) massive free fields in AdS and (b) CFTs in AdS.

3. We show that the higher-spin symmetries cannot be preserved by:
    (a) interacting deformations of a massive free field in AdS,
    (b) relevant deformations of Virasoro CFTs in AdS by a primary operator,

    at least for generic values of the mass in the free-field case and for spin-4 charges in the CFT case.

Let us explain the meaning of the first result: in flat space, it is possible to conserve only null components of higher-spin currents, without conserving the full current. In fact, the algebra of charges of an integrable theory is typically an infinite abelian subalgebra of a larger higher-spin algebra present in the UV [22]. Point 1 above is the statement that this is not possible in AdS.

---

[1]In higher-dimensional flat space, higher-spin charges famously lead to trivial scattering [21]. Whether a similar theorem applies for QFT in AdS$_{d \geq 3}$ is an interesting problem on its own, but in this paper we focus on two-dimensional physics because of the connection to integrability.

The symmetries mentioned in Point 2 are familiar to experts in higher-spin algebras [23–27] and integrable structures in 2d CFT [28, 29], but here we present them in the language of QFTs in AdS and their boundary correlators to make the consequences more tangible.

Point 3 is reminiscent of the finding of [30–32] that local conformal field theories in $d \geq 3$ with higher-spin currents are constrained to have a free sub-sector. The higher-spin symmetries of those theories are gauged in the bulk [33–38] according to the usual AdS/CFT dictionary [39–41], while ours are global. Correspondingly, the 1d CFTs of our interest do not have local currents, but only conserved charges.

Another motivation for investigating integrable QFTs in $AdS_2$ is the role of this space as an IR regulator. While massless 2d scalars are IR-divergent [42], they superficially appear in the semi-classical expansion of sigma-models around the 'trivial' vacuum, even in integrable cases with non-perturbative mass generation (such as the Principal Chiral Model or the $O(N)$ sigma-model). An algorithmic way to couple integrable models to $AdS_2$ would elegantly regulate their IR and open a path to classifying integrable sigma-models by imposing factorised scattering (while naive IR regulators seem to destroy integrability [43–45]—see also the recent approach [46]).

Aside from integrability, a further reason for this study came from the conformal bootstrap, where optimal bootstrap bounds are saturated by so-called extremal solutions [47]. As well as saturating unitarity bounds (just like integrable S-matrices in flat space), these special conformal correlators generically exhibit sparse spectra. As an example, let us consider free fields in $AdS_2$, whose boundary $CFT_1$ are generalized free fields $\phi$. In the 4-point correlator $\langle \phi\phi\phi\phi \rangle$, the exchanged operators consist of a single two-particle tower, given by the operator product expansion (OPE) $\phi \times \phi = \sum_n (\phi \, \partial^{2n} \phi)$. In contrast, adding generic interaction vertices in AdS would generate an exponentially denser OPE,[2] containing all multi-particle operators $\partial^n \phi^m$. However, there exist some carefully tuned deformations of generalized free field 4-point functions that (i) solve the crossing equation, (ii) are as sparse as the free solution and (iii) saturate bootstrap bounds [48]. The fact these these solutions only exchange deformed 'two-particle' operators $\phi \, \partial^{2n} \phi$ is reminiscent of integrable scattering. A direct connection between sparse spectra and integrable theories can indeed be established in the flat space limit, using the phase-shift formula of [1]. If, in the flat space limit, the only contribution to the OPE is a tower of two-particle states with dimension $\Delta_n = 2\Delta_\phi + 2n + \gamma_n$,[3] then the four-point function leads to an elastic 2-to-2 S-matrix [7]. Of course, the extremal solutions of [48] may not extend beyond a single correlator, but one may wonder if some of them do belong to fully fledged theories, and if they can be interpreted as integrable QFTs on $AdS_2$. Our results show that such theories cannot have higher-spin charges.

---

[2]More precisely, the full generalized free field density of states grows as $\exp(\sqrt{\Delta})$, which is the same as the Cardy formula for a local 2d CFT.

[3]The anomalous dimension $\gamma_n$ should be slowly varying in the same limit, in the sense that $\gamma_n(\Delta_\phi) \sim \gamma(n/\Delta_\phi)$ as $\Delta_\phi \to \infty$.

While these results seem to disfavour the existence of integrable field theories in $\text{AdS}_2$, it is of course possible that they exist, but simply do not have higher-spin charges. We will discuss other possibilities in the conclusions.

We now give an outline of the paper, emphasising the main results of each section. We begin in Section 2 by defining higher-spin charges for the boundary theory as integrals of higher-spin currents contracted with Killing tensors. We study the condition of finiteness of these charges near the boundary and analyze in detail the case of massive free fields in AdS, dual to generalized free fields on the boundary. Using explicit higher-spin currents in the bulk, we determine the charges' action on boundary operators, justifying integer spacing in the spectrum of primaries. In Section 3, we study the systematics of higher-spin currents and charges in $\text{AdS}_2$: after proving that the currents cannot be 'partially' conserved in AdS, we construct the associated charges and derive their commutation relations with conformal generators. In Section 4, we study the consequences of higher-spin (HS) charges from the perspective of the boundary CFT. After understanding the algebra of HS charges, we use them to derive Ward identities on conformal correlators. We then use these HS Ward identities to prove one of our key results: the absence of perturbative interacting deformations of generalized free fields that preserve the higher-spin charges. In Section 4.5, which is slightly out of the main flow of the paper, we turn our attention to the bulk, using HS conservation to derive sum rules that constrain correlators involving bulk fields.

We then change gears in Section 5 to consider bulk 2d CFTs with Virasoro symmetry. Such theories have an algebra of higher-spin charges, embedded in the universal enveloping algebra of Virasoro. We then use conformal perturbation theory to show that, in all but one case, relevant deformations of bulk CFTs triggered by a single Virasoro primary operator break the integer spacing guaranteed by spin-4 charges. The exception is the thermal deformation of the Ising model, i.e. precisely the generalized free fermion theory discussed above. We present our conclusions in Section 6.

Appendix A contains additional technical results on the generalized free boson theory, Appendix B studies the boundary operator expansion of HS currents, Appendix C describes the space of AdS Killing tensors and Appendix D gives further details on the derivation of the sum rules of Section 4.5.

## 2 Higher-spin currents and charges

An interesting feature of QFTs in AdS is the way they realize the conformal symmetry of the boundary theory. The conformal generators are not given by integrals of conserved conformal currents on the boundary (which are absent).[4] Instead, they are given by integrals of the **bulk** stress tensor, contracted with AdS Killing vectors:

$$Q_\xi = \int_\Sigma \sqrt{\gamma}\, dS^\mu\, \xi^\nu\, T_{\mu\nu}\,, \tag{2.1}$$

---

[4]This may seem surprising given Noether's theorem that continuous symmetries of the action are associated to local currents. Of course, the theorem assumes a local Lagrangian so there is no contradiction here (see related discussion in [49]).

Here the integral is over a co-dimension 1 surface $\Sigma$ in AdS, $\gamma$ is the determinant of the induced metric on $\Sigma$, and $dS_\mu$ is the normal oriented surface element on $\Sigma$. We are interested in two classes of surfaces: compact ones, and open surfaces which end on the boundary of AdS. As we will review momentarily, it is the latter that implement the action of the symmetries on the boundary. Since the stress tensor $T_{\mu\nu} = T_{(\mu\nu)}$ is conserved,

$$\nabla^\mu T_{\mu\nu} = 0 \,, \tag{2.2}$$

and $\xi$ is Killing (i.e. $\nabla_{(\mu}\xi_{\nu)} = 0$), then the current $J_\mu = \xi^\nu T_{\mu\nu}$ is conserved. Hence the charge (2.1) is conserved in the sense that it is topological: it is invariant under continuous deformations of the surface $\Sigma$.

As a maximally symmetric 2d space with curvature radius $L$, $\text{AdS}_2$ has three such Killing vector fields $\xi \equiv \xi^\mu \partial_\mu$, which are given in the Poincaré coordinates,

$$ds^2 = L^2 \frac{dx^2 + dy^2}{y^2} \,, \tag{2.3}$$

by

$$p = \partial_x \,, \quad d = x\partial_x + y\partial_y \,, \quad k = (x^2 - y^2)\partial_x + 2xy\partial_y \,. \tag{2.4}$$

The associated conserved charges generate the 1d conformal algebra $sl(2,\mathbb{R})$:

$$[D, P] = P \,, \quad [D, K] = K \,, \quad [K, P] = 2D \,. \tag{2.5}$$

Conservation means the surface $\Sigma$ can be freely dragged around and, when it crosses bulk operator insertions, we pick up its $sl(2,\mathbb{R})$ isometry action on the operators (i.e. the charge integrated on small balls around them). For surfaces that touch the AdS boundary $y = 0$, or if the entire surface is pushed toward the boundary, we can also pick up the isometries' actions on any boundary operators that it passes through (see Figure 1). In that case the

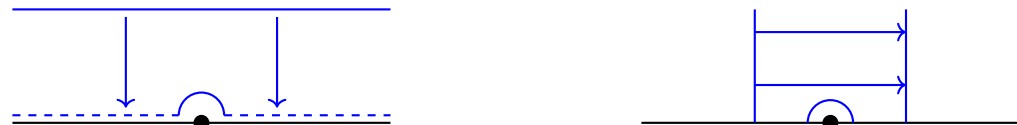

**Figure 1**. Deforming contours both parallel and perpendicular to the boundary of AdS to obtain the action on a local boundary operator. The dashed line emphasizes that the integrand in (2.1) must vanish, up to a total derivative at most, when pushed to the boundary away from operator insertions. Then, a countour ending on the boundary can be deformed as in the right panel.

AdS isometries reduce to boundary conformal transformations acting on the local boundary operators. This construction requires that the flux of the current vanishes when integrated along the boundary of AdS, which is the same as invariance of the boundary conditions under the isometries. In section 2.2, we review why this happens, even when the Boundary Operator Expansion (BOE) of the stress tensor is singular.

In the rest of this section we will generalize the previous discussion to the case of higher-spin currents, and carefully take into account boundary contributions to the charge. This will involve constructing appropriate improvement terms for the higher-spin currents. We will then explicitly construct the higher-spin charges in the examples of the massive free boson and fermion in AdS and compute their action on local operators.

## 2.1 General structure

To streamline notation, let us consider the case of a QFT with a spin-4 conserved current:

$$\nabla^\mu T_{\mu\nu\rho\sigma} = 0\,, \qquad T_{\mu\nu\rho\sigma} = T_{(\mu\nu\rho\sigma)}\,. \tag{2.6}$$

All the equations in this section trivially generalize to any integer spin, and this is done in Section 3.

Conserved charges are obtained, as usual, by contracting the current with a Killing tensor $\zeta^{\mu\nu\rho}$, and integrating:

$$Q_{J=4,\zeta} = \int_\Sigma \sqrt{\gamma}\, dS^\mu\, \zeta^{\nu\rho\sigma}\, T_{\mu\nu\rho\sigma}\,, \tag{2.7}$$

The Killing tensors satisfy

$$\nabla_{(\mu}\zeta_{\nu\rho\sigma)} = 0\,, \tag{2.8}$$

and their space is well understood in AdS [50, 51]. It is spanned by symmetrized products of Killing vectors, i.e.

$$\zeta^{\mu\nu\rho} = \alpha^{ABC}\, \xi_A^{(\mu}\, \xi_B^\nu\, \xi_C^{\rho)}\,, \tag{2.9}$$

where we used the summation convention over the indices $A, B, C$ which run over the Killing vectors $p, d$ and $k$. More generally, in $\mathrm{AdS}_{d+1}$, the space of Killing tensors is classified by two-row Young tableaux of $so(d+2)$ with the same number of boxes in each row [25, 52–54] (while $so(3)$ has no antisymmetric representations). This product structure of the Killing tensors is suggestive of the universal enveloping algebra of $sl(2,\mathbb{R})$, which will repeatedly appear throughout the text. Concretely, in $\mathrm{AdS}_2$ the tensor $\alpha$ in (2.9) can be interpreted as a tensor in embedding space [55, 56]. Correspondingly, the Killing tensors naturally organize into finite-dimensional representations of $sl(2,\mathbb{R}) \simeq so(1,2)$, as we discuss in detail in Appendix C. In the following, when referring to the spin of a Killing tensor, we will always mean the spin of the associated $sl(2,\mathbb{R})$ representation. This number agrees with the top component of the spacetime spin of the tensor $\zeta$.

A useful example is the Killing tensor associated to the cube of boundary translations, which, in Poincaré coordinates, reads

$$\zeta_{p^3}^{\mu\nu\rho} = p^\mu\, p^\nu\, p^\rho = \ \delta^\mu{}_x\, \delta^\nu{}_x\, \delta^\rho{}_x\,. \tag{2.10}$$

If we consider a straight contour anchored at a point $x$ on the boundary of AdS, we have $\sqrt{\gamma} = 1/y$ and $dS^\mu = dy\, n^\mu = y\, dy\, e_x^\mu$, leading to the charge

$$Q_{J=4,p^3} = \int_0^\infty dy\, T_{xxxx}(x,y)\,. \tag{2.11}$$

We will make extensive use of this specific charge throughout the paper, as it is in some sense the simplest we can build from the spin-4 current.

## 2.2 BOE, improvement terms and finiteness of charges

In this subsection, we address the question of finiteness of the higher-spin charges. This issue arises when the surface $\Sigma$ in (2.7) is open and ends on the boundary. A key tool for analysing these possible divergences is the BOE, which allows the bulk higher-spin current to be expanded as a sum of boundary operators—see Section 4.5 for more details. If the current's BOE is not sufficiently suppressed, then its integral (2.7) does not converge. In the following, we argue that *it is always possible to find counterterms which make the higher-spin charges finite, without spoiling their conservation, under a mild assumption on the BOE.* This assumption can be understood as the request that the integrated flux of the current across the whole boundary of AdS vanishes and, in the case of a CFT in AdS, reduces to (generalized) Cardy conditions.

We give a complete proof for currents of rank $J = 1$, 2, 3, 4. In the general case, we prove a weaker version of the statement, which disregards descendants in the BOE of the current. While our considerations generalize to any dimension, we focus on AdS$_2$. We also assume that all components of the current are conserved: as we will see in Section 3.1, this assumption is not restrictive.

Our strategy follows appendix E of [9], which discussed the stress tensor in a manner similar in spirit to a famous argument by Cardy in the context of boundary CFT [57]. Let us start from the contour in Figure 2. Finiteness and conservation are guaranteed, because the surface is compact. We will show under which conditions the contribution of each conformal multiplet in the BOE of the current is purely a total derivative in the horizontal segments of the contour. Then one can replace the horizontal segments by the corresponding boundary terms as in Figure 3, without altering the value and topological properties of the integral. When the endpoints of the arcs are continued to the boundary, the integral is therefore still finite, with the singular contributions from the BOE automatically fine-tuned by the appropriate counterterms. The endpoints can be moved along the boundary without changing the value of the charge, because the arcs still compute the original conserved quantity. This procedure generalizes the construction of [39] to any QFT in AdS and any higher-spin charge.

Let us begin by identifying the conditions under which a higher-spin charge on a surface ending on the boundary is well-defined without counterterms. By the argument above, we must require the contribution of the horizontal segments in Figure 2 to vanish when they are pushed to the boundary. This ensures that the charge is not only finite, but conserved.

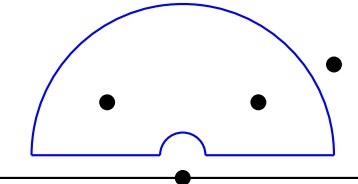

**Figure 2**. In blue, a compact surface $\Sigma$ leading to a higher-spin charge with potentially divergent segments that approach the boundary. Bulk and boundary operators are denoted by black dots.

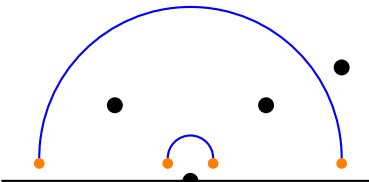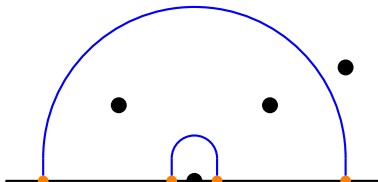

**Figure 3**. On the left, the same contour as Figure 2, with the horizontal segments replaced by their endpoint contributions. On the right, the countours are further pushed towards the boundary, leading to a finite result once the appropriate counterterms are included.

Since any unit vector scales as $n^\mu \sim y$, while $\sqrt{\gamma} \sim 1/y$, one gets the integrand

$$\zeta^{\mu_1 \dots \mu_{J-1}} T_{y\mu_1 \dots \mu_{J-1}} \, dx \ . \tag{2.12}$$

It follows from (2.4) that even the least suppressed Killing tensors are regular at the boundary, and that their non-vanishing boundary components are along the $x$ direction. On the other hand, every $y$ index in $\zeta$ is suppressed by a factor $y$. Then, a rank-$J$ current whose components are all conserved leads to finite charges acting on the boundary Hilbert space, without the need for counterterms, if

$$y^{k-1} \, T_{\underbrace{y \dots y}_{k} x \dots x}(x,y) \to 0 \quad \text{as } y \to 0 \ , \qquad \text{for } k = 1, \dots, J \ . \tag{2.13}$$

To proceed further, we analyze the BOE of the current. We focus on QFTs in AdS invariant under parity $(x \to -x)$, and on parity-even bulk currents.

A parity-even primary can only appear in the BOE of a component with an even number of $x$ indices. On the other hand, for components with an odd number of $x$ indices, the first descendant gives the leading contribution. Hence, we get the strongest condition on the parity-even (odd) primaries by selecting $k$ to be minimal in (2.13), with the constraint that $J - k$ is even (odd). For even $J$, we respectively pick $k = 2$ and $k = 1$ for the parity-even and parity-odd sectors, and vice versa for odd $J$. Let us also recall that a generic operator of scaling dimension $\Delta$ contributes to the BOE a term proportional to $y^{\Delta - J}$. Then, (2.13) translates to

$$\Delta_+ > J - 1 \ , \qquad\qquad \Delta_- > J \ , \qquad\qquad (J \text{ even}) \tag{2.14a}$$

$$\Delta_+ > J \ , \qquad\qquad \Delta_- > J - 1 \ , \qquad\qquad (J \text{ odd}) \tag{2.14b}$$

where the scaling dimensions are those of a primary, whose parity is denoted by the lower index. For the special case of $J = 1$, only $k = 1$ is possible and the parity-odd condition is obtained from the contribution of the first descendant, while the primary does not contribute to (2.12).

If all conditions (2.14) are satisfied, the charge defined by (2.7) is finite on any surface $\Sigma$, including the ones ending on the boundary. However, those conditions are too restrictive. For instance, a free boson in AdS with Neumann boundary conditions corresponds to a GFB with $\Delta_\phi < 1/2$, and the canonical stress tensor contains the relevant parity-even

primary $\phi^2$ in its boundary OPE. This violates (2.14a), but the conformal charges certainly exist. This is the problem solved by [39]—see also earlier discussion in [58]. Similarly, the charges originating from the $J = 4$ current are not naively finite for $\Delta_\phi < 3/2$, and more divergences, corresponding to other two-particle operators, appear as $\Delta_\phi$ decreases or as $J$ increases. Yet, an algebra of higher spin charges does act on the Hilbert space of a GFB, and we now show why.[5]

Let us first discuss the case where $J$ is even. For the parity-odd operators, one can easily convince oneself that the bound (2.14a) is sharp, by contracting the current with the Killing tensor $p^{J-1}$, which corresponds to the $k = 1$ case above. A parity-odd boundary primary $O_-$ appears in the OPE as

$$T_{x...xy}(x, y) \sim b_- \, y^{\Delta_- - J} \, O_-(x) + \dots \, , \qquad J \text{ even} , \qquad (2.15)$$

and therefore we must require $b_-$ to vanish when $\Delta_- \leq J$. There is an interesting constraint on the scaling dimensions allowed in (2.15). As we review in Appendix B, the only parity-odd primaries allowed in the BOE of a rank-$J$ conserved tensor have dimensions $\Delta_- = \ell$, with $\ell = 2, 4, \dots, J$ an even number. In view of (2.14a),

> *Conserving an even higher-spin charge requires that no parity-odd primary appears in the BOE of the current.*

This criterion is satisfied, for example, by free fields in AdS, which will be discussed in the following subsections.

Let us consider the parity-even spectrum, still with $J$ even. Crucially, we will see that the corresponding condition in (2.14a) can be relaxed, at the price of introducing boundary counterterms in the definition of the charge. As we show in Appendix B, the parity-even spectrum consists of operators with unconstrained dimension $\Delta_+ > 0$. Covariant conservation fixes the tensor structure in the OPE up to a single BOE coefficient. Then, we make the following claim: *The conformal multiplet of a parity-even boundary primary contributes a derivative to the flux of the higher-spin current across the boundary, when the latter is contracted with any Killing tensor of rank $J - 1$:*[6]

$$\zeta^{\mu_1...\mu_{J-1}}(x, y) \, T_{\mu_1...\mu_{J-1}y}(x, y)\big|_{O_+} = \partial_x \Big( B_\zeta[y, \partial_x] \, O_+(x) \Big) \, , \qquad J \text{ even} , \qquad (2.16)$$

where the expression on the left hand side is restricted to the contribution of the conformal multiplet of $O_+$, and the operator $B$ on the right hand side resums the descendants. While we do not prove this claim in general, we do prove it up to $J = 4$, and there is no technical obstruction in proceeding to higher rank. In Appendix B, we also show that (2.16) is true for any $J$, at least when restricted to the conformal primary. By the logic described at the beginning of the section, the counterterms to be added to the charge are simply the

---

[5]For the GFB, we also present an alternative argument in Appendix A.1.

[6]The claim is in fact more general: the hodge dual of the conserved current obtained by contracting $\zeta$ and the restriction of $T$ to any parity-even conformal family is an exact form. The claim reported in the main text is sufficient for our purposes.

(singular part of) the boundary terms obtained by integrating the right hand side of (2.16) along $x$.

The proof of (2.16) was given for spin 2 in [59]. The property follows from the existence of an identically conserved tensor:

$$\Delta T_{\mu\nu}(x,y) = \left(g_{\mu\nu}\Box - \nabla_{(\mu}\nabla_{\nu)} - \frac{1}{L^2}g_{\mu\nu}\right)\mathcal{O}(x,y)\,, \qquad \nabla^\mu\Delta T_{\mu\nu} = 0\,, \qquad (2.17)$$

where $\mathcal{O}$ is an arbitrary parity-even bulk scalar. Since the rank of the tensor and its covariant conservation fully fix the BOE, the 'improvement' term (2.17) and the stress tensor have the same BOE in the parity-even sector, up to the coefficient of each primary.[7] Now, a consequence of the identical conservation of (2.17) is that the flux of $\xi^\mu\Delta T_{\mu\nu}$, with $\xi$ a Killing vector, across a surface, only depends on the surface's boundary: in other words, $\sqrt{\gamma}\xi^\mu n^\mu\Delta T_{\mu\nu}$ is a derivative in the direction orthogonal to $n^\mu$. This proves the property (2.16).

It is clear that the generalization of this proof only requires finding identically conserved symmetric tensors of the appropriate rank. For $J = 4$, starting from an ansatz with at most four derivatives, and setting $L = 1$ henceforth, we obtain

$$\Delta T_{\mu\nu\rho\sigma}(x,y) = \left(g_{(\mu\nu}g_{\rho\sigma)}\left(1 - \frac{4}{3}\Box + \frac{1}{9}\Box^2\right) + g_{(\mu\nu}\nabla_\rho\nabla_{\sigma)}\left(\frac{14}{9} - \frac{2}{9}\Box\right)\right.$$
$$\left. + \frac{1}{9}\nabla_{(\mu}\nabla_\nu\nabla_\rho\nabla_{\sigma)}\right)\mathcal{O}(x,y)\,. \qquad (2.18)$$

The existence of this operator proves equation (2.16) for $J = 4$.

We now move on to the case of a current with odd rank. In this case, the role of parity-even and odd boundary operators is reversed. Indeed, the exchange of parity-**even** operators is now highly restricted. Compatibility with conservation of the current ensures that the only parity-even primaries allowed in the BOE have dimensions $\Delta_+ = \ell$, with $\ell = 1, 3, \ldots, J$ and $J$ now an odd number. For finiteness of the higher-spin charges, we require that these are absent.

Instead, the exchange of parity-odd operators is allowed for any scaling dimension $\Delta_- > 0$, which can be in tension with (2.14b). Nonetheless it is possible to define the charge in the presence of such operators by again noting that the flux of the current is a total derivative

$$\zeta^{\mu_1\ldots\mu_{J-1}}(x,y)\,T_{\mu_1\ldots\mu_{J-1}y}(x,y)\big|_{O_-} = \partial_x\left(B'_\zeta[y,\partial_x]O_-(x)\right)\,, \qquad J \text{ odd}\,. \qquad (2.19)$$

This is verified for all $J$ at the level of primaries in Appendix B, and we explicitly prove it for $J = 1, 3$ by producing identically conserved tensors as we did in the even-spin case. For the rank-1 case, the epsilon tensor (normalized such that $\varepsilon^{xy} = 1$) allows us to write

$$\Delta T_\mu(x,y) = \sqrt{g}\,\varepsilon_{\mu\nu}\nabla^\nu\mathcal{O}_-(x,y)\,, \qquad (2.20)$$

---

[7]Notice that, instead, $\Delta T_{\mu\nu}$ does not couple to parity-odd boundary operators.

for an arbitrary parity-odd bulk scalar $O_-$. It is then possible to use this basic object to build higher-spin improvements. For spin 3 we find

$$\Delta T_{\mu\nu\rho}(x,y) = \left(\nabla_{(\mu}\nabla_\nu - g_{(\mu\nu}\Box + 4\,g_{(\mu\nu}\right)\Delta T_{\rho)}\,, \tag{2.21}$$

and it would be straightforward to proceed to higher rank.

As a final comment, let us discuss the case of a CFT in AdS, i.e. a boundary CFT. There, the higher spin currents are traceless, and one can easily see that the only boundary operators allowed by conservation have $\Delta = J$, in accordance with the fact that the BOE of a holomorphic operator can never be singular. Therefore, only the parity-odd primaries saturate the bound (2.14a) and only the parity-even ones saturate the bound (2.14b). The request that their contribution vanishes is then simply equivalent to the statement that

$$T_{\text{z}...\text{z}}(x,0) - T_{\bar{\text{z}}...\bar{\text{z}}}(x,0) = 0\,, \tag{2.22}$$

where we picked complex coordinates $\text{z} = x + iy$, $\bar{\text{z}} = x - iy$. Notice that, since parity swaps $\text{z} \leftrightarrow -\bar{\text{z}}$, the combinations in (2.22) have the correct parity. The property (2.22) is well known as the Cardy gluing condition.

## 2.3   Generalized Free Boson (GFB)

Having established the conditions under which the higher-spin charges are well-defined, we now proceed to illustrate them in the 1d generalized free boson theory, which is simply the boundary description of a free massive boson $\Phi$ in AdS$_2$, with the action

$$S = \frac{1}{2}\int d^2x\sqrt{g}\left(\partial_\mu\Phi\partial^\mu\Phi + m^2\Phi^2\right) = \frac{1}{2}\int dxdy\left((\partial_x\Phi)^2 + (\partial_y\Phi)^2 + \frac{m^2}{y^2}\Phi^2\right)\,, \tag{2.23}$$

where we specialized to Poincaré coordinates after the second equality. The existence of higher-spin charges in this theory stems from the fact that it possesses one conserved current at each even spin. The first of these is of course the stress-tensor, whose canonical form is

$$T_{\mu\nu} = \partial_\mu\Phi\partial_\nu\Phi - \frac{1}{2}g_{\mu\nu}\left(\partial_\rho\Phi\partial^\rho\Phi + m^2\Phi^2\right)\,. \tag{2.24}$$

As we discussed above, conserved currents are only defined up to improvement terms and quantities proportional to equations of motion. For example, another possible stress-tensor is given by

$$T'_{\mu\nu} = -\Phi\overleftrightarrow{\nabla}_{(\mu}\overleftrightarrow{\nabla}_{\nu)}\Phi - g_{\mu\nu}\Phi^2\,, \tag{2.25}$$

which has the advantage of eliminating the explicit mass dependence. It also introduces the left-right derivative, which acts as $A\overleftrightarrow{\nabla}_\mu B = (\nabla_\mu A)B - A(\nabla_\mu B)$. In fact, following [60], it is possible to write the spin-4 current in a similar manner, namely

$$T_{\mu\nu\rho\sigma} = \Phi\overleftrightarrow{\nabla}_{(\mu}\overleftrightarrow{\nabla}_\nu\overleftrightarrow{\nabla}_\rho\overleftrightarrow{\nabla}_{\sigma)}\Phi + 14g_{(\mu\nu}\Phi\overleftrightarrow{\nabla}_\rho\overleftrightarrow{\nabla}_{\sigma)}\Phi + 9g_{(\mu\nu}g_{\rho\sigma)}\Phi^2\,. \tag{2.26}$$

There are similar expressions for currents of arbitrary spin $J$ in terms of left-right derivatives and metrics, with coefficients given in closed form in [60]. Importantly, this current is only conserved on-shell, unlike the identically conserved improvement terms discussed

above. For the case of the complex scalar, it is also possible to write down odd-spin currents. For example, the simplest one of spin-3, is given by

$$T_{\mu\nu\rho} = -i\Phi \overleftrightarrow{\nabla}_{(\mu} \overleftrightarrow{\nabla}_{\nu} \overleftrightarrow{\nabla}_{\rho)} \Phi^* - 5ig_{(\mu\nu}\Phi \overleftrightarrow{\nabla}_{\rho)}\Phi^* . \tag{2.27}$$

In the following, we will revert to a real scalar every time we consider even spin charges. A complex scalar, on the other hand, has three currents at each even spin, distinguished by their $U(1)$ charge.

In the context of the free boson, it is clear how to determine the action of charges on boundary operators. One computes a 'form factor' for the conserved current, for example by inserting two boundary operators and the bulk current, and then integrates the bulk operator over a co-dimension one surface. A convenient choice, depicted in Figure 4, uses a straight contour anchored at the boundary, i.e. a constant $x$ slice. We pick Dirichlet

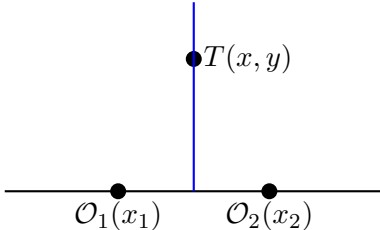

**Figure 4**. Poincaré AdS configuration to compute the matrix element of a charge between two boundary operators $\mathcal{O}_1$ and $\mathcal{O}_2$. The bulk current $T$ is integrated over its $y$ coordinate.

boundary conditions, under which the AdS dictionary equation,

$$m^2 = \Delta_\phi(\Delta_\phi - 1), \tag{2.28}$$

is solved in the branch

$$\Delta_\phi = \frac{1 + \sqrt{1 + 4m^2}}{2}. \tag{2.29}$$

In the previous equations, we denoted by $\Delta_\phi$ the scaling dimension of the leading Dirichlet mode $\phi$. The Dirichlet branch is compatible with taking the flat space limit $m \to \infty$. The bulk-to-boundary propagator between the bulk field and its leading Dirichlet mode, normalized to have unit two-point function, reads

$$\langle \phi(x_1)\Phi(x,y)\rangle = K_{\Delta_\phi}(x_1; x, y) = \sqrt{N_{\Delta_\phi}} \left(\frac{y}{y^2 + (x - x_1)^2}\right)^{\Delta_\phi}, \tag{2.30}$$

with

$$N_{\Delta_\phi} = \frac{\Gamma(\Delta_\phi)}{\sqrt{\pi}\,(2\Delta_\phi - 1)\,\Gamma(\Delta_\phi - \frac{1}{2})}. \tag{2.31}$$

As a warm-up, let's now consider the simplest possible action of a charge: the translation generator $P$ sandwiched between two boundary single-particle states $\phi$. This is given by

$$\langle \phi(x_1)\,P\,\phi(x_2)\rangle = \int_0^\infty dy\, \langle \phi(x_1)\,T_{xx}(x,y)\,\phi(x_2)\rangle. \tag{2.32}$$

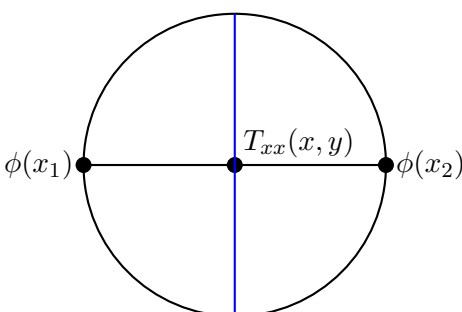

**Figure 5**. Witten diagram describing the action of the momentum generator $P$ on the two boundary operators. The black lines denote bulk-boundary propagators and the blue line denotes the region of the co-dimension 1 integral over the position of $T_{xx}$.

Due to $T$ being a quadratic operator, the computation can be mapped to the Witten diagram in Figure 5. The explicit expression for the integrand is given by an appropriate combination of derivatives of the bulk-to-boundary propagator:[8]

$$\langle \phi\, T_{xx}\, \phi \rangle = N_{\Delta_\phi} \Big( \partial_x K_{\Delta_\phi}(x_2,x,y)\partial_x K_{\Delta_\phi}(x_1,x,y) - \partial_y K_{\Delta_\phi}(x_2,x,y)\partial_y K_{\Delta_\phi}(x_1,x,y)$$
$$- \frac{\Delta_\phi(\Delta_\phi - 1)}{y^2} K_{\Delta_\phi}(x_1,x,y)K_{\Delta_\phi}(x_2,x,y) \Big)\,, \tag{2.33}$$

which upon integration simply gives

$$\langle \phi(x_1)\, P\, \phi(x_2) \rangle = -\frac{2\Delta_\phi}{x_{12}^{2\Delta_\phi}} = \partial_{x_2} \langle \phi(x_1)\phi(x_2) \rangle\,, \tag{2.34}$$

when $x_1 < x < x_2$ and 0 whenever $x > x_2$ or $x < x_1$. This is straightforward to interpret: when the charge is inserted between the operators we can wrap it to act, say on $\phi(x_2)$ and we realize the action of the translation generator

$$[P, \phi(x_2)] = \partial_{x_2}\phi(x_2)\,. \tag{2.35}$$

Instead, when the charge is either to the left or the right of both operators, we simply act on the vacuum, which is translationally invariant, leading to the vanishing of the correlator. As follows from the analysis in Section 2.2, the vacuum of the free scalar is also invariant under the higher-spin symmetries, so we expect the answer to have the same structure when we replace $P$ with a higher-spin charge.

Let us indeed move on to the action of the simplest among them, in the case of the free complex scalar. Using the spin-3 current, we define

$$Q_{J=3,p^2} = \int_0^\infty dy\, T_{xxx}(x,y)\,, \tag{2.36}$$

where we again made use of the Killing tensor $\zeta^{\mu\nu} = p^\mu p^\nu$. We can now repeat the previous computation, sandwiching the higher spin charge between the boundary mode $\phi$ and its

---

[8]In performing these explicit curved-space computations, we used the Mathematica package *RGTC*.

complex conjugate:

$$\langle\phi(x_1)\,Q_{J=3,p^2}\,\phi^*(x_2)\rangle = \int_0^\infty dy\,\langle\phi(x_1)\,T_{xxx}(x,y)\,\phi^*(x_2)\rangle\,. \tag{2.37}$$

Since the current operator is linear in $\phi$ and $\phi^*$ there is again a unique Wick contraction, and the computation consists of integrating a product of two bulk-to-boundary propagators, appropriately dressed with covariant derivatives. Once the dust settles, we find

$$\langle\phi(x_1)\,Q_{J=3,p^2}\,\phi^*(x_2)\rangle = \langle\phi^*(x_1)\,Q_{J=3,p^2}\,\phi(x_2)\rangle^* = 4i\frac{2\Delta_\phi(2\Delta_\phi+1)}{x_{12}^{2\Delta_\phi+2}}\,, \tag{2.38}$$

when $x_1 < x < x_2$, and 0 when $x > x_2$ or $x < x_1$, which is consistent with higher-spin invariance of the vacuum. Remarkably, the action on $\phi$ coincides with that of a second derivative

$$\langle\phi(x_1)\,Q_{J=3,p^2}\,\phi^*(x_2)\rangle = 4i\,\partial_{x_2}^2\langle\phi(x_1)\phi^*(x_2)\rangle\,, \tag{2.39}$$

determining the action of the charge on the boundary modes

$$[Q_{J=3,p^2},\phi(x)] = -4i\partial_x^2\phi(x)\,, \quad [Q_{J=3,p^2},\phi^*(x)] = 4i\partial_x^2\phi^*(x)\,, \tag{2.40}$$

which is compatible with charge conjugation. The action is also compatible with naive dimensional analysis: it increases the weight of the operator it acts on by two, generating the level-two descendant of $\phi$.[9] We also note that the presence of the factor of $i$ in (2.40) follows from reality of the charge, while the factor of 4 can be changed by normalizing the current differently. At this stage, it seems the higher-spin charge might simply act as a power of the conformal generators, i.e, $Q_{J=3,p^2} \sim P^2$. Fortunately, the action is much more interesting than that. To see this, let us try to act on two-particle boundary operators, e.g. $\phi^2$. Dimensional analysis implies

$$[Q_{J=3,p^2},\phi^2(x)] = \alpha\,\partial^2(\phi^2(x)) + \beta\,\mathcal{O}_{2,2}(x)\,, \tag{2.41}$$

where $\mathcal{O}_{2,2}$ is the conformal primary of dimension $2\Delta_\phi + 2$, given in some normalization by

$$\mathcal{O}_{2,2} = \partial\phi\partial\phi - \frac{2\Delta_\phi}{1+2\Delta_\phi}\phi\partial^2\phi\,. \tag{2.42}$$

To determine the coefficients $\alpha$ and $\beta$ in (2.41), we can compute the two form factors

$$\langle\mathcal{O}_{2,2}^*\,Q_3\,\phi^2\rangle\,, \quad \langle(\phi^*)^2\,Q_3\,\phi^2\rangle\,, \tag{2.43}$$

and, after carefully dividing by the norms of the two operators, read off the ratio $\beta/\alpha$. The computation of the integrated form factor is straightforward. Since the current is quadratic in $\phi$, it can only connect to one field in each of the composite boundary propagators. Hence, the answer is a product of (derivatives of) the single $\phi$ form factor, computed previously, and free boundary propagators. This is schematically depicted in the Witten diagram of Figure 6. It turns out that both form-factors are non-vanishing and the ratio of coefficients

---

[9]The reader might complain that there could be other terms in the action (2.40) which are projected out due to orthogonality of the two-point function. For generic $\Delta_\phi$ this can be excluded on the basis of dimensional analysis. In Section 4, we will exclude this possibility in full generality using Ward Identities.

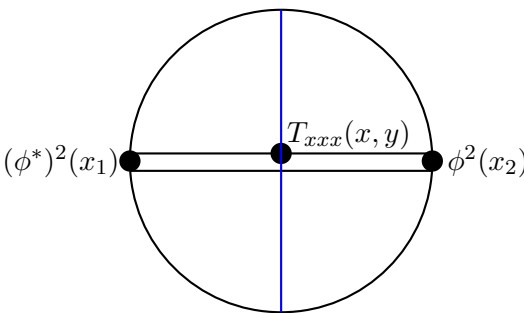

**Figure 6**. Witten diagram describing the action of the momentum generator $Q_{J=3,p^2}$ on the two composite boundary operators. Since the current is quadratic in the fields it can only attach to one of the constituents of the boundary $\phi^2$, leaving a free propagator between the ones that remain.

is $\beta/\alpha = -2$. Notably, this means that the action of the higher-spin charge on two-particle operators generates new primaries. As we will argue on general grounds in Section 4, higher-spin charges carry integer scaling dimension: this gives a justification for the integer spacing of primary scaling dimensions in GFB, using higher-spin symmetry. Furthermore, by looking at the actual form of the resulting operator (up to some normalization $k_{\Delta_\phi}$)

$$[Q_{J=3,p^2}, \phi^2(x)] = k_{\Delta_\phi} \phi \partial^2 \phi \,, \tag{2.44}$$

we observe that the charge acts as $P^2$ but on each individual constituent of the multi-particle operator. This could have also been anticipated by the Witten diagram computation, since only one boundary field at a time is acted upon by the bulk current. This feature is analogous to the action of higher-spin charges on multi-particle scattering states in flat space [19]:

$$Q_{J=3,p^2}|p_1, p_2, \ldots, p_n\rangle = (p_1^2 + p_2^2 + \cdots + p_n^2)|p_1, p_2, \ldots, p_n\rangle \,. \tag{2.45}$$

Crucially, this differs from $(p_1 + p_2 + \cdots + p_n)^2$, and in fact this property is typically used to prove factorization of the multi-particle S-matrix (see, e.g. [61]).

The case of the spin-4 current proceeds in complete analogy. Computing the correlator of the charge $Q_{J=4,p^3}$ with real $\phi$'s, one finds

$$\langle \phi(x_1) Q_{J=4,p^3} \phi(x_2)\rangle = 2 \cdot 8 \, \partial_{x_2}^3 \langle \phi(x_1)\phi(x_2)\rangle \,, \tag{2.46}$$

meaning that the action on $\phi$ reads

$$[Q_{J=4,p^3}, \phi(x)] = 2 \cdot 8 \, \partial_x^3 \phi(x) \,. \tag{2.47}$$

More generally, naive dimensional analysis allows for multiple operators to appear. For the two-particle operator, a straightforward computation gives

$$[Q_{J=4,p^3}, \phi^2(x)] = k'_{\Delta_\phi} \phi \partial^3 \phi \,, \tag{2.48}$$

which once again respects the structure of acting constituent-wise on composite operators. The result has overlap with both $\partial^3(\phi^2)$ and $\partial\mathcal{O}_{2,2}$—see (2.42)—meaning we are once again able to generate new primaries with the action of the higher-spin charge.

Turning to charges with general higher-spin, the spin-$J$ charge built from $p^{J-1}$ must act by dimensional analysis as

$$[Q_{J,p^{J-1}}, \phi] \propto \partial^{J-1}\phi . \tag{2.49}$$

Furthermore, it is easy to see that any higher-spin charge acts constituent-wise. These two facts imply that one can build the full set of $n$-particle operators by acting with higher-spin charges on $\phi^n$. In other words, every sector with fixed particle number forms a higher-spin irreducible representation. This is understood in terms of higher-spin representation theory (see for example the AdS$_2$ version of the Flato-Fronsdal theorem [62] discussed in [63]), but let us see this explicitly. For simplicity, consider the two-particle sector. Since the charge $Q_{J,p^{J-1}}$ acts constituent-wise, we can generate $\phi \partial_x^{J-1}\phi$ acting on $\phi^2$. From (2.49), it also follows that $[Q_{J,p^{J-1}}, \partial_x^m \phi] \propto \partial_x^{J-1+m}\phi$. Then, acting on the subspace $\phi\partial_x^m\phi$, we get $\partial_x^m\phi\,\partial_x^{J-1}\phi + \phi\,\partial_x^{J-1+m}\phi$. The second term can be subtracted off using $[Q_{J+m,p^{J+m-1}}, \phi^2]$, and by arbitrariness of $m$ and $J$, all the two-particle Fock space has been generated.

Let us briefly comment on the action of charges obtained with more complicated Killing tensors. For this, let us go back to the spin-3 current of the complex boson and consider the Killing tensor

$$\zeta_{pd}^{\mu\nu} = p^{(\mu}d^{\nu)} = \delta_x^{(\mu}x^{\nu)} , \tag{2.50}$$

where $x^0 = x$, $x^1 = y$, which is obtained as a symmetrized product of the Killing vectors associated to translations and dilations. We then use the general recipe (2.7) to obtain

$$Q_{J=3,pd} = \int_0^\infty dy\,(x\,T_{xxx} + y\,T_{xxy}) , \tag{2.51}$$

and compute the usual integrated form-factors. This allows one to read-off the action of the charge

$$[Q_{J=3,pd}, \phi] \propto (DP + PD)\phi . \tag{2.52}$$

These charges act in general on the GFB $\phi$ as the corresponding elements of the universal enveloping algebra (UEA) of $sl(2,\mathbb{R})$ [63], i.e. as sums of products of conformal generators. For example, we also find

$$[Q_{J=3,kd}, \phi] \propto (DK + KD)\phi , \quad [Q_{J=3,k^2}, \phi] \propto K^2\phi . \tag{2.53}$$

There is a slight ambiguity in translating Killing tensors to UEA elements, namely that weight-zero elements can be shifted by a central term:

$$\big[Q_{3,d^2}, \phi\big] \propto (D^2 + c_1)\phi , \quad [Q_{3,pk}, \phi] \propto (PK + KP + c_2)\phi , \tag{2.54}$$

for some constants $c_{1,2}$. This can be better understood by taking into account the precise definition of the algebra of symmetries of the generalized free field [63]. This is given by the universal enveloping algebra of $sl(2,\mathbb{R})$, modded out by the identification of the Casimir operator $\mathcal{C}_2 = D^2 - (PK + KP)/2$ with its constant eigenvalue in any given irreducible representation, known as $gl[\lambda]$ (where $\lambda$ is related to the mass). This is not a simple lie Algebra and is known to contain a central $u(1)$ piece, in accordance to the action we found

in (2.54). The simple higher-spin algebra $hs[\lambda]$ is defined by removing the commuting factor from $gl[\lambda]$. We will give further details on this algebra in Section 4.1.

In summary of our results for the generalized free boson, we explicitly determined the action of the $p^3$ charge on the single-particle state (2.47) as well as its spin-$J$ generalization (2.49). Other charges act on single-particle states according to the UEA. We also explained that, since the higher-spin currents are always quadratic, they act constituent-wise on multi-particle states and hence preserve particle number. For a fixed number of particles, the states have integer-spaced scaling dimensions, and furnish a single higher-spin representation.

### 2.4 Generalized Free Fermion (GFF)

Along with the free massive boson dual to a generalized free boson, we can also find a HS symmetric theory by considering a free massive Majorana fermion in $AdS_2$, dual to a generalized free fermion. In flat space a massive Majorana fermion has one conserved current for each even spin [64]. Presumably the same is true for the theory in AdS, but we will content ourselves with writing down the spin-4 current which, as far as we know, has not appeared elsewhere in the literature. We will write the two-component fermion $\Psi$ and its conjugate $\bar{\Psi}$ as

$$\Psi = \begin{pmatrix} \psi \\ -i\bar{\psi} \end{pmatrix}, \quad \bar{\Psi} = (i\bar{\psi}, \, \psi) \tag{2.55}$$

with the two Grassmann field $\psi$ and $\bar{\psi}$ identified on the boundary through the relation $\psi = -\bar{\psi}$ leaving a single degree of freedom that we denote by $\hat{\psi}$. The action can then be written as

$$\int d^2x \sqrt{g}\, \bar{\Psi}(\gamma^\mu \nabla_\mu - m)\Psi = 2 \int \frac{dxdy}{y^2}(y\,\psi\bar{\partial}\psi + y\,\bar{\psi}\partial\bar{\psi} - im\bar{\psi}\psi)\,, \tag{2.56}$$

where $\gamma^\mu$ are the curved space gamma matrices and $\nabla_\mu$ is the fermionic covariant derivative which includes the spin connection; see [9,65] for additional details on the Majorana fermion in (curved) $AdS_2$. The stress tensor, written on-shell has the simple form

$$T_{\mu\nu} = \frac{1}{2}\bar{\Psi}\gamma_{(\nu}\nabla_{\mu)}\Psi + mg_{\mu\nu}\bar{\Psi}\Psi. \tag{2.57}$$

We then found the following spin-4 conserved current

$$\begin{aligned}
T_{\mu\nu\rho\sigma} = \frac{8}{5}\Big( &\bar{\Psi}\gamma_{(\mu}\nabla_\nu\nabla_\rho\nabla_{\sigma)}\Psi - 3\nabla_{(\nu}\bar{\Psi}\gamma_\mu\nabla_\rho\nabla_{\sigma)}\Psi - 15\nabla_{(\rho}\nabla_\nu\bar{\Psi}\gamma_\mu\nabla_{\sigma)}\Psi - \nabla_{(\sigma}\nabla_\rho\nabla_\nu\bar{\Psi}\gamma_{\mu)}\Psi \\
&+ 6g_{(\mu\nu}\nabla_\rho\nabla^\alpha\bar{\Psi}\gamma_{\sigma)}\nabla_\alpha\Psi + 9mg_{(\mu\nu}\bar{\Psi}\nabla_\rho\nabla_{\sigma)}\Psi - 3mg_{(\mu\nu}\nabla_\rho\bar{\Psi}\nabla_{\sigma)}\Psi \\
&+ (\frac{59}{2} - 18m^2)g_{(\mu\nu}\bar{\Psi}\gamma_\rho\nabla_{\sigma)}\Psi \Big). 
\end{aligned} \tag{2.58}$$

We note that this contains three-, two- and one-derivative terms, meaning there is an expansion in power of $1/(mL)$ around the flat space limit. This is contrast with the bosonic case where only even powers appeared. We can then integrate the current to determine the action of the higher-spin charge on the boundary generalized free fermion $\hat{\psi}$

$$\langle \hat{\psi}(x_1)\, Q_{J=4,p^3}\, \hat{\psi}(x_2)\rangle = \int_0^\infty dy\, \langle \hat{\psi}(x_1)\, T_{xxxx}(x,y)\, \hat{\psi}(x_2)\rangle\,. \tag{2.59}$$

The computation proceeds as in the bosonic case, but we must now make use of the pair of fermionic bulk-to-boundary propagators

$$\left\langle \psi\left(x_1, y_1\right) \hat{\psi}\left(x_2\right) \right\rangle = 2\sqrt{C_m} \frac{\sqrt{y_1}}{\left(x_1 - x_2\right) + i y_1} \left( \frac{2 y_1}{\left(x_1 - x_2\right)^2 + y_1^2} \right)^m , \qquad (2.60)$$

$$\left\langle \bar{\psi}\left(x_1, y_1\right) \hat{\psi}\left(x_2\right) \right\rangle = -2\sqrt{C_m} \frac{\sqrt{y_1}}{\left(x_1 - x_2\right) - i y_1} \left( \frac{2 y_1}{\left(x_1 - x_2\right)^2 + y_1^2} \right)^m , \qquad (2.61)$$

with $\Delta_\psi = m + 1/2$ the dimension of $\hat{\psi}$ and

$$C_m \equiv \frac{\Gamma(m+1)}{\sqrt{\pi}\, 4^{2+m} \Gamma(m + \frac{1}{2})} . \qquad (2.62)$$

We then find exactly the same answer as for the boson

$$\langle \hat{\psi}(x_1) Q_{J=4, p^3} \hat{\psi}(x_2) \rangle = 2 \cdot 8\, \partial_{x_2}^3 \langle \hat{\psi}(x_1) \hat{\psi}(x_2) \rangle , \qquad (2.63)$$

which means that the charge acts as

$$[Q_{J=4, p^3}, \hat{\psi}(x)] = 2 \cdot 8\, \partial_x^3 \hat{\psi}(x) . \qquad (2.64)$$

It is then straightforward to recover the action on two-particle operators. Using the same argument as for the boson, one finds that the charge acts on each constituent of the composite operator at a time. For the simplest two-particle state $\hat{\psi}\partial\hat{\psi}$, we have

$$[Q_{J=4, p^3}, \hat{\psi}\partial\hat{\psi}(x)] = k_\psi \left( \hat{\psi}\partial^4 \hat{\psi}(x) + \partial\hat{\psi}\partial^3\hat{\psi}(x) \right) . \qquad (2.65)$$

It is easy to see that not only a level-3 descendant of $\hat{\psi}\partial\hat{\psi}$ is generated, but also a level-1 descendant of $\mathcal{O}_{2,3}$ the primary of dimension $2\Delta_\psi + 3$

$$[Q_{J=4, p^3}, \hat{\psi}\partial\hat{\psi}(x)] = \alpha\, \partial^3(\hat{\psi}\partial\hat{\psi}) + \beta\, \partial(\mathcal{O}_{2,3}) . \qquad (2.66)$$

This follows from the fact that (2.65) is not proportional to $\partial^3(\hat{\psi}\partial\hat{\psi}) = (\hat{\psi}\partial^4\hat{\psi}(x) + 4\,\partial\hat{\psi}\partial^3\hat{\psi}(x))$ and that the space of dimension $2\Delta_\psi + 4$ operators is two-dimensional. The fact that the action (2.66) mixes different conformal primaries offers higher-spin symmetry as an explanation for the integer spacing between them in GFF theory.

# 3 Systematics of higher-spin conservation in AdS$_2$

Having discussed the higher-spin symmetry of free fields, in this section we will consider conservation properties of higher-spin currents and charges for general QFTs in AdS$_2$. First, in Section 3.1 we will prove that, unlike flat space, it is impossible in AdS$_2$ to conserve only a part of a higher-spin current. Then in Section 3.2, we will reconsider higher-spin charges and characterise their conservation as a Heisenberg-type equation.

### 3.1 No 'partial' higher-spin conservation in AdS$_2$

We would like to consider deforming free fields in AdS$_2$ by interactions that preserve a part of their higher-spin currents. In flat space, it is possible to conserve only the particular (anti)holomorphic components,

$$\partial^\mu T_{\mu zz\cdots z} = 0 \ , \qquad \partial^\mu T_{\mu \bar{z}\bar{z}\cdots \bar{z}} = 0 \ , \qquad \text{(flat space)} \tag{3.1}$$

and this is what happens for well-known integrable models such as sine-Gordon and Toda field theories.[10] Here $z$ and $\bar{z}$ are the (anti-)holomorphic coordinates of flat 2d space,

$$ds^2 = dz\, d\bar{z} \ . \tag{3.2}$$

These conservation equations are invariant under flat-space isometries: they are clearly translation-invariant, and are covariant under rotations $(z, \bar{z}) \to (e^{i\alpha} z, e^{-i\alpha}\bar{z})$.

This 'partial' conservation is naturally formulated in our language in the form

$$\nabla^\mu \big( \zeta^{\nu_1 \cdots \nu_{J-1}} T_{\mu\nu_1\cdots\nu_{J-1}} \big) = 0 \tag{3.3}$$

for a symmetric tensor $T_{\mu\nu_1\cdots\nu_{J-1}} = T_{(\mu\nu_1\cdots\nu_{J-1})}$ contracted with some Killing tensor $\zeta^{\nu_1\cdots\nu_{J-1}}$. The flat-space example (3.1) above corresponds to the choices

$$\zeta^{\nu_1\cdots\nu_{J-1}} = \delta^{\nu_1}{}_z \cdots \delta^{\nu_{J-1}}{}_z \qquad \text{and} \qquad \zeta^{\nu_1\cdots\nu_{J-1}} = \delta^{\nu_1}{}_{\bar{z}} \cdots \delta^{\nu_{J-1}}{}_{\bar{z}} \ . \tag{3.4}$$

We will now show that such a 'partial' conservation is impossible in AdS$_2$ space. If one assumes the partial conservation (3.3) of a higher-spin current along any single Killing tensor of AdS$_2$, then the current is automatically fully conserved:

$$\nabla^\mu T_{\mu\nu_1\cdots\nu_{J-1}} = 0 \ . \tag{3.5}$$

We will argue by acting with AdS isometries on the partial conservation equation (3.3) in order to obtain the same equation with different choices of Killing vectors, eventually producing enough partial conservation equations to conclude full conservation. The key difference that makes this argument possible in AdS, but not in flat space, is that the AdS isometry algebra $sl(2, \mathbb{R})$ is simple, while the 2d Poincaré group is not. This is related to the existence of the special invariant Killing tensors (3.4) in flat space that cannot be related to anything else using isometries.

Without loss of generality, we will assume the partial conservation (3.1) for a Killing tensor $\zeta^{\nu_1\cdots\nu_{J-1}}$ that contains at least some traceless part. Otherwise it would be factorisable into a lower-rank tensor and a metric factor,

$$\zeta^{\nu_1\cdots\nu_{J-1}} = g^{(\nu_1\nu_2} \tilde{\zeta}^{\nu_3\cdots\nu_{J-1})} \tag{3.6}$$

and the current would be effectively traced down to a lower-spin one,

$$\nabla^\mu \big( \tilde{\zeta}^{\nu_3\cdots\nu_{J-1}} \tilde{T}_{\mu\nu_3\cdots\nu_{J-1}} \big) = 0 \ , \qquad \tilde{T}_{\mu\nu_3\cdots\nu_{J-1}} := g^{\nu_1\nu_2} T_{\mu\nu_1\cdots\nu_{J-1}} \ . \tag{3.7}$$

---

[10]For example, the sine-Gordon model preserves the components (3.1) of currents with all even spin [66]. A richer structure is exhibited by Toda theories [67, 68].

In the representation (2.9) of AdS$_2$ Killing tensors as linear combinations of symmetrised products of the Killing vectors $p, d, k$, this condition translates to the assumption that the Killing tensor is not 'divisible' by the quadratic Casimir $C_2 = d^2 - \frac{1}{2}(pk + kp)$, since the corresponding Killing tensor $d^\mu d^\nu - k^{(\mu} p^{\nu)} \propto g^{\mu\nu}$. As discussed around (2.9) and in Appendix C, Killing tensors belong to representations of $sl(2, \mathbb{R})$: the assumption above guarantees that the maximal spin $J - 1$ representation[11] appears in the decomposition of $\zeta^{\nu_1 \cdots \nu_{J-1}}$. This single condition suffices, because the metric is the only tensor invariant under all isometries, or equivalently, there are no higher Casimirs of $sl(2, \mathbb{R})$.

Let us begin by considering how isometries act on the equation (3.3). For an isometry with Killing vector $\xi$, let $Q_\xi$ denote the associated conserved charge. Considering (3.3) as an operator equation and taking its commutator with $Q_\xi$ gives

$$\zeta^{\nu_1 \cdots \nu_{J-1}} \left[ Q_\xi, \nabla^\mu T_{\mu\nu_1 \cdots \nu_{J-1}} \right] = 0 \ , \tag{3.8}$$

where the $c$-number $\zeta^{\nu_1 \cdots \nu_{J-1}}$ can be pulled outside, recalling that Killing tensors satisfy $\nabla^{(\mu} \zeta^{\nu_1 \cdots \nu_{J-1})} = 0$. We know that $Q_\xi$ acts on the local tensor $\nabla^\mu T_{\mu\nu_1 \cdots \nu_{J-1}}$ as the Lie derivative $\mathcal{L}_\xi$, giving

$$\zeta^{\nu_1 \cdots \nu_{J-1}} \mathcal{L}_\xi \left( \nabla^\mu T_{\mu\nu_1 \cdots \nu_{J-1}} \right) = 0 \ . \tag{3.9}$$

On the other hand, simply redefining the coordinates as $\delta x^\mu = \xi^\mu$ leads to a transformation of the whole quantity as a local tensor (including $\zeta$),

$$\mathcal{L}_\xi \left( \zeta^{\nu_1 \cdots \nu_{J-1}} \nabla^\mu T_{\mu\nu_1 \cdots \nu_{J-1}} \right) = 0 \ . \tag{3.10}$$

Subtracting (3.10) from (3.9) and using the Leibniz identity gives

$$\left( \mathcal{L}_\xi \zeta^{\nu_1 \cdots \nu_{J-1}} \right) \nabla^\mu T_{\mu\nu_1 \cdots \nu_{J-1}} = 0 \ . \tag{3.11}$$

Clearly $\mathcal{L}_\xi \zeta$ is also a Killing tensor and we have produced a new partial conservation equation.

We now need to explore which Killing tensors can be produced by applying $\mathcal{L}_\xi$ to an arbitary Killing tensor $\zeta$. This is essentially a counting problem. The rank-$(J - 1)$ Killing tensors are naturally seen as the elements of the universal enveloping algebra of $sl(2, \mathbb{R})$ consisting of $(J - 1)$-fold products. The Lie derivative $\mathcal{L}_\xi$ acts on them by the usual Lie bracket. Modulo the quadratic Casimir $C_2$, i.e. ignoring all elements divisible by $C_2$, these Killing tensors form the $(2J - 1)$-dimensional representation of $sl(2, \mathbb{R})$,

$$p^{J-1} \quad \xrightarrow{[k,-]} \quad \mathrm{ad}_k \, p^{J-1} \quad \xrightarrow{[k,-]} \quad \cdots \quad \xrightarrow{[k,-]} \quad (\mathrm{ad}_k)^{2(J-1)} \, p^{J-1} \propto k^{J-1} \ , \tag{3.12}$$
$$\xleftarrow{[p,-]} \qquad\qquad \xleftarrow{[p,-]} \qquad\qquad \xleftarrow{[p,-]}$$

where $\mathrm{ad}_\xi \equiv [\xi, -]$. For completeness, in Appendix C we derive the explicit form of the basis elements. Our initial Killing tensor is thus some linear combination of these plus

---

[11]We are labeling irreducible representations of $sl(2, \mathbb{R})$ according to the familiar nomeclature of the the complexified $su(2)$ algebra. The spin $J - 1$ representation is the $(2J - 1)$-dimensional one.

something divisible by the Casimir:

$$\zeta = \sum_{n=0}^{N} a_n \, (\mathrm{ad}_k)^n p^{J-1} + \text{traces} \qquad \text{with} \quad a_N \neq 0 \; , \quad N \leq 2(J-1) \; , \tag{3.13}$$

where 'traces' refers to quantities of the form $C_2 \hat{\zeta}$ (divisible by $C_2$). Since $C_2$ commutes with everything, then acting enough times with $\mathcal{L}_p$ will give

$$(\mathcal{L}_p)^N \, \zeta \quad \propto \quad p^{J-1} + \text{traces} \; . \tag{3.14}$$

We can dispense with the traces by acting with $(\mathcal{L}_p)^{2(J-1)}(\mathcal{L}_k)^{2(J-1)}$:

$$(\mathcal{L}_p)^{2(J-1)} \, (\mathcal{L}_k)^{2(J-1)} \, (\mathcal{L}_p)^N \, \zeta \quad \propto \quad p^{J-1} \; , \tag{3.15}$$

as this operator maps $p^{J-1}$ back to a non-zero multiple of itself, and maps all traces to zero since they are of strictly lower spin (i.e. they belong to a multiplet of the form (3.12) for a lower value of $J$). Now acting with $\mathcal{L}_k$ furnishes each basis element (3.12),

$$(\mathcal{L}_k)^n \, (\mathcal{L}_p)^{2(J-1)} \, (\mathcal{L}_k)^{2(J-1)} \, (\mathcal{L}_p)^N \, \zeta \quad \propto \quad (\mathrm{ad}_k)^n p^{J-1} \; . \tag{3.16}$$

One has thus produced the $2J-1$ independent Killing tensors in (3.12) by acting with the isometries.

Globally speaking, these Killing tensors are all independent. However, at any given point in AdS$_2$, there must be position-dependent relations among them, since the space of 2d rank-$(J-1)$ symmetric tensors is only $J$-dimensional. Since any point of AdS is equivalent (it being maximally symmetric), we can count these relations by focusing on the point $x = 0$, $y = 1$ in the Poincaré coordinates, where the relation $k = -p$ appears among the Killing vectors (2.4). One finds that this leads to $J-1$ relations between the Killing vectors (3.12) (identifying them under a reflection across the middle of (3.12)).[12] As we will check below, there are no further relations, so the Killing tensors evaluated at a point form a basis for all symmetric tensors. Hence we have proven that partial conservation equations like (3.1) are satisfied with $\zeta$ replaced with *any symmetric tensor*. It therefore follows that the current is fully conserved, i.e. equation (3.5), concluding the proof.

**Characterisation of Killing tensors at a point**   First, we notice *en passant* that, in the simplest case of $J = 2$, the Killing vectors evaluated at a point certainly generate the tangent space of the manifold. This is a simple consequence of (Euclidean) AdS being a symmetric space:

$$\mathrm{EAdS}_2 \simeq \frac{\mathrm{SL}(2, \mathbb{R})}{\mathrm{O}(2)} \; , \tag{3.17}$$

and is made explicit by the following choice of basis for the Killing vectors

$$\xi^0 = \frac{1}{2}(p + k) \; , \quad \xi^1 = \frac{1}{2}(p - k) \; , \quad \xi^2 = d \; . \tag{3.18}$$

---

[12]This can be seen by noting that the same elements are obtained (up to proportionality) starting from either the left or the right: $(\mathrm{ad}_k)^n p^{J-1} \propto (\mathrm{ad}_p)^{2(J-1)-n} k^{J-1}$. Now identifying $k = -p$, we obtain an identification of each element $(\mathrm{ad}_k)^n p^{J-1}$ with (a multiple of) its image under $n \to 2(J-1) - n$. There is precisely one fixed point under this mapping, so the $2J-1$ elements acquire $J-1$ relations.

In embedding space, the three $\xi^A$ are the generators of rotations and boosts (see Appendix C), with $\xi^0$ generating rotations around the timelike axis. In physical space, it precisely generates the $O(2)$ little group of the point $(0, 1)$. In Appendix C, we check that all symmetric tensors at a point are spanned by the Killing tensors, thanks to the explicit expressions for the basis elements (3.12).

A more elegant way to derive the same fact is by using the isomorphism shown in Appendix C between Killing tensors belonging to the representation (3.12) and harmonic polynomials that are homogeneous in the three variables $\xi^1$, $\xi^2$, $\xi^3$, where $\xi^3 = i\xi^0$. The polynomials are harmonic because the spin-$(J-1)$ irreducible representation is obtained by choosing $\alpha^{A_1\cdots A_{J-1}}$ to be traceless in equation (2.9). Then, the fact that Killing tensors at a point generate all symmetric tensors is equivalent to the following: any homogeneous polynomial in two variables can be completed into a homogeneous harmonic polynomial in three variables of the same degree. Indeed, consider a polynomial $P_0(\xi^1, \xi^2)$, homogeneous of degree $\ell$ in the two variables. We define the following polynomial $H(\xi^1, \xi^2, \xi^3)$:

$$H(\xi^1, \xi^2, \xi^3) = P_0(\xi^1, \xi^2) + \sum_{n=1}^{\ell} \left(\xi^3\right)^n P_n(\xi^1, \xi^2) . \tag{3.19}$$

$H$ is homogeneous if $P_n$ has degree $\ell - n$. Furthermore, one can check that $(\partial_1^2 + \partial_2^2 + \partial_3^2)H = 0$ if

$$P_{n+2}(\xi^1, \xi^2) = -\frac{1}{(n+1)(n+2)}\left(\partial_1^2 + \partial_2^2\right)P_n(\xi^1, \xi^2) = 0 , \qquad n = 0, \ldots, \ell - 2 , \tag{3.20}$$

which uniquely fixes all the auxiliary polynomials $P_n$ for $n > 0$.

## 3.2   Charge conservation from Heisenberg's equation

As we have established, to build conserved charges by contracting an AdS$_2$ higher-spin current $T_{\nu_1\cdots\nu_J}$ with a Killing tensors, the minimal assumption is full conservation, $\nabla^\mu T_{\mu\nu_1\cdots\nu_{J-1}} = 0$. This is equivalent to having partial conservation $\nabla^\mu\left(\zeta^{\nu_1\cdots\nu_{J-1}} T_{\mu\nu_1\cdots\nu_{J-1}}\right) = 0$ for *all* Killing tensors $\zeta$. We will use the latter formulation in what follows.

For any Killing tensor $\zeta$, a charge is defined on co-dimension 1 surfaces $\Sigma$ (either closed or ending on the boundary of AdS) by

$$Q_\zeta = \int_\Sigma \sqrt{\gamma}\, dS^\mu\, \zeta^{\nu_1\cdots\nu_{J-1}} T_{\mu\nu_1\cdots\nu_{J-1}} , \tag{3.21}$$

generalizing the spin-3 and 4 currents constructed in Section 2.

To state the conservation of these charges, let us consider an infinitesimal deformation of the surface $\Sigma$ by acting on it with an isometry $Q_\xi$ associated to a Killing vector $\xi$. One can always choose coordinates where the isometry is given by a 'translation' along some coordinate $u$,

$$\xi = \partial_u , \qquad \partial_u g_{\mu\nu} = 0 . \tag{3.22}$$

The statement of conservation then translates to a Heisenberg equation,

$$0 = \frac{dQ_\zeta}{du} = Q_{\frac{\partial\zeta}{\partial u}} + [Q_\xi, Q_\zeta], \tag{3.23}$$

where $Q_\xi \in sl(2, \mathbb{R})$ is the isometry generator associated to $\xi$. The second term on the RHS is the standard symmetry action on an operator. The first term comes from the explicit $u$-dependence of the Killing tensor $\zeta$, which is given by the the Lie derivative, $Q_{\frac{\partial \zeta}{\partial u}} = Q_{\mathcal{L}_\xi \zeta}$. Hence (3.23) gives

$$[Q_\xi, Q_\zeta] = -Q_{\mathcal{L}_\xi \zeta} . \tag{3.24}$$

On the RHS is a charge built from a transformed Killing tensor. Equation (3.24) gives an alternative proof of the fact that a charge labeled by a Killing tensor implies conservation of the charges in the $sl(2, \mathbb{R})$ orbit of that tensor. The commutation relations between these higher-spin charges and $sl(2, \mathbb{R})$ generators are thus fixed by the fact that they are conserved.

For example, let us focus on the spin-4 charge built from the Killing tensor $p^3$ that we considered above for free fields. Since $\mathcal{L}_p(p^3) = 0$ then (3.24) gives

$$\left[ P, Q_{p^3} \right] = 0 . \tag{3.25}$$

Similarly, since $\mathcal{L}_d(p^3) = -3p^3$ then we obtain

$$\left[ D, Q_{p^3} \right] = 3 Q_{p^3} . \tag{3.26}$$

More explicitly, one may derive the RHS of (3.26) by working in the AdS global coordinates,

$$ds^2 = \cosh^2 \rho \, d\tau^2 + d\rho^2 , \qquad -\infty < \tau < \infty , \quad 0 < \rho < \infty , \tag{3.27}$$

where dilatations are generated by shifts of $\tau$. In these coordinates the Killing vector $p$ takes the form

$$p = \partial_x = e^{-\tau} (\tanh \rho \, \partial_\tau + \partial_\rho) . \tag{3.28}$$

so the relevant Killing tensor $p^3 = \zeta^{\nu\rho\sigma} \partial_\nu \partial_\rho \partial_\sigma$ satisfies $\partial_\tau \zeta^{\nu\rho\sigma} = -3\zeta^{\nu\rho\sigma}$ as claimed.

Physically, equation (3.25) (or respectively (3.26)) may be understood as quantizing the theory in constant time (or radial) slices, with the corresponding Heisenberg equations of motion describing invariance under time evolution in $x$ (or $\tau$) generated by the Hamiltonian $P$ (or $D$).

## 4 Consequences of higher-spin symmetries

We will now assume the existence of higher-spin conserved charges and consider the resulting constraints on the boundary theory. This is analogous to deriving factorised scattering from higher-spin charges in flat space [18–20]. In AdS, the outcome will be much more constraining, and we will prove a no-go theorem: higher spin charges force the theory to be free, at least in the perturbative vicinity of a single generalized free field with generic mass. At the end of the section, we also present some constraints on the data of the bulk theory.

### 4.1 Algebra of higher-spin charges

Let us assume the existence of a charge $Q_{-3}^{(4)}$ satisfying the commutation relations

$$\left[P, Q_{-3}^{(4)}\right] = 0, \qquad \left[D, Q_{-3}^{(4)}\right] = 3 Q_{-3}^{(4)}, \tag{4.1}$$

meaning it commutes with translations and has weight 3. As we established above, such a charge (built with the Killing tensor $p^3$) follows from the existence of a local, spin-4 current in the bulk and in that context we referred to it as $Q_{p^3}$. Here we will simply take these commutation relations as an axiom, which is a perfectly meaningful assumption purely from the boundary's perspective. To emphasize this, we have introduced the notation $Q_{-w}^{(J)}$ for a charge of weight $w$ belonging to a $(2J-1)$-dimensional representation of $sl(2, \mathbb{R})$—without making reference to a bulk Killing tensor.[13] We also assume that $Q_{-3}^{(4)}$ annihilates the vacuum.

In this subsection, we are going to build an algebra of higher-spin charges by acting on $Q_{-3}^{(4)}$ with conformal generators. Using the generator $K$, we define the weight-2 object

$$Q_{-2}^{(4)} \equiv \left[Q_{-3}^{(4)}, K\right]. \tag{4.2}$$

The Jacobi identity between $Q_{-3}^{(4)}$, $P$ and $K$ then imposes that

$$\left[P, Q_{-2}^{(4)}\right] = 6 Q_{-3}^{(4)}. \tag{4.3}$$

In particular, this implies that $Q_{-2}^{(4)}$ is non-vanishing and also annihilates the vacuum. Similarly we can generate charges with lower and lower weight,

$$Q_{n+1}^{(4)} \equiv \left[Q_n^{(4)}, K\right], \tag{4.4}$$

and Jacobi identities imply that all such charges are non-vanishing until one reaches[14]

$$\left[Q_{+3}^{(4)}, K\right] = 0. \tag{4.5}$$

In particular there is one charge for each integer weight in the interval $[-3, 3]$, meaning the charges furnish the 7-dimensional representation of $sl(2, \mathbb{R})$. For completeness we give the remaining commutators:

$$\left[P, Q_{-1}^{(4)}\right] = 10 Q_{-2}^{(4)}, \left[P, Q_0^{(4)}\right] = 12 Q_{-1}^{(4)}, \left[P, Q_{+1}^{(4)}\right] = 12 Q_0^{(4)}, \tag{4.6}$$

$$\left[P, Q_{+2}^{(4)}\right] = 10 Q_{+1}^{(4)}, \left[P, Q_{+3}^{(4)}\right] = 6 Q_{+2}^{(4)}, \tag{4.7}$$

and the commutators with $D$ simply read off the weights, $\left[D, Q_w^{(4)}\right] = -w Q_w^{(4)}$.

---

[13]The subscript $-3$ (rather than 3) is chosen to agree with the standard notation for the charges of the Virasoro algebra in Section 5 below.

[14]This follows since $\left[Q_{+3}^{(4)}, K\right]$ has negative weight, so annihilates primary states. Since it commutes with $P$ (by Jacobi identities), then it also annihilates all descendents.

These are exactly the charges $Q_\zeta^{(4)}$ obtained in (3.21) by contracting a spin-4 bulk current with all possible Killing tensors. As discussed above, there is precisely one charge per weight, modulo traces, which form representations of strictly lower spin.

Aside from the commutation relations of the charges $Q_w^{(4)}$ with conformal generators, we can determine their commutators with themselves. This can produce charges of weight outside the original range $[-3, 3]$, and hence must open up new sectors $Q_w^{(J)}$ with $J > 4$. For example, by defining

$$Q_{-5}^{(6)} \equiv \left[ Q_{-2}^{(4)}, Q_{-3}^{(4)} \right], \tag{4.8}$$

we construct a weight 5 charge which further commutes with translations $\left[ Q_{-5}^{(6)}, P \right] = 0$, as follows from a Jacobi identity. We can now repeat the procedure used for $Q_w^{(4)}$ and generate a tower of charges by commuting with $K$, for example

$$Q_{-4}^{(6)} \equiv \left[ Q_{-5}^{(6)}, K \right], \tag{4.9}$$

The Jacobi identity fixes further commutation relations, for instance

$$\left[ Q_{-1}^{(4)}, Q_{-3}^{(4)} \right] = Q_{-4}^{(6)}, \quad \left[ P, Q_{-4}^{(6)} \right] = 10 \, Q_{-5}^{(6)}, \quad \left[ P, \left[ Q_{-1}^{(4)}, Q_{-2}^{(4)} \right] \right] = 6 \, Q_{-4}^{(6)}. \tag{4.10}$$

This new set of charges $Q_w^{(6)}$ suggests the existence of a local spin-6 current in the bulk. In fact, it turns out that continuing this procedure generates $2J - 1$ charges for each even spin $J$. This is the even-spin subalgebra of the one-parameter family of higher-spin algebras $hs[\lambda]$ [34, 69–72], which can be seen as the algebra of linear symmetries of a free complex boson with mass $m(\lambda) = (\lambda^2 - 1)/4$ [63]. While it is not guaranteed in general,[15] at least for free fields in $\text{AdS}_2$ with generic mass, all of the charges even spins $J \geq 4$ are non-zero. This will be important below when we consider deforming free fields by interactions.

The fact that the charges have integer weight will strongly constrain their action on local operators. We already saw this in the example of the free boson, where we used naive dimensional analysis to determine the possible operators produced by acting with a higher-spin charge: descendants and integer-spaced primaries. In the next section we will extend this argument to a general theory with higher-spin conserved charges, using non-perturbative conformal invariance. We will systematically constrain the action on local operators and derive Ward identities for correlation functions.

## 4.2 Ward Identities

For the highest-weight spin-4 charge $Q_{-3}^{(4)}$, the commutation relations with momentum and dilation strongly restrict its action on local operators: it must be translation-invariant and increase the weight by 3. Let's consider the action on a non-degenerate primary $\mathcal{O}_0$ of dimension $\Delta_0$ that we choose to be a HS primary, defined as an operator that is annihilated by all charges that lower the weight. Simple examples are $\phi^n$ operators in GFB or, as we will see later, Virasoro primaries. Then the action is constrained to be of the form

$$\left[ Q_{-3}^{(4)}, \mathcal{O}_0(x) \right] = \alpha \, \partial^3 \mathcal{O}_0(x) + \omega \, \partial^2 \mathcal{O}_1(x) + \beta \, \partial \mathcal{O}_2(x) + \gamma \, \mathcal{O}_3(x), \tag{4.11}$$

---

[15]This is related to the fact that the higher-spin algebra $hs[\lambda]$ at certain non-generic values of $\lambda$ admits ideals that can be quotiented out, sometimes leading to finite dimensional algebras [73].

where $\alpha, \beta, \gamma, \omega$ are constants, and $\mathcal{O}_1, \mathcal{O}_2, \mathcal{O}_3$ are conformal primary operators of dimension $\Delta_1 = \Delta_0 + 1$, $\Delta_2 = \Delta_0 + 2$ and $\Delta_3 = \Delta_0 + 3$, respectively. Henceforth, we will set $\omega = 0$, since this will be the case in all the examples we will consider. However, it is straightforward to generalize the analysis below to the case $\omega \neq 0$. Note that, because of the number of derivatives, $\mathcal{O}_2$ must have the same parity as $\mathcal{O}_0$, while $\mathcal{O}_3$ must have the opposite parity. No terms with more than three derivatives appear in this case due to the HS primary assumption. This can be seen by acting four times with the special conformal generator $K$ on (4.11) and noticing that the left-hand side of the equation vanishes due to the Jacobi identity and the HS primary assumption.

We immediately observe that the higher-spin charge is very constraining, since its action involves primary operators with integer-spaced scaling dimensions. On one hand, we expect that interacting theories will almost never have such operators. But on the other hand, if the operators $\mathcal{O}_2$ and $\mathcal{O}_3$ are absent ($\beta = \gamma = 0$), then (4.11) leads to Ward identities that are very simple differential equations. As we will prove in the following subsection, the only solutions are linear combinations of the generalized free boson (GFB) $\mathcal{O}_0 = \phi$ and the generalized free fermion (GFF) $\mathcal{O}_0 = \psi$.

Before focusing on perturbative deformations of free fields, let us consider the Ward identities implied by the higher-spin charge $Q_{-3}^{(4)}$ in the most general case where it acts as (4.11) with $\beta, \gamma \neq 0$. As we saw, this forces the existence of integer-spaced primaries $\mathcal{O}_2$ and $\mathcal{O}_3$. We will first consider a 3-point function $\langle \mathcal{O}_0 \mathcal{O}_0 \mathcal{O}_0 \rangle$. We assumed $Q_{-3}^{(4)}$ to annihilate the in-vacuum $|0\rangle$, and it annihilates the out-vacuum $\langle 0|$ because it has positive weight. Hence, inserting the charge acting on the in/out vacuua, and commuting it past the local operators, we obtain a Ward identity

$$\left\langle \left[ Q_{-3}^{(4)}, \mathcal{O}_0(x_1) \right] \mathcal{O}_0(x_2) \mathcal{O}_0(x_3) \right\rangle + (x_1 \leftrightarrow x_2) + (x_1 \leftrightarrow x_3) = 0 \,. \tag{4.12}$$

With the charge acting as (4.11), this Ward identity yields a differential equation satisfied by the correlator,

$$\left( \alpha \, \partial_{x_1}^3 \langle \mathcal{O}_0(x_1) \mathcal{O}_0(x_2) \mathcal{O}_0(x_3) \rangle + \beta \, \partial_{x_1} \langle \mathcal{O}_2(x_1) \mathcal{O}_0(x_2) \mathcal{O}_0(x_3) \rangle \right. \tag{4.13}$$
$$\left. + \gamma \, \langle \mathcal{O}_3(x_1) \mathcal{O}_0(x_2) \mathcal{O}_0(x_3) \rangle \right) + (x_1 \leftrightarrow x_2) + (x_1 \leftrightarrow x_3) = 0 \,.$$

By parity alone, we know that $\langle \mathcal{O}_3 \mathcal{O}_0 \mathcal{O}_0 \rangle = 0$. Hence the differential equations leads to the following simple identity between OPE coefficients and scaling dimensions, involving only $\mathcal{O}_0$ and $\mathcal{O}_2$:

$$3\alpha \, C_{000} \, \Delta_0^2 (\Delta_0 + 1) - \beta \, (\Delta_0 + 2) \, C_{002} = 0 \,. \tag{4.14}$$

We clearly see that conserved higher-spin charges are very constraining, as such a relation between OPE coefficients has to hold for any higher-spin primary operator $\mathcal{O}_0$. It is straightforward to check (4.14) in the example of $\mathcal{O}_0 = \phi^2$ and $\mathcal{O}_2 = \mathcal{O}_{2,2}$ in GFB theory, by computing the four-point functions $\langle \mathcal{O}_0 \mathcal{O}_0 \mathcal{O}_0 \mathcal{O}_0 \rangle$ and $\langle \mathcal{O}_2 \mathcal{O}_0 \mathcal{O}_0 \mathcal{O}_0 \rangle$ and expanding in conformal blocks. It is also straightforward to check in the case where $\mathcal{O}_0$ is a Virasoro primary and $\mathcal{O}_2$ is a level 2 quasi-primary. We will return to this in Section 5.

While it is tempting to try to extend this computation by considering several different three-point functions, it becomes hard to systematically select them in a way that the

number of equation matches the number of unknowns. Instead, let us turn to the Ward identity satisfied by 4-point correlation functions $\langle \mathcal{O}_0(x_1)\mathcal{O}_0(x_2)\mathcal{O}_0(x_3)\mathcal{O}_0(x_4)\rangle$:

$$
\begin{aligned}
&\alpha \left( \partial_{x_1}^3 \langle \mathcal{O}_0\mathcal{O}_0\mathcal{O}_0\mathcal{O}_0\rangle + \partial_{x_2}^3 \langle \mathcal{O}_0\mathcal{O}_0\mathcal{O}_0\mathcal{O}_0\rangle + \partial_{x_3}^3 \langle \mathcal{O}_0\mathcal{O}_0\mathcal{O}_0\mathcal{O}_0\rangle + \partial_{x_4}^3 \langle \mathcal{O}_0\mathcal{O}_0\mathcal{O}_0\mathcal{O}_0\rangle \right) \\
&\quad + \beta \left( \partial_{x_1} \langle \mathcal{O}_2\mathcal{O}_0\mathcal{O}_0\mathcal{O}_0\rangle + \partial_{x_2} \langle \mathcal{O}_0\mathcal{O}_2\mathcal{O}_0\mathcal{O}_0\rangle + \partial_{x_3} \langle \mathcal{O}_0\mathcal{O}_0\mathcal{O}_2\mathcal{O}_0\rangle + \partial_{x_4} \langle \mathcal{O}_0\mathcal{O}_0\mathcal{O}_0\mathcal{O}_2\rangle \right) \\
&\quad + \gamma \left( \langle \mathcal{O}_3\mathcal{O}_0\mathcal{O}_0\mathcal{O}_0\rangle + \langle \mathcal{O}_0\mathcal{O}_3\mathcal{O}_0\mathcal{O}_0\rangle + \langle \mathcal{O}_0\mathcal{O}_0\mathcal{O}_3\mathcal{O}_0\rangle + \langle \mathcal{O}_0\mathcal{O}_0\mathcal{O}_0\mathcal{O}_3\rangle \right) \\
&= 0 \,.
\end{aligned}
\tag{4.15}
$$

Using the prefactor conventions

$$
\begin{aligned}
&\langle \mathcal{O}_0\mathcal{O}_0\mathcal{O}_0\mathcal{O}_0\rangle = \frac{f(z)}{x_{12}^{2\Delta_1} x_{34}^{2\Delta_1}} \,, && z \equiv \frac{x_{12}x_{34}}{x_{13}x_{24}} \,, \\
&\langle \mathcal{O}_2\mathcal{O}_0\mathcal{O}_0\mathcal{O}_0\rangle = \frac{g(z)}{x_{12}^{2\Delta_0+2} x_{34}^{2\Delta_0}} \left(\frac{x_{24}}{x_{14}}\right)^2 \,, && \langle \mathcal{O}_3\mathcal{O}_0\mathcal{O}_0\mathcal{O}_0\rangle = \frac{h(z)}{x_{12}^{2\Delta_0+3} x_{34}^{2\Delta_0}} \left(\frac{x_{24}}{x_{14}}\right)^3 \,,
\end{aligned}
\tag{4.16}
$$

one sees that the differential equation naively depends on $g$ and $h$ at different points. However, one can use cyclic permutations to reduce all the correlators to ones where the cross-ratio is either $z$ or $1 - z$. Furthermore, using the non-cyclic permutation $x_2 \leftrightarrow x_4$, combined with Bose symmetry, and a reflection, reduces all correlators to cross-ratio $z$. Note that the last crossing relation involves a reflection, and hence the function $h(1-z)$ acquires an extra minus sign under said crossing, since the operator $\mathcal{O}_3$ has the opposite parity of $\mathcal{O}_0$ and $\mathcal{O}_2$. While there are *a priori* different functions of positions that cannot be reabsorbed into functions of the cross-ratio, the resulting equations turn out to be proportional, and one only obtains one independent ODE for the three functions, which reads:

$$
3\alpha \, z(1-z)^2 \, \mathcal{D}^{(3)} f(z) + \beta \, \mathcal{D}^{(1)} g(z) + 3\gamma \, h(z) = 0 \,,
\tag{4.17}
$$

where we define the differential operators

$$
\mathcal{D}^{(1)} g(z) \equiv 3(z-1)zg'(z) + g(z)(4\Delta_\phi - 2(\Delta_\phi + 2)z + 2) \,,
\tag{4.18}
$$

$$
\mathcal{D}^{(3)} f(z) \equiv z \left( (z-1)zf'''(z) + (4\Delta_\phi - 2\Delta_\phi z + 4z - 2)f''(z) \right) - 2(2\Delta_\phi - 1)(\Delta_\phi + z)f'(z) \,.
$$

Once again, the correlators $\langle \phi^2\phi^2\phi^2\phi^2\rangle$ and $\langle \mathcal{O}_{2,2}\,\phi^2\phi^2\phi^2\rangle$ in GFB theory provide a simple check of this ODE in the case $\gamma = 0$, with the ratio $\beta/\alpha = -(6\Delta_\phi\sqrt{2\Delta_\phi + 1}\sqrt{4\Delta_\phi + 1})/(\Delta_\phi + 1)$ determined by equation (2.48). We study the relevant four-point functions in Appendix A.2.

This situation is rather unconstrained, since the multiple functions $f(z), g(z), h(z)$ are constrained by just a single ODE (4.17). For the general case $\alpha, \beta, \gamma \neq 0$, GFB itself provides a simple explanation for this fact: $\mathcal{O}_0$ could be any of the operators $\phi^n$ with $n \geq 3$, which provide an infinite number of solutions to the ODE. Here, the fact that $\gamma \neq 0$ is important, since it excludes the $n = 1, 2$ cases of $\phi$ and $\phi^2$, where $\gamma = 0$ and there is no primary of dimension $n\Delta_\phi + 3$. We provide an explicit check of the action of the charge $Q_{-3}^{(4)}$ on the operator $\phi^3$ in Appendix A.3.

**Special case $\gamma = 0$:**  One might expect to reach stronger conclusions by focusing on $\gamma = 0$, but in fact the correlators are still not uniquely fixed.[16] One example in this class is the operator $\mathcal{O}_0 = \phi^2$ in the GFB theory. As we will see in Section 5, there are infinitely many other examples in the context of CFTs in AdS$_2$.

Universal features of this class of solutions can, however, be described by extracting from the ODE (4.17) an infinite family of relations between OPE coefficients, generalising the one (4.14) obtained from the 3-point Ward identity. Let us expand the correlators in conformal blocks,

$$f(z) = \sum_\Delta C_{00\Delta}^2 \, z^\Delta \, {}_2F_1(\Delta, \Delta, 2\Delta, z) \,, \tag{4.19}$$

$$g(z) = \sum_\Delta C_{20\Delta} C_{00\Delta} \, z^\Delta \, {}_2F_1(\Delta - 2, \Delta, 2\Delta, z) \,, \tag{4.20}$$

and substitute them into the ODE (4.17). Then, for a HS primary of dimension $\Delta$ (not necessarily in the same multiplet as $\mathcal{O}_0$), we find that

$$C_{20\Delta} = -\frac{3\Delta(\Delta - 2\Delta_0 - 1)(\Delta - 2\Delta_0)}{(\beta/\alpha)(3\Delta - 2 - 4\Delta_0)} C_{00\Delta} \,. \tag{4.21}$$

It is straightforward to extend this to HS descendants by expanding to higher order. For example, in the HS family of the external operators we find

$$C_{202} = \frac{\left(3C_{002}^2 (\Delta_0 - 2)(\Delta_0 - 1)(\Delta_0 + 2) - 3C_{000}^2 \Delta_0^2 (\Delta_0 + 1)\right)}{(\beta/\alpha)C_{002}(\Delta_0 - 4)} \,. \tag{4.22}$$

### 4.3 No-go theorem for higher-spin preserving deformations of a free field

Now let us turn to one of our main results. We shall denote a generalized free field as $\chi = \phi$ or $\psi$, which may be either bosonic or fermionic, and its scaling dimension by $\Delta_\chi$.

**No-go theorem on the boundary:**  *Starting from a generalized free field, consider a continuous deformation that preserves the spin-4 charge $Q_{-3}^{(4)}$ and conformal invariance. Then the deformed theory is still a generalized free field.*

By 'continuous' deformation, we mean that the OPE coefficients and scaling dimensions depend continuously on the deformation parameter, and no new states are introduced in the Hilbert space. The charge $Q_{-3}^{(4)}$ is assumed to satisfy the commutation relations (4.1). This has the following consequence for QFTs in AdS$_2$:

**No-go theorem in AdS:**  *Starting from a massive free scalar or Majorana fermion in AdS$_2$, consider continuously deforming by an interaction that respects AdS isometries. Then it is impossible to preserve any higher-spin conserved currents.*

---

[16]Similar charge conservation identities in local CFTs play a crucial role in deriving the no-go theorems in [30]. See also [74] for a recent discussion. In that case, the local operators are spinning and are in fact conserved currents themselves. This leads to more independent differential equations, which have a finite dimensional space of solutions. Having sacrificed locality on the boundary, we are in a far less constrained situation.

Indeed, the boundary no-go theorem implies the AdS one because, if a spin-4 current is even 'partially' conserved along a certain Killing tensor, then by Section 3.1, it is conserved along all Killing tensors. Hence all of the charges $Q_w^{(4)}$ ($-3 \leq w \leq 3$) are obtained by integrating it along different Killing tensors, and they satisfy the algebra discussed in Section 4.1. In particular, it follows that the charge $Q_{-3}^{(4)}$ exists and is non-zero, so the no-go theorem applies. If instead the deformed theory has spin $J > 4$ conserved currents, then one can simply take traces until obtaining a spin-4 one. One cannot get stuck with a traceless conserved current in the vicinity of massive free fields, since one can check that their currents are all traceful and lead to non-trivial charges.

To prove the no-go theorem, we will first show that action of particular HS charges on the generalized free field is not modified by the deformation. Then, we will prove that the corresponding Ward identities, together with the continuity assumption, constrain the theory to remain either GFB or GFF.

Recall the higher-spin charge $Q_{-3}^{(4)}$ acts on $\chi$ with $\beta = \gamma = 0$ in (4.11), i.e.

$$\left[ Q_{-3}^{(4)}, \chi(x) \right] = \alpha\, \partial^3 \chi(x) \,. \tag{4.23}$$

Now let us consider perturbing the generalized free field theory by a general interaction, parametrised by a coupling $\lambda$. Boundary conformal invariance guarantees that physics can only depend on the dimensionless coupling $\tilde\lambda = \lambda^{1/\Delta_\lambda} L$, where $\Delta_\lambda$ is the UV mass dimension of the bulk coupling and we briefly reinstated the $L$ dependence to clarify dimensional analysis. This absence of dimensionful couplings is precisely the reason why only integer spaced operators are allowed to appear when acting on an arbitrary HS primary as in (4.11). This means that the action can only get modified to

$$\left[ Q_{-3}^{(4)}, \mathcal{O}_0 \right] = \partial^3 \mathcal{O}_0 + \beta(\tilde\lambda)\, \partial \mathcal{O}_2 + \gamma(\tilde\lambda)\, \mathcal{O}_3 \,. \tag{4.24}$$

where $\mathcal{O}_0$ reduces to $\chi$ in the $\tilde\lambda \to 0$ limit and we chose to normalize the charge such that the first coefficient is fixed to 1.

Crucially, operators $\mathcal{O}_2$ and $\mathcal{O}_3$ are required to exist in the theory for small values of $\tilde\lambda$. Even if their couplings $\beta(\tilde\lambda), \gamma(\tilde\lambda)$ may switch off in the $\tilde\lambda \to 0$ limit, the operators must continue to exist in the $\tilde\lambda = 0$ theory, by the assumption of continuity. But for generic values of $\Delta_\chi$ this is impossible: there are simply no candidate primaries of dimension $\Delta_\chi + 2$ or $\Delta_\chi + 3$. Hence we conclude that

$$\beta(\tilde\lambda) \equiv 0 \,, \qquad \gamma(\tilde\lambda) \equiv 0 \,, \tag{4.25}$$

and the charge acts the same as (4.23) for values of $\tilde\lambda$ in a neighbourhood of zero. We will discuss special values of $\Delta_\chi$ at the end of this section.

In fact, the preservation of the charge $Q_{-3}^{(4)}$ implies that all of the higher-spin charges $Q_w^{(J)}$ with $J$ even are preserved. This follows by taking commutators as in Section 4.1, and the resulting charges cannot vanish in the vicinity of the generalised free field, since they do not vanish in that limit. To prove the theorem, we will also need the following generalization of (4.23) to these even spin-$J$ charges: for the generalized free field $\chi$,

$$\left[ Q_{-J+1}^{(J)}, \chi(x) \right] = \alpha_J\, \partial^{J-1} \chi(x) \,, \qquad J \text{ even} \,. \tag{4.26}$$

As above, for generic values of $\Delta_\chi$, there are no other operators that could appear on the RHS when $\chi$ is replaced by $\mathcal{O}_0$. Therefore, also in the deformed theory,

$$\left[Q^{(J)}_{-J+1}, \mathcal{O}_0(x)\right] = \alpha_J \, \partial^{J-1} \mathcal{O}_0(x) \, , \qquad J \text{ even} \, . \tag{4.27}$$

We will now use this to prove that all $n$-point functions of $\mathcal{O}_0$ remain those of a generalized free field under the deformation. Let us start by considering the 4-point function $\langle \mathcal{O}_0 \mathcal{O}_0 \mathcal{O}_0 \mathcal{O}_0 \rangle$.

**4-point function**   As a consequence of (4.25) and (4.24), the 4-point Ward identity (4.15) takes the special form

$$\left(\partial^3_{x_1} + \partial^3_{x_2} + \partial^3_{x_3} + \partial^3_{x_4}\right) \langle \mathcal{O}_0(x_1)\mathcal{O}_0(x_2)\mathcal{O}_0(x_3)\mathcal{O}_0(x_4)\rangle = 0 \, . \tag{4.28}$$

The essential reason why there is only a discrete set of solutions in this case is that the differential equation (4.28) involves a single correlator. Hence the resulting ODE (4.17) involves a single function of the conformal cross-ratio:

$$\mathcal{D}^{(3)} f(z) = 0 \, , \tag{4.29}$$

where we recall the definition $\langle \mathcal{O}_0(x_1)\mathcal{O}_0(x_2)\mathcal{O}_0(x_3)\mathcal{O}_0(x_4)\rangle = x_{12}^{-2\Delta_0} x_{34}^{-2\Delta_0} f(z)$, and the 3rd order differential operator $\mathcal{D}^{(3)}$ was defined in (4.18).

Solving this equation generates three integration constants, two of which are fixed by imposing $t$-channel crossing and unit normalization of the identity contribution. One then finds the one-parameter family of solutions

$$f(z) = t \, f_{\text{GFB}}(z) + (1 - t) \, f_{\text{GFF}}(z) \, . \tag{4.30}$$

For generic $t \in [0, 1]$, the correlator (4.30) exchanges both bosonic and fermionic operators. Since the deformation is assumed not to introduce new states, we conclude that either $t = 0$ or $t = 1$.

**Higher-point functions**   Now let us carry out the same argument for all $n$-point functions of $\mathcal{O}_0$. To do this, we will need to use spin $J \geq 6$ charges $Q^{(J)}_{-J+1}$ as well as $Q^{(4)}_{-3}$. As discussed in Section 4.1, charges of all even spins are produced from $Q^{(4)}_{-3}$ by taking commutators, and they are non-vanishing for generalized free fields with generic values of $\Delta_\chi$, and hence in a neighborhood of those theories.

As a consequence of (4.27), the Ward identities for $n$-point functions are

$$\left(\partial^{J-1}_{x_1} + \cdots + \partial^{J-1}_{x_n}\right) \langle \mathcal{O}_0(x_1) \ldots \mathcal{O}_0(x_n)\rangle = 0 \, . \tag{4.31}$$

In fact, the only conformal solutions of this infinite set of Ward identities are linear combinations of products of 2-point functions. To show this, it is convenient to work in momentum space, where the Ward identities read

$$\left(\sum_{i=1}^{n} p_i^{J-1}\right) \langle \mathcal{O}_0(p_1) \ldots \mathcal{O}_0(p_n)\rangle = 0 \, , \tag{4.32}$$

for all even $J$. Closely mirroring the standard argument, using higher-spin charges, that momenta are individually conserved in integrable 2d scattering (reviewed in [61]), we conclude that the momentum-space correlator satisfies 'pairwise conservation' of momenta. This means that for odd $n$ the correlator thus vanishes, while for even $n$ it takes the form:

$$\langle \mathcal{O}_0(p_1) \dots \mathcal{O}_0(p_n) \rangle = \delta(p_1 + p_2) \cdots \delta(p_{n-1} + p_n) \, \tilde{f}^{(12)\cdots(n-1,n)}(p_i) + (\text{perms}) . \qquad (4.33)$$

Translating back to position space, this means that

$$\langle \mathcal{O}_0(x_1) \dots \mathcal{O}_0(x_n) \rangle = f^{(12)\cdots(n-1,n)}(x_{12}, \dots x_{n-1,n}) + (\text{perms}) . \qquad (4.34)$$

Conformal invariance then forces these functions to be proportional to products of 2-point functions:

$$\langle \mathcal{O}_0(x_1) \dots \mathcal{O}_0(x_n) \rangle = \frac{c^{(12)\cdots(n-1,n)}}{x_{12}^{2\Delta_0} \cdots x_{n-1,n}^{2\Delta_0}} + (\text{perms}) . \qquad (4.35)$$

The only unfixed parameters are the coefficients $c^{(12)\cdots(n-1,n)}$, which are constant but possibly depend on the insertions' order. In the 4-point function case above, the 3 permutations are the 3 solutions of the ODE.

We will now fix all of the coefficients recursively over even $n$: let us assume that the $(n-2)$-point correlators of $\mathcal{O}_0$ are fixed to be purely those of GFB or GFF. Superficially there are $n$ possible OPE contractions between neighbouring points on the circle. But in fact continuity of the deformation implies that the statistics of the field $\mathcal{O}_0$ are preserved (either bosonic of fermionic). Thus, after using Bose or Fermi symmetry to permute the external operators, we can effectively consider OPEs between all $\binom{n}{2}$ pairs of operators. The identity block contribution of that OPE needs to reproduce the known $(n-2)$-point function. One can check that imposing this in all of the OPE channels unambiguously fixes all of the $n$-point function's coefficients.

Thus we conclude that all $n$-point functions of $\mathcal{O}_0$ must match the original GFB or GFF theory for the coupling $\tilde{\lambda}$ in a neighbourhood of zero. Given our continuity assumption that no new states appear, this implies that the theory is still GFB or GFF.

**Special values of $\Delta_\chi$**   For a discrete set of special values of $\Delta_\chi$, some coincidences occur that allow certain operators in the generalized free field theory to play the role of $\mathcal{O}_{2,3}$. For example, for a massless boson with $\Delta_\phi = 1$, the operator $\phi^3$ is allowed to appear since

$$\left[ Q_{-3}^{(4)}, \phi \right] = \partial^3 \phi + \beta(\tilde{\lambda}) \, \partial(\phi^3) , \qquad (4.36)$$

is compatible with scaling. While we will not be able to conclusively rule out the existence of higher-spin preserving interacting deformations in such cases, we believe that they do not exist. Indeed, in the massless example above, expanding the equation (4.36) to leading non-trivial order gives

$$(\partial_1^3 + \partial_2^3 + \partial_3^3 + \partial_4^3) \langle \phi \phi \phi \phi \rangle^{(\tilde{\lambda})} \qquad (4.37)$$
$$= -\beta'(0) \left( \partial_1 \langle \phi^3 \phi \phi \phi \rangle^{(0)} + \partial_2 \langle \phi \phi^3 \phi \phi \rangle^{(0)} + \partial_3 \langle \phi \phi \phi^3 \phi \rangle^{(0)} + \partial_4 \langle \phi \phi \phi \phi^3 \rangle^{(0)} \right) .$$

The source on the RHS, however, trivialises upon setting $\Delta_\phi = 1$, leading to precisely the same Ward identity (4.28), whose unique solutions are generalized free fields. For other rational values of $\Delta_\phi$ below 1, more complicated $\phi^n$ operators can play the role of $\mathcal{O}_2$, but in this case the four point function $\langle \mathcal{O}_2 \phi \phi \rangle$ vanishes identically. It would be interesting to understand if these arguments can be pushed to all orders in perturbation theory or if there are actual exceptions to the no-go theorem for these special values of $\Delta_\phi$.

Our argument above applied for deformations of both GFB and GFF. In the fermionic case, the leading local deformation in AdS is an irrelevant two–derivative four-fermion coupling that can be identified with the CFT operator $T\overline{T}$,[17] which famously preserves integrability in flat space [78, 79] including the associated higher-spin charges [80, 81]. In contrast, we have shown that the $T\overline{T}$ deformation of a single Majorana fermion in $AdS_2$ cannot preserve any higher-spin charges.

Finally, let us comment on deformations of multiple free fields. For generic, non-identical choices of their masses, the argument above clearly goes through. However, if the dimensions of the different fields are integer spaced, the actions of the charges have new candidate terms on the right-hand side and we need to work harder to exclude interacting deformations. In fact, in [63], it was suggested that an infinite tower of integer spaced scalar fields perturbed by a cubic interaction could preserve the higher-spin symmetry. It would be interesting to study these possibilities using the framework developed here, but we leave this for future work.

### 4.4 Example: searching for sine-Gordon in AdS

We have shown that it is impossible in $AdS_2$ to deform a free field by interactions that preserve higher-spin conserved currents and charges. Let us now see in an example what goes wrong if one tries to do this in practice. Famously, the sine-Gordon model in flat space is the unique $\mathbb{Z}_2$-preserving deformation of a free boson that preserves some higher-spin charges, leading to an integrable model with factorized scattering [61]. One can verify the preservation of higher-spin charges at leading order in the coupling, where the sine-Gordon theory with equation of motion $(\Box + m^2)\Phi = \lambda \Phi^3 + O(\lambda^2)$ is indistinguishable from $\Phi^4$ theory. This equation of motion can be used to explicitly construct the leading correction to the massive free boson's higher-spin currents in flat space.[18] Indeed expanding the current,

$$T_{\mu\nu\rho\sigma} = T^{(0)}_{\mu\nu\rho\sigma} + \frac{\lambda}{m^2} T^{(1)}_{\mu\nu\rho\sigma} + O(\lambda^2) \qquad \text{(flat space)} , \tag{4.38}$$

around the massive free boson's conserved current,[19]

$$T^{(0)}_{\mu\nu\rho\sigma} = \Phi \overleftrightarrow{\partial}_{(\mu} \overleftrightarrow{\partial}_\nu \overleftrightarrow{\partial}_\rho \overleftrightarrow{\partial}_{\sigma)} \Phi , \tag{4.39}$$

---

[17]See discussion on how to define the $T\overline{T}$ deformation in $AdS_2$ [75, 76]. In conformal perturbation theory around a CFT in AdS, the $T\overline{T}$ deformation can be unambiguously defined and given at the leading order by this four-fermion interaction [77].

[18]These currents were studied for example in [66, 82] and can also be found in [83]. They are typically studied in complex coordinates, but here we work covariantly to facilitate the transition to AdS.

[19]This is nothing but the flat-space limit of the free boson's spin-4 current (2.26) written above in AdS.

the leading correction $T^{(1)}_{\mu\nu\rho\sigma}$ clearly needs to be quartic in $\Phi$ in order for the equation of motion to imply conservation of this current at $O(\lambda)$.

The most general quartic ansatz is

$$T^{(1)}_{(\mu\nu\rho\sigma)} = a_1 \partial_{(\mu}\Phi\partial_\nu\Phi\partial_\rho\Phi\partial_{\sigma)}\Phi + a_2\Phi\partial_{(\mu\nu}\Phi\partial_\rho\Phi\partial_{\sigma)}\Phi + a_3\Phi^2\partial_{(\mu\nu}\Phi\partial_{\rho\sigma)}\Phi + a_4\Phi^2\partial_{(\mu\nu\rho}\Phi\partial_{\sigma)}\Phi$$
$$+ a_5\Phi^3\partial_{(\mu\nu\rho\sigma)}\Phi + m^2 b_1 g_{(\mu\nu}\Phi^3\partial_{\rho\sigma)}\Phi + m^2 b_2 g_{(\mu\nu}\Phi^2\partial_\rho\Phi\partial_{\sigma)}\Phi \,. \tag{4.40}$$

However, one finds that it is not possible to consistently satisfy all the components of the conservation equation $\partial^\mu T_{\mu\nu\rho\sigma} = 0$. Indeed, integrable models are not expected to be fully higher-spin symmetric. Instead, as we discussed in Section 3.1, it is possible to arrange 'partial' conservation along (anti)holomorphic directions,

$$\partial^\mu T_{\mu zzz} = \partial^\mu T_{\mu\bar{z}\bar{z}\bar{z}} = 0 \,, \tag{4.41}$$

by choosing the following values of the coefficients:

$$a_1 = 12 \,, \ a_2 = -24 \,, \ a_3 = 6 \,, \ a_4 = 24/5 \,, \ a_5 = -2/5 \,, \ b_1 = -96/5 \,, \ b_2 = 72/5 \,. \tag{4.42}$$

In particular this means it is possible to conserve the top spin component $T_{zzzz}$ by adding a spin-2 trace $T_{\bar{z}zzz}$, but this spin-2 trace operator cannot itself be conserved.

Now let us consider the prospect of a sine-Gordon theory in $\text{AdS}_2$, which might be in some sense integrable. As we explained, the partial conservation (4.41) is consistent in flat space because the associated Killing tensors $\zeta^{\nu\rho\sigma} = \delta^\nu{}_z\delta^\rho{}_z\delta^\sigma{}_z$ and its complex conjugate are just mapped to multiples of themselves under the 2d Lorentz group. In contrast, in AdS, we showed that all Killing tensors (with non-trivial traceless part) can be mapped to each other by acting with AdS isometries. We therefore know that the minimal useful higher-spin current conservation in AdS would be the full conservation $\nabla^\mu T_{\mu\nu\rho\sigma} = 0$. We of course expect that this cannot be arranged for any sine-Gordon deformation in AdS since it is forbidden by the no-go theorem above. Moreover, in the flat-space limit it would imply that the current (4.38) is fully, rather than partially, conserved. As expected, starting from the conserved spin-4 current of a free boson in AdS and modifying the ansatz (4.40) by replacing $\partial \to \nabla$, we indeed find that there is no choice of coefficients $a_i, b_i$ (even depending on the AdS radius) giving current conservation. The conclusion is that, if an integrable sine-Gordon model does exist in AdS2, then it does not have higher spin conserved currents (unlike in flat space).

To further illustrate the difference between 2d flat space and AdS, let us consider the role of similar (anti)holomorphic coordinates in AdS,

$$ds^2 = \frac{4\,dw\,d\bar{w}}{(1 - w\bar{w})^2} \,, \tag{4.43}$$

which is just the Poincaré disk presentation. Performing a rotation that fixes the center of AdS $w \to e^{i\alpha}w$, one finds that the 2d abelian spin of a local operator continues to be counted by the number of holomorphic ($w$) minus the number of anti-holomorphic ($\bar{w}$)

indices. Additionally, in the flat space limit one recovers the standard (anti)holomorphic coordinates. Moreover, just like in flat space, it is also possible in AdS to set

$$\delta^\nu{}_w \, \delta^\rho{}_w \, \delta^\sigma{}_w \left( \nabla^\mu T_{\mu\nu\rho\sigma} \right) = 0 \,, \tag{4.44}$$

by choosing the coefficients

$$a_1 = \frac{60m^2}{6 + 5m^2} \,, \ a_2 = \frac{-120m^2}{6 + 5m^2} \,, \ a_3 = \frac{30m^2}{6 + 5m^2} \,, \ a_4 = \frac{24m^2}{6 + 5m^2} \,,$$
$$a_5 = \frac{-2m^2}{6 + 5m^2} \,, \ b_1 = \frac{-4(19 + 24m^2)}{6 + 5m^2} \,, \ b_2 = \frac{36(1 + 2m^2)}{6 + 5m^2} \ , \tag{4.45}$$

which reproduce (4.42) in the flat space limit. However, the key difference is that translations along the Poincaré disk coordinates are **not** isometries of AdS, and in particular equation (4.44) does not violate the theorem proven in Section 3.1. It is therefore not a standard conservation equation, and cannot be used to construct conserved charges since[20]

$$\left( \nabla^\mu \, \delta^\nu{}_w \, \delta^\rho{}_w \, \delta^\sigma{}_w \right) T_{\mu\nu\rho\sigma} \neq 0 \,. \tag{4.46}$$

This is compatible with flat-space integrability because in the flat space limit the vector $\xi^\mu = \delta^\mu{}_w$ does become Killing, and commutes with covariant derivatives. In AdS of radius $L$, the failure of the Killing equation and the breaking of the current's conservation is of order $1/L^2$.

Our inability to build higher-spin conserved currents and charges for the tree level sine-Gordon perturbation is unsurprising. Were we able to preserve our higher-spin charges, we would expect operators that start in the same higher-spin multiplet, like $\phi^2$ and $\mathcal{O}_{2,2}$, to remain integer-spaced as we turn on the perturbation. However, it is well known that anomalous dimensions for double trace operators are not constant for a $\Phi^4$ interaction. They are instead given by [84, 85]

$$\gamma_n = \frac{(2n)!(\Delta_\phi)_n^4(4\Delta_\phi - 1)_{2n}}{(n!)^2(2\Delta_\phi)_n^2(2\Delta_\phi)_{2n}^2} \,, \tag{4.47}$$

where $\gamma_n$ is the anomalous dimension of the primary operator $\mathcal{O}_{2,2n}$ with dimension $2\Delta_\phi + 2n$. Crucially, these anomalous dimensions depend on $n$ and asymptote to $1/n^2$ at large $n$. The same argument also implies that the extremal solutions to the crossing equation discussed in [48], which at leading order in the vicinity of GFB agree with a quartic interaction in AdS, cannot preserve higher-spin charges.

## 4.5 Sum rules for bulk data

So far in this section, we have derived consequences of higher-spin conserved charges from the boundary's point of view, without making reference to the bulk operators. In this last subsection, which closely parallels the work of [9], we will explicitly consider the bulk currents and study their two-point functions and form-factors to derive constraints on the

---

[20]We thank Petr Kravchuk for reporting to us that he arrived at the same conclusion independently.

Boundary Operator Expansion (BOE). This is the bulk-to-boundary expansion, which expresses local AdS fields as convergent sums of boundary conformal primary and descendant operators [1] (see also review in [8]), schematically

$$\Phi_{\text{AdS}}(x,y) = \sum_{\Delta} b_{\Phi\Delta}\, y^{\Delta}\, (\mathcal{O}_{\Delta}(x) + \dots)\,, \tag{4.48}$$

The sum runs over CFT primary operators, $b_{\Phi\Delta}$ are the BOE coefficients, and the dots denote descendant contributions.

**Bulk two-point functions**  Our goal in the first part of this section will be to derive a sum rule relating the BOE coefficients to the short-distance behaviour of the bulk theory, making use of higher-spin conservation. We begin by assuming that a local spin-4 current is covariantly conserved in AdS$_2$,

$$\nabla^{\mu} T_{\mu\nu\rho\sigma} = 0\,. \tag{4.49}$$

This in particular implies that the spin-2 trace of the spin-4 current is individually conserved, and hence we expect it to be a linear combination of the stress tensor and the possible improvement terms discussed in Section 2.2 (labeled by the index $i$):

$$\nabla^{\mu} T^{(4)}_{\mu\nu} = 0\,, \qquad T^{(4)}_{\mu\nu} = g^{\rho\sigma} T_{\mu\nu\rho\sigma} = \sum_{i} \mu_i T^i_{\rho\sigma}\,. \tag{4.50}$$

Importantly, the improvement terms can contribute to the BOE data, so cannot be neglected. Nonetheless, we will be able to derive relations between the BOE coefficients of these spin-2 objects and the original spin-4 current.

We first write the conservation equation in complex coordinates $\mathrm{z} = x + iy, \bar{\mathrm{z}} = x - iy$ defined in terms of the Poincaré metric (2.3), and consider the zzz component

$$\nabla_{\bar{\mathrm{z}}} T_{\mathrm{zzzz}} + \nabla_{\mathrm{z}} T_{\bar{\mathrm{z}}\mathrm{zzz}} = -(\mathrm{z} - \bar{\mathrm{z}})^2 \partial_{\bar{\mathrm{z}}} T_{\mathrm{zzzz}} + \sum_{i} \mu_i \left( \frac{4 T^i_{\mathrm{zz}}}{\mathrm{z} - \bar{\mathrm{z}}} + \partial_{\mathrm{z}} T^i_{\mathrm{zz}} \right) = 0\,, \tag{4.51}$$

where we used the definition (4.50) of the trace and wrote out the covariant derivatives explicitly. We can now compute a bulk two-point function between this equation and our spin-4 current. This will lead to a differential relation involving the two-point function of two spin-4 currents and the two-point function of the spin-4 current with the spin-2 current. Such spinning bulk two-point functions admit a block expansion, which can be derived from the BOE. The general form, for bulk operators of $O(2)$ spin $\ell_1$ and $\ell_2$ reads (denoting $T_{(\ell_1)} \equiv T_{(\ell_1)\,\mathrm{z}\dots\mathrm{z}}$)

$$\langle T_{(\ell_1)}(\mathrm{z}_1, \bar{\mathrm{z}}_1) T_{(\ell_2)}(\mathrm{z}_2, \bar{\mathrm{z}}_2) \rangle = \tag{4.52}$$

$$\sum_{\Delta} b_{\ell_1,\Delta}\, b_{\ell_2,\Delta}\, y_1^{-|\ell_1|} y_2^{-|\ell_2|} \left( \frac{\bar{\mathrm{z}}_1 - \bar{\mathrm{z}}_2}{\mathrm{z}_1 - \bar{\mathrm{z}}_2} \right)^{\ell_1} \left( \frac{\mathrm{z}_1 - \bar{\mathrm{z}}_2}{\mathrm{z}_1 - \mathrm{z}_2} \right)^{\ell_2} (4\xi)^{-\Delta}\, {}_2F_1\left( \Delta - \ell_1, \Delta + \ell_2, 2\Delta, -\frac{1}{\xi} \right),$$

where $\xi$ is the chordal distance

$$\xi = \frac{(x_1 - x_2)^2 + (y_1 - y_2)^2}{4 y_1 y_2}\,. \tag{4.53}$$

We can now plug the block decomposition into the two-point function between the spin-4 current and the conservation equation and require it to be satisfied block by block. This leads to the following relation between spin-4 and spin-2 BOE coefficients

$$4b_{\ell=4,\Delta}(\Delta - 4) - (\Delta + 2) \sum_i \mu_i b_{\ell=2,\Delta}^i = 0 \,. \qquad (4.54)$$

This is the spin-4 analogue of the relation between the BOE coefficients of the stress tensor and its scalar trace derived in [9] and reported below in (4.61). We discuss the spin-$J$ version in Appendix B.

A second constraint on the BOE data can be derived by analysing the UV limit of the two-point function, which is assumed here to be a CFT. This short distance limit is simply given by

$$\langle T_{(\ell)}(z_1, \bar{z}_1) T_{(\ell)}(z_2, \bar{z}_2) \rangle \sim \frac{c_\ell/2}{(z_1 - z_2)^{2\ell}} \,, \qquad (4.55)$$

which manifests the chiral structure of the UV CFT, and introduces the constants $c_\ell$ sometimes known as higher-spin central charges [86–88]. Remarkably, this short distance power law turns out to be satisfied term by term in the block expansion.[21] Matching the short distance limit of the block expansion (4.52) to the UV CFT (4.55) leads to a sum rule relating the UV to the BOE data for any value of the coupling along the RG flow

$$\sum_\Delta \frac{b_{\ell,\Delta}^2 \Gamma(2\ell) \Gamma(2\Delta)}{4^{\Delta - \ell} \Gamma(\Delta + \ell)^2} = c_\ell/2 \,, \qquad (4.56)$$

which once again is the higher-spin generalization of the two-point sum rules of [9].

**Two-particle form factors and local blocks**  Another interesting observable that leads to constraints on the BOE data is the two-particle form-factor, i.e. a three-point function with one bulk and two boundary operators. When the bulk operator is a conserved current and we integrate it over a co-dimension 1 slice, we get a correlator with two-boundary operators and a conserved charge. We have made extensive use of this construction in Section 2.3, to determine the action of charges on local operators. Now we will instead use it to constrain the BOE. To fix notation, we first review the sum rule that comes from the action of the momentum operator studied in [9]. Some further technical details are given in Appendix D.

By the definition of the momentum operator as an integral of the stress tensor, we have

$$\langle \mathcal{O}(x_1) \, P \, \mathcal{O}(x_2) \rangle = \int_0^\infty dy \, \langle \mathcal{O}(x_1) T_{xx}(x, y) \mathcal{O}(x_2) \rangle = -\frac{2\Delta_{\mathcal{O}}}{x_{12}^{2\Delta_{\mathcal{O}}+1}} \,, \qquad (4.57)$$

where $\mathcal{O}$ is an arbitrary scalar primary on the boundary. We already saw this equation in the case of the free massive boson, but is true non-perturbatively as the Ward identity for translations. The idea is to use the fact that the form factor admits a block decomposition

---

[21]This happens just for conserved operators. The short distance singularity is generically only reproduced by the infinite sum over boundary operators.

that follows from expanding the bulk operator with the BOE. We first decompose the stress-tensor into its spin-2 traceless and spin-0 irreducible components: $T_{xx} = T_{(2)\,xx} + \frac{1}{2}g_{xx}\Theta$. The traceless spin-2 component can be expanded as

$$\langle \mathcal{O}(x_1)T_{(2)\,xx}(x,y)\mathcal{O}(x_2)\rangle = \frac{t_{xx}}{x_{12}^{2\Delta_{\mathcal{O}}}} \sum_\Delta c_{\mathcal{O}\mathcal{O}\Delta} b_{\ell=2,\Delta} h_\Delta(\chi)\,. \tag{4.58}$$

Here, the cross-ratio $\chi$ is given by

$$\chi = \frac{y^2(x_1 - x_2)^2}{(y^2 + (x - x_1)^2)(y^2 + (x - x_2)^2)}\,, \tag{4.59}$$

the conformal block $h_\Delta(\chi)$ is given explicitly in (D.4) and $t_{xx}$ is the position-space tensor structure obtained from (D.2). Similarly, we have the scalar trace part

$$\langle \mathcal{O}(x_1)\Theta(x,y)\mathcal{O}(x_2)\rangle = \frac{1}{x_{12}^{2\Delta_{\mathcal{O}}}} \sum_\Delta c_{\mathcal{O}\mathcal{O}\Delta} b_{\Theta\Delta}\, g_\Delta(\chi)\,, \tag{4.60}$$

with the conformal block given in (D.6). Conservation of the stress tensor relates the BOE coefficients of the traceless and trace part

$$\Delta b_{\Theta\Delta} - 4(\Delta - 2)b_{\ell=2,\Delta} = 0\,. \tag{4.61}$$

Combining the contributions of the traceless and traceful parts and performing the integral in (4.57) term by term in the BOE expansion (which is only valid for small enough $\Delta_{\mathcal{O}}$) one gets the naive sum rule

$$\sum_\Delta \frac{4\sqrt{\pi}\,\Delta\,\Gamma(\Delta + \frac{1}{2})}{(\Delta + 1)\,\Gamma(\frac{\Delta}{2} + 1)^2}\, c_{\mathcal{O}\mathcal{O}\Delta} b_{\ell=2,\Delta} = -\Delta_{\mathcal{O}}\,. \tag{4.62}$$

However, this sum actually diverges for $\Delta_{\mathcal{O}}$ sufficiently large. The way around this problem is to use the fact that the bulk theory is assumed to be local: this forces physical form factors such as (4.60) only to be singular when operators enter each other's lightcones, namely at $\chi \leq 0$. However, in addition to this physical branch cut, each block $g_\Delta$ has an unphysical branch cut at $\chi \geq 1$, which must cancel out in the sum over blocks [8, 9, 89]. Analogously to the Polyakov bootstrap in the context of the conformal crossing equation, this locality constraint can be characterised by defining some manifestly 'local blocks' [8], with the same $\chi \leq 0$ cut as a regular block, but with no $\chi \geq 1$ cut:

$$G_\Delta^\alpha(\chi) = \chi^\alpha \int_{-\infty}^0 \frac{d\chi'}{2\pi i}\frac{1}{\chi' - \chi}\mathrm{Disc}(\chi'^{-\alpha}g_\Delta(\chi'))\,. \tag{4.63}$$

For $\alpha$ sufficiently large, a form factor is local if and only if one may replace the blocks by local blocks $(g_\Delta \to G_\Delta^\alpha)$ in its BOE expansion.

We can similarly define local blocks for any spin $\ell$ of the bulk operator. Following the technique described in [9], it is straightforward to perform the dispersion integral directly, giving

$$G_{\Delta,\ell}^\alpha = \chi^{\Delta/2}\,{}_2F_1\left(\frac{\Delta}{2} - \frac{\ell}{2}, \frac{\ell}{2} + \frac{\Delta}{2}; \Delta + \frac{1}{2}; \chi\right) \tag{4.64}$$

$$- \chi^\alpha \frac{\Gamma\left(\Delta + \frac{1}{2}\right)\Gamma\left(\alpha - \frac{\ell}{2}\right)\Gamma\left(\frac{\ell}{2} + \alpha\right)}{\Gamma\left(\alpha - \frac{\Delta}{2} + 1\right)\Gamma\left(\alpha + \frac{\Delta}{2} + \frac{1}{2}\right)\Gamma\left(\frac{\Delta-\ell}{2}\right)\Gamma\left(\frac{\ell+\Delta}{2}\right)}\,{}_3F_2\left(\begin{matrix}1 & \alpha - \frac{\ell}{2} & \frac{\ell}{2} + \alpha \\ \alpha - \frac{\Delta}{2} + 1 & \alpha + \frac{\Delta}{2} + \frac{1}{2}\end{matrix}; \chi\right)\,,$$

where the first line coincides with the ordinary spin-$\ell$ block and the second line can be interpreted as subtracting away the unphysical branch cut at $\chi \geq 1$. These local blocks then have the property that the sum over exchanged operators can always be swapped with the integral in (4.57). Hence, by using local blocks instead of blocks, the tail of the naive sum rule (4.62) is effectively subtracted, leaving a sum rule that is valid and converges for all values of $\Delta_{\mathcal{O}}$.

**Spin-4 form factor sum-rule** To derive the spin-4 sum rule, we begin with the integrated form factor associated to $Q_{-3}^{(4)}$

$$\langle \mathcal{O}(x_1) \, Q_{-3}^{(4)} \, \mathcal{O}(x_2) \rangle = \int_0^\infty dy \, \langle \mathcal{O}(x_1) \, T_{xxxx}(x, y) \, \mathcal{O}(x_2) \rangle = \frac{2 \cdot 8(2\Delta_{\mathcal{O}})_3}{x_{12}^{2\Delta_{\mathcal{O}}+3}} \,, \qquad (4.65)$$

which follows from equation (4.11) and from orthogonality of the boundary two-point function. We now need to introduce a block decomposition for the form-factor of the spin-4 current. The spin-4 current should first be decomposed into its traceless and lower spin components. It is easy to show that

$$T_{xxxx} = \mathcal{T}_{xxxx} + g_{xx} \mathcal{T}_{xx}^{(4)} + \frac{3}{8} g_{xx} g_{xx} \Theta^{(4)} \,, \qquad (4.66)$$

where $\mathcal{T}_{xxxx}$ is the traceless spin-4 tensor, the traceless spin-2 component is $\mathcal{T}_{xx}^{(4)}$ and $\Theta^{(4)}$ is the scalar trace.[22] Therefore, we only need to determine the block decomposition of the traceless spin-4 contribution. Using embedding space, we find

$$\langle \mathcal{O}(x_1) \, \mathcal{T}_{xxxx}(x, y) \, \mathcal{O}(x_2) \rangle = \frac{(t^2)_{xxxx}}{x_{12}^{2\Delta_{\mathcal{O}}}} \sum_\Delta c_{\mathcal{O}\mathcal{O}\Delta} b_{\ell=4,\Delta} \, f_\Delta(\chi) \,, \qquad (4.67)$$

with $f_\Delta(\chi)$ the spin-4 conformal block given in (D.9). We once again proceed by performing the integral term by term, which leads to the naive sum rule

$$\sum_{i,\Delta} \mu_i \, b_{\ell=2,\Delta}^i c_{\mathcal{O}\mathcal{O}\Delta} \kappa_4(\Delta) = 2(2\Delta_{\mathcal{O}})_3 \,, \qquad (4.68)$$

where we introduced the naive spin-4 kernel

$$\kappa_4(\Delta) \equiv \frac{4\sqrt{\pi} \, \Delta \, \Gamma(\Delta + \frac{1}{2})}{(\Delta + 1) \, \Gamma(\frac{\Delta}{2} + 1)^2} \frac{(\Delta - 2)(\Delta + 1)}{(\Delta - 4)(\Delta + 3)} \,. \qquad (4.69)$$

For sufficiently small $\Delta_{\mathcal{O}}$ the sum rule actually converges and we can test it in GFB as discussed in Appendix D. In the general case, the sum rule is divergent and we will instead make use of the aforementioned local blocks. The integral over local blocks leads to a new kernel $\kappa_4(\Delta, \alpha)$, containing a piece proportional to the old kernel plus a subtraction term. This kernel is rather complicated, but is given explicitly in (D.11). The sum rule then reads

$$\sum_{i,\Delta} \mu_i \, b_{\ell=2,\Delta}^i c_{\mathcal{O}\mathcal{O}\Delta} \, \kappa_4(\Delta, \alpha) = 2(2\Delta_{\mathcal{O}})_3 \,. \qquad (4.70)$$

---

[22]We use $\mathcal{T}$ and $\mathcal{T}^{(4)}$ instead of the previous notation for $O(2)$ irreducible representations to avoid confusion with the stress tensor—see (4.58).

Despite the complicated-looking kernel, the sum rules can be tested, for example numerically, in the GFB. The sum rules converge faster as $\alpha$ is increased, in a manner that could be estimated as in [9] for the spin-2 case. It is also interesting to note that for the special values $\alpha = \Delta_{\mathcal{O}} + n$, the kernels develop zeroes at an infinite number of double trace locations and only a finite number of GFB operators contribute.

While we have introduced new constraints on the bulk data stemming from the existence of higher-spin conserved currents, it is not clear from this point of view why theories with higher-spin symmetries should be so few and far between—i.e. why the above no-go theorem for deformations of free fields holds. This may follow because, thanks to the trace relations (4.54), the new sum rules (4.70) contain the same spin-2 data as (4.62). It would be interesting to study these sum rules more systematically, perhaps by including further higher-spin currents and trying to narrow down the space of AdS$_2$ theories that satisfy them. It could also be fruitful to study the flat-space limit and the connection to well-known integrable models, but we leave these directions for future work.

## 5   Bulk CFTs and Virasoro symmetry

So far we have understood the consequences and structure of higher-spin currents and charges for both bulk and boundary observables. However, all of our applications have been centered around massive free fields in AdS (i.e. generalized free field CFTs). We saw that free fields have an intricate tower of higher-spin current and charges, with the consequences presented in Section 4. Moreover, we showed that they do not admit interacting perturbations preserving any of these higher-spin currents or charges. In this section, we explore a different class of examples that satisfy the higher-spin constraints and will also turn out to resist deformations: 2d CFTs with Virasoro symmetry subject to boundary conditions that satisfy the Cardy property [57]

$$T(z) = \overline{T}(\overline{z}) , \qquad y = 0 , \tag{5.1}$$

where we once again used complex coordinates $z = x + iy$ and the standard 2d CFT notation $T = T_{zz} , \overline{T} = \overline{T}_{\bar{z}\bar{z}}$. These theories contain one diagonal copy of the left and right Virasoro algebras, which preserves the boundary and ensures that the boundary spectrum is still organised into Virasoro multiplets.

There is a 1-1 equivalence between CFTs in AdS space and boundary conformal field theories (BCFTs) on the flat upper half-plane (UHP), since these spaces are related by a Weyl transformation, $g_{\mu\nu}^{\mathrm{AdS}} = y^{-2} g_{\mu\nu}^{\mathrm{UHP}}$. The correlation functions are related by

$$\Big\langle \mathcal{O}_1^{(\ell_1)}(x_1, y_1) \cdots \mathcal{O}_n^{(\ell_n)}(x_n, y_n) \, \hat{\mathcal{O}}_{n+1}(x_{n+1}) \cdots \hat{\mathcal{O}}_{n+m}(x_{n+m}) \Big\rangle_{\mathrm{AdS}} \tag{5.2}$$

$$= \Big( \prod_{i=1}^{n} y_i^{\Delta_i - \ell_i} \Big) \Big\langle \mathcal{O}_1^{(\ell_1)}(x_1, y_1) \cdots \mathcal{O}_n^{(\ell_n)}(x_n, y_n) \, \hat{\mathcal{O}}_{n+1}(x_{n+1}) \cdots \hat{\mathcal{O}}_{n+m}(x_{n+m}) \Big\rangle_{\mathrm{UHP}} ,$$

where $\mathcal{O}_i^{(\ell_i)}$ are spin-$\ell_i$ bulk primaries with scaling dimensions $\Delta_i$, and $\hat{\mathcal{O}}_i$ are boundary primaries. Crucially, we see that boundary correlation functions are not modified. Therefore, to understand how Virasoro symmetry realizes the higher-spin symmetries, we may simultaneously treat both the flat UHP and AdS on an equal footing.

In particular, conserved charges are identical in AdS and the UHP, since they can be written as $Q = \int dx^\rho \sqrt{g} g^{\mu\nu} \varepsilon_{\nu\rho} J_\mu$, where $J_\mu \equiv \zeta^{\nu_2 \cdots \nu_J} T_{\mu\nu_2 \cdots \nu_J}$. The line element only depends on the 2d metric through the Weyl-invariant combination $\sqrt{g} g^{\mu\nu}$. Moreover, conserved currents $T_{\mu\nu_2 \cdots \nu_J}$ with $\Delta = \ell$ are invariant under the transformation (5.2), as are the conformal Killing tensors $\zeta^{\nu_2 \cdots \nu_J}$.

## 5.1 Embedding Virasoro in the HS paradigm

Any bulk 2d CFT has multiple infinite towers of higher-spin conserved currents that are trivially built from the (anti)holomorphic stress tensor:

$$\bar{\partial} T(z) = \bar{\partial} : T^2(z) := \bar{\partial} : T^3(z) := \bar{\partial} : (\partial T)^2(z) := \ \ldots \ = 0 \,. \tag{5.3}$$

Here we used the notation of flat space but, importantly, since conformal invariance implies that these currents are all traceless, then covariant conservation in AdS is actually guaranteed by flat space conservation. For example, for a spin-4 current, we have

$$\nabla_{\bar{z}} T_{zzzz,\mathrm{AdS}} = \left( \frac{z - \bar{z}}{2i} \right)^2 \bar{\partial} T_{zzzz,\mathrm{flat}} = 0 \,. \tag{5.4}$$

Adding the fact that the BOE of chiral operators is regular, which avoids the boundary issues discussed in Section 2.2, the charges in AdS are therefore guaranteed to exist. One might naively think that the (diagonal) Virasoro modes $\hat{L}_n$ are the analogues of our higher-spin charges, but this is not the case, since all but the global ones do not annihilate both in- and out-vacuua:

$$\hat{L}_{-n}|0\rangle \neq 0 \,, \qquad \langle 0| \hat{L}_n \neq 0 \,, \qquad n \geq 2 \,. \tag{5.5}$$

This is because they are obtained by integrating a current against a *conformal* Killing vector, not a Killing vector, of AdS, which is hence not regular everywhere. Instead, these charges produce non-trivial states when acting on the vacuum, for example

$$\hat{L}_{-3}|0\rangle = \left( \int_S dz \frac{1}{z^2} T(z) - \int_S d\bar{z} \frac{1}{\bar{z}^2} \overline{T}(\bar{z}) \right) |0\rangle = \partial \mathrm{D}(0)|0\rangle \,, \tag{5.6}$$

where D is the displacement operator, the (regular) boundary limit of the bulk stress-tensor $T$. On the other hand, the higher-spin charges of our interest should annihilate the vacuum, and be obtained by integrating a higher-spin current against an AdS Killing tensor. As we reviewed in Section 2, rank-$(J-1)$ Killing tensors are obtained as sums of $(J-1)$-fold products of the Killing vectors: translation, dilation and special conformal Killing vectors, which scale as 1, $z$ and $z^2$ respectively. This means that multiplying the current (which has a finite BOE, by holomorphy) by any non-negative power of $z$ up to $z^{2(J-1)}$ leads to a regular integral ensuring that the resulting charges annihilate the vacuum, as required.[23] Let us then take the bulk spin-4 current obtained by normal ordering

$$T_4(z) \equiv : T^2(z) := L_{-2} L_{-2}(z) = \oint_z d\zeta \frac{T(\zeta) T(z)}{\zeta - z} \,, \tag{5.7}$$

---

[23]As expected, the AdS charges are identical to the UHP charges, since the three boundary-preserving combinations of the six regular conformal Killing vectors of $\mathbb{R}^2$ are precisely the three ordinary Killing vectors of AdS$_2$.

and define a charge by integrating this current against the constant Killing tensor associated to translation, in analogy with $Q_{J=4,p^3} \equiv Q_{-3}^{(4)}$ defined previously. Here we get [90, 91]

$$Q_{J=4,p^3} = \int_S dz\, T_4(z) - \int_S d\bar{z}\, \overline{T}_4(z) = 2 \sum_{m=-1}^{\infty} \hat{L}_{-3-m}\hat{L}_m \,, \tag{5.8}$$

which follows from a standard calculation with modes of normal ordered products. Crucially, this charge lives in the universal enveloping algebra of Virasoro and not in the algebra itself, but clearly annihilates both the in- and out-vacuua. In fact, the zero-weight mode of the same higher-spin current plays a crucial role in the integrable structure of 2d CFTs, and is part of the so-called KdV hierarchy of commuting higher-spin charges [29, 92–94], recently reviewed in [95]. In our case, the charge $Q_{J=4,p^3}$ manifestly has weight 3 and indeed commutes with the momentum generator (cf. the KdV charges, which instead commute with the dilation operator). This puts us in the same axiomatic situation as Section 4, meaning the higher-spin charges and their properties derived there are guaranteed to hold here as well. By taking commutators as in Section 4.1, one in fact builds a copy of the higher-spin algebra $hs[\lambda]$ inside the universal enveloping algebra of Virasoro, where $\lambda$ is related to the CFT's central charge $c$.

The consequences of the higher-spin charges can be straightforwardly checked by standard Virasoro manipulations. Beginning with the action of HS charges, we note that all Virasoro primaries $\phi_h$ (with weight $h$) must be HS primaries.[24] The HS charges are therefore expected to act on them as (see equation (4.11))

$$\left[Q_{J=4,p^3}, \phi_h\right] = \alpha\, \partial^3\phi_h + \beta\, \partial\mathcal{O}_2 + \gamma\, \mathcal{O}_3 \,. \tag{5.9}$$

Indeed, using the state-operator correspondence, and acting with $Q_{J=4,p^3}$ on a Virasoro primary state, we get

$$Q_{J=4,p^3}|h\rangle = 2\hat{L}_{-2}\hat{L}_{-1}|h\rangle + 2h\hat{L}_{-3}|h\rangle \,. \tag{5.10}$$

The RHS can be checked to agree with (5.9) with the coefficients

$$\alpha = \frac{1+5h}{1+3h+2h^2}\,, \quad \beta = \frac{6h}{2+h}\,, \quad \gamma = \frac{2h(1-h)}{2+h}\,, \tag{5.11}$$

where $\mathcal{O}_2$ and $\mathcal{O}_3$ correspond to the following Virasoro descendants (which are global conformal primaries):

$$|\mathcal{O}_2\rangle = \left(\hat{L}_{-2} - \frac{3}{2(1+2h)}\hat{L}_{-1}\hat{L}_{-1}\right)|h\rangle \,,$$
$$|\mathcal{O}_3\rangle = \left(\hat{L}_{-3} - \frac{2}{h}\hat{L}_{-2}\hat{L}_{-1} + \frac{1}{h(1+h)}\hat{L}_{-1}\hat{L}_{-1}\hat{L}_{-1}\right)|h\rangle \,. \tag{5.12}$$

We can now verify the higher-spin Ward identities for three- and four-point functions. For the four-point case (4.15), we simply need to compute the correlation functions

---

[24]This follows because the HS charges do not mix different Virasoro multiplets and Virasoro primaries are lowest-weight states.

$\langle \mathcal{O}_2 \phi_h \phi_h \phi_h \rangle$ and $\langle \mathcal{O}_3 \phi_h \phi_h \phi_h \rangle$. Since $|\mathcal{O}_2\rangle$ and $|\mathcal{O}_3\rangle$ are Virasoro descendents of $|h\rangle$, then these are fixed by Virasoro symmetry in terms of $\langle \phi_h \phi_h \phi_h \phi_h \rangle$, by the relation

$$\langle \left( \hat{L}_{-k} \phi_h(x) \right) \phi_h(x_1) \dots \phi_h(x_n) \rangle = \mathcal{L}_{-k}^{(x)} \langle \phi_h(x) \phi_h(x_1) \dots \phi_h(x_n) \rangle , \qquad (5.13)$$

with the differential operator

$$\mathcal{L}_{-k}^{(x)} = \sum_{i=1}^{n} \left( \frac{(k-1)h}{(x_i - x)^k} - \frac{1}{(x_i - x)^{k-1}} \partial_{x_i} \right) . \qquad (5.14)$$

One can then verify that the four-point Ward identity (4.15) is indeed satisfied for an arbitrary 4-point function of identical Virasoro primaries $\langle \phi_h \phi_h \phi_h \phi_h \rangle$.

The fact that this Ward identity was not enough to fix the functional form of this correlator in Section 4.2 is therefore unsurprising: any boundary conformal field theory contains solutions! Even in the special case $\gamma = 0$ (which distinguished $\phi^2$ from other HS primaries in GFB theories), bulk Virasoro CFTs also provide many solutions. Indeed, by taking $\phi_h$ to have a null state at level 3—e.g. the primary $\phi_{1,3}$ in any minimal model with appropriate boundary conditions—one can set $\gamma = 0$, since the corresponding null operator $\mathcal{O}_3$ vanishes in correlation functions.

## 5.2 No-go for higher-spin preserving deformations of CFTs in AdS

In Section 4.3, we saw that interacting deformations of generalized free fields preserving any HS charge are generically impossible. We now derive an analogous result for CFTs in AdS:

**No-go theorem for deformations of CFTs in AdS:** *Starting from a unitary Virasoro CFT in AdS$_2$ and deforming by a single relevant or irrelevant Virasoro primary, it is impossible to preserve any spin-4 charges.*[25]

In particular, this excludes an AdS analog of the integrable $\phi_{1,3}$ deformation of minimal models. As will be clear momentarily, the generalization of the theorem to any spin is conceptually straightforward, but technically involved.

The boundary spectrum of a CFT in AdS has a Virasoro vacuum module, which contains the displacement operator D of dimension 2; and other quasi-primaries of integer dimension, such as D$^2$ with dimension 4:

$$|\mathrm{D}\rangle = \hat{L}_{-2}|0\rangle , \qquad |\mathrm{D}^2\rangle = \left( \hat{L}_{-4} - \frac{5}{3} \hat{L}_{-2} \hat{L}_{-2} \right) |0\rangle . \qquad (5.15)$$

First, let us assume that a spin-4 charge is conserved. Using the Virasoro algebra and the $Q_{J=4,p^3}$ charge constructed above, it is easy to see that acting with $Q_{J=4,p^3}$ on D generates only D and D$^2$:

$$\left[ Q_{J=4,p^3}, \mathrm{D} \right] = \alpha \, \partial^3 \mathrm{D} + \beta \, \partial \mathrm{D}^2 , \qquad (5.16)$$

---

[25]We will see below that there is one and only one exception: The thermal deformation of the Ising model which is equivalent to a GFF.

with coefficients $\alpha = (22 + 5c)/30$, $\beta = -9/5$, $\gamma = 0$, with $c$ the UV central charge of the bulk theory.

Now let us imagine turning on a coupling $\lambda$ in the bulk associated to a single Virasoro primary $\Phi$. The action of the charge can be modified at order $\lambda$, but since D and D$^2$ appear with a non-zero coefficient in the UV, by continuity they cannot acquire a zero coefficient perturbatively. This implies that their anomalous dimensions must be equal,

$$\delta\Delta_{\mathrm{D}^2} = \delta\Delta_{\mathrm{D}} , \tag{5.17}$$

However, by conformal perturbation theory in AdS, one can explicitly show that [77, 96]

$$\delta\Delta_{\mathrm{D}^2} = \frac{20c + 25\Delta_\Phi(\Delta_\Phi - 2) + 64}{2(5c + 22)} \delta\Delta_{\mathrm{D}} . \tag{5.18}$$

These are consistent in one and only one unitary case:[26]

$$c = \frac{1}{2} \qquad \text{and} \qquad \Delta_\Phi = 1 , \tag{5.19}$$

which is precisely the mass deformation of the free fermion CFT, i.e. GFF theory.

Similarly, increasingly cumbersome but straightforward calculations are expected to show that, if a spin-$J$ current is conserved, integer spacing between the scaling dimensions of D and D$^2$ is guaranteed by the charge $Q^{(J)}_{-J+1}$. Then (5.18) continues to ensure a unique HS-preserving deformation. It would be interesting to carefully derive this in future.

## 6    Conclusions and outlook

In this paper, we initiated the study of QFTs in AdS$_2$ with higher-spin charges, regarded as a likely feature of putative integrable theories. We discussed the construction of boundary charges in terms of bulk currents, while carefully understanding boundary contributions justifying the finiteness of these charges. We also understood how these charges are realized in generalized free fields, noting that this justifies the integer spacing in their spectrum of primaries.

We then established in full generality that conservation of a higher-spin current in AdS$_2$ along any Killing tensor implies that the current is fully conserved, leading to a large set of HS charges. Using these charges, we tried to bootstrap the space of theories which are HS symmetric. We established integer spacing of the spectrum and determined relations between correlation functions of operators in the same HS multiplet. Using the same techniques, we proved for generic masses that weakly interacting deformations of generalized free fields cannot preserve any higher-spin charges. We also studied constraints on correlation functions involving bulk operators, establishing sum rules for bulk two-point functions and form factors. Finally, we considered Virasoro CFTs in the bulk and showed how they realize the HS symmetry. Furthermore, we showed that up to one exception, any relevant or irrelevant deformation by a single Virasoro primary must break the spin-4

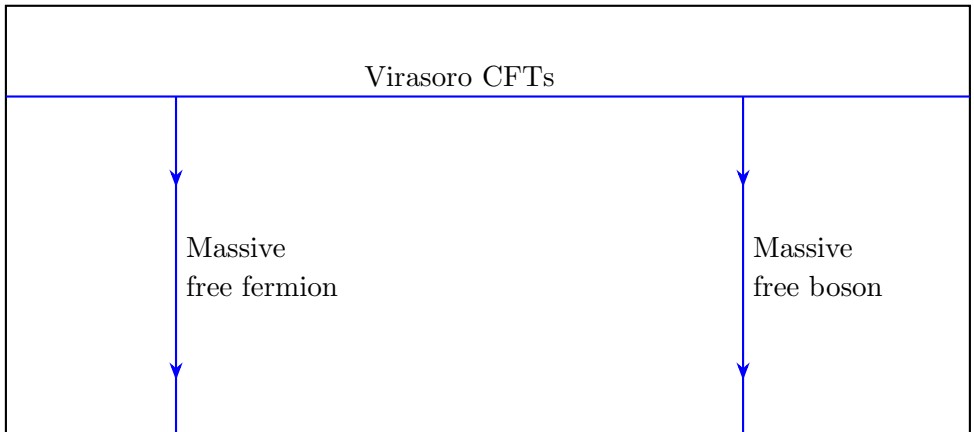

**Figure 7**. A cartoon of the QFTs in $AdS_2$, with those known to have higher-spin symmetries shown in blue. The RG flow runs schematically from top to bottom here. In this work we have shown that there are no other QFTs with higher-spin symmetries perturbatively close to the blue lines. On the other hand there could be others far away from the blue lines.

charges. We present a cartoon of the space of QFTs in $AdS_2$ known to have HS symmetries in Figure 7.

While higher-spin charges often appear in integrable QFTs in flat space, we have shown that they are disfavoured in interacting QFTs in $AdS_2$. This is fundamentally related to the fact that the Poincaré algebra is non-simple, allowing currents to be partially conserved along special null directions in flat space, while the AdS symmetry algebra is simple and forces currents to be fully conserved—a strong requirement not even satisfied by integrable models in flat space. Natural analogues of our questions about integrable models on $AdS_2$ would also apply in the maximally symmetric spaces $S^2$ and $dS_2$, and we would expect similar conclusions, since their symmetry algebras are also simple.[27] Instead, perhaps it would be more fruitful to consider integrable QFTs on non-trival 2d geometries with non-simple symmetry algebras, which may admit partial conservation equations along special directions. A possible class of examples are the theories with explicit dependence on the 2d coordinates constructed in [98]: these theories are promising candidates to be integrable and some of them may be interpretable as field theories coupled to curved 2d geometries— although we are not aware of any cases interpretable as an $AdS_2$ geometry.

Although higher-spin charges seem difficult to arrange in AdS, one can certainly write down theories whose flat-space limits are integrable and admit higher-spin conservation laws (e.g. sine-Gordon theory). It would be interesting in the future to consider the weak breaking of higher-spin conservation at large AdS radius, where it is still expected to place constraints on the theory. A similar setting where weakly broken higher-spin currents have

---

[26]The only other possibility is that $\delta\Delta_D = \delta\Delta_{D^2} = 0$, but this can only happen for a marginal deformation [77, 96].

[27]The 2d Schwinger model was recently shown to be exactly solvable in de Sitter space [97]. This is presumably consistent with our expectations since, at least in flat space, it is related to a massive free scalar by bosonisation.

been found to impose substantial constraints is in 3d conformal models at large $N$ [54, 99–102].

On the other hand, as well as local higher-spin charges, an important role is played in integrable models by non-local currents and charges (see, e.g., [103, 104]). These are typically related to algebraic structures beyond standard Lie algebras, like quantum groups and Yangians [105–107]. It is possible that such non-local charges may still be arranged in AdS, even when local higher-spin ones cannot—and it would be very interesting to explore this in the future.

In fact, it is those algebraic structures that appear more prevalently in the literature concerning integrability in $\mathcal{N} = 4$ sYM [108] and its $\frac{1}{2}$-BPS circular Wilson line. That 1d defect CFT's planar spectrum has an integrable description [109–111] inherited from the 4d theory, and is solvable by the Quantum Spectral Curve [112, 113]. Moreover, it is dual to an $\mathrm{AdS}_2$ theory of fluctuations around the minimal-area string worldsheet ending on the Wilson line, given by the Nambu-Goto string in $\mathrm{AdS}_5 \times \mathrm{S}^5$ in a static-type gauge [114]. This open-string worldsheet theory may naturally be expected to be integrable, like its integrable closed-string equivalent, and is one of the most interesting examples motivating the exploration of integrable models in $\mathrm{AdS}_2$. However, the theory itself has non-local features, being given by a square-root-type action. It may therefore be natural to expect it to support non-local, rather than local, charges. On top of the integrability of its spectral problem, an interesting question is whether the Wilson line's correlation functions—i.e. the boundary correlators of the $\mathrm{AdS}_2$ worldsheet theory—have some special integrable properties analogous to flat-space factorised scattering [115, 116].

## Acknowledgments

We are very grateful to Andrea Cavaglià for collaboration at the early stages of this project. We are happy to thank Benjamin Basso, Florent Baume, Connor Behan, Carlos Bercini, Max Downing, Barak Gabai, Victor Gorbenko, Sebastian Harris, Alexandre Homrich, Apratim Kaviraj, Shota Komatsu, Petr Kravchuk, Sylvain Lacroix, Jeremy Mann, Dalimil Mazáč, Miguel Paulos, João Penedones, Slava Rychkov, Volker Schomerus, Jörg Teschner, Arkady Tseytlin and Sasha Zhiboedov for useful discussions and comments. AA further thanks Shailesh Lal for introducing him to higher-spin symmetry and for pointing out reference [60] which sparked the development of this work, Davide Lai for discussions on the history of integrability, and Andreia Gonçalves for continued support.

AA received funding from the German Research Foundation DFG under Germany's Excellence Strategy – EXC 2121 Quantum Universe – 390833306 and is funded by the European Union (ERC, FUNBOOTS, project number 101043588). Views and opinions expressed are however those of the author(s) only and do not necessarily reflect those of the European Union or the European Research Council Executive Agency. Neither the European Union nor the granting authority can be held responsible for them. NL was supported at the École Normale Supérieure by the Institut Philippe Meyer, and at the University of Amsterdam by the European Research Council under the European Union's Seventh Framework Programme (FP7/2007-2013), ERC Grant agreement ADG 834878.

MM is funded by the Italian Ministry of University and Research through the Rita Levi Montalcini program.

## A   More on the generalized free boson

In this appendix we perform explicit computations in GFB theory that illustrate some points in the main text. In Appendix A.1 we describe an explicit fine-tuning of the spin-4 current using improvement terms. In Appendix A.2 we compute explicit four-point functions of two-particle operators and in Appendix A.3 we determine the action of higher-spin charges on three-particle operators.

### A.1   Explicit improvements

In Section 2.2 of the main text, we showed that higher-spin charges can be made finite, regardless of potential divergences in the BOE of the higher-spin currents, by showing that the flux of the current contracted with a Killing tensor is a total derivative cf. (2.16).

In this appendix, we will give an alternative proof of this fact for GFB, by showing that it is possible to pick appropriate improvement terms that can be tuned to eliminate the contributions of the dangerous light operators to the BOE of the current. We define a modified current $T'$,

$$T'_{\mu\nu\rho\sigma} = T_{\mu\nu\rho\sigma} + \eta_0 \, \Delta T^{(0)}_{\mu\nu\rho\sigma} + \eta_2 \, \Delta T^{(2)}_{\mu\nu\rho\sigma} \,, \tag{A.1}$$

which has two free parameters $\eta_0$ and $\eta_2$ meant to tune away the divergences associated to the two boundary operators $\phi^2$ and $\mathcal{O}_{2,2}$ with dimensions $2\Delta_\phi$ and $2\Delta_\phi+2$ respectively, and could potentially spoil the finiteness of the spin-4 charge for $\Delta_\phi < 1/2$. The parameters multiply improvement terms, $\Delta T^{(0)}_{\mu\nu\rho\sigma}$ and $\Delta T^{(2)}_{\mu\nu\rho\sigma}$ which are conserved tensor operators with a non-trivial BOE but that do not modify the higher-spin charge. The $\Delta T^0$ term which is identically conserved, was given in equation (2.18) and we will use $\mathcal{O} = \Phi^2$ for the GFB.[28] A second improvement-like term $\Delta T^{(2)}$ can be obtained by acting on the energy-momentum tensor with a differential operator

$$\Delta T^{(2)}_{\mu\nu\rho\sigma} = \big(g_{(\mu\nu}(\Box - 9) - \nabla_{(\mu}\nabla_{\nu})T_{\rho\sigma)} \,, \tag{A.2}$$

but is only conserved upon imposing conservation of the spin-2 operator that was used to build it and is therefore not identically conserved. Nonetheless, it does not modify the higher-spin charges, as would be expected, since such a 'fake' higher-spin current exists in any theory with a conserved energy-momentum tensor. To see this, we consider a charge on a closed surface cf. (2.7), and use Stokes theorem to obtain a bulk integral of the divergence of the current. This divergence is given by

$$\nabla^\mu \Delta T^{(2)}_{\mu\nu\rho\sigma} = \big(g_{(\nu\rho}(\Box - 16) - \nabla_{(\nu}\nabla_{\rho})\nabla^\mu T_{\sigma)\mu} \,, \tag{A.3}$$

---

[28]Additional scalars such as $(\partial\Phi)^2$ could also be used to generate additional improvement terms.

and is proportional to the divergence of the stress-tensor. Integrating by parts to act on the Killing tensor, we find that

$$\left(g_{\nu\rho}(\Box - 16) - \nabla_{(\nu}\nabla_{\rho)}\right)\zeta^{\nu\rho\sigma} = 0\,, \tag{A.4}$$

for Killing tensors that only have overlap with spin-3 (in the sense of section 3). This is because the differential operator above is a projector to lower-spin: the difference between the casimir operator and the associated eigenvalue. Hence, despite not being identically conserved, we can freely use $\Delta T^{(2)}$ to improve the BOE of our spin-4 current as it gives vanishing higher-spin charges.

With two improvement terms at hand, we now want to compute the bulk-boundary correlators

$$\langle T_{\mu\nu\rho\sigma}(x,y)\,\phi^2(x_1)\rangle\,,\quad \langle T_{\mu\nu\rho\sigma}(x,y)\,\mathcal{O}_{2,2}(x_1)\rangle\,, \tag{A.5}$$

as well as the analogous correlators with $T \to \Delta T^{(0)}$ and $T \to \Delta T^{(2)}$. They are easily evaluated by Wick contractions (taking appropriate covariant derivatives of bulk-to-boundary propagators), but lead to rather long expressions. However, all the correlators agree in their position dependence (which is tensorial as discussed above), and is obtained from the embedding space expression [55, 56]

$$\langle \mathcal{O}(P_1)\, T_{M_1\ldots M_J}(X)\rangle W^{M_1}\ldots W^{M_J} = \frac{(W \cdot P_1)^J}{(P_1 \cdot X)^{\Delta_\mathcal{O}+J}}\,. \tag{A.6}$$

through a projection to AdS

$$\langle \mathcal{T}_{xxxx}(x,y)\phi^2(x_1)\rangle = b_{\ell=4,\phi^2}\frac{\partial X^A}{\partial x}\frac{\partial X^B}{\partial x}\frac{\partial X^C}{\partial x}\frac{\partial X^D}{\partial x}\frac{K_A K_B K_C K_D (W \cdot P_1)^4}{4!(\frac{d-1}{2})_4 (P_1 \cdot X)^{\Delta_\phi^2+4}}\,, \tag{A.7}$$

where we considered the traceless part of the spin 4 tensor, $K_A$ is the Todorov operator of [56], and the limit $d \to 1$ is finite. After the universal position dependence is extracted out, it is easy to find the ratios of spin 4 BOE coefficients

$$\frac{b_{\Delta T_4^{(0)},\phi^2}}{b_{T_4,\phi^2}} = \frac{3 + 4\Delta_\phi(2+\Delta_\phi)}{27}\,,\quad \frac{b_{\Delta T_4^{(0)},\mathcal{O}_{2,2}}}{b_{T_4,\mathcal{O}_{2,2}}} = \frac{(3+2\Delta_\phi)(5+2\Delta_\phi)}{243} \tag{A.8}$$

and

$$\frac{b_{\Delta T_4^{(2)},\phi^2}}{b_{T_4,\phi^2}} = -\frac{\Delta_\phi(3+2\Delta_\phi)}{6}\,,\quad \frac{b_{\Delta T_4^{(2)},\mathcal{O}_{2,2}}}{b_{T_4,\mathcal{O}_{2,2}}} = -\frac{(\Delta_\phi - 1)(5+2\Delta_\phi)}{54}\,, \tag{A.9}$$

which in turn determine the desired coefficients

$$\eta_0 = -\frac{27(5+4\Delta_\phi)}{15 + 16\Delta_\phi + 4\Delta_\phi^2}\,,\qquad \eta_2 = -\frac{24}{5+2\Delta_\phi}\,. \tag{A.10}$$

## A.2 Correlators with $\mathcal{O}_{2,2}$

In the main text we discussed the four-point Ward identity (4.15) in the case $\gamma = 0$, where the operators are identified as two-particle operators in GFB, namely $\phi^2$ and $\mathcal{O}_{2,2}$. The properly normalized operators are given by

$$\mathcal{O}_0 = \phi^2/\sqrt{2}\,,\qquad \mathcal{O}_2 = \left(\partial\phi\partial\phi - \frac{2\Delta_\phi}{1+2\Delta_\phi}\phi\partial^2\phi\right)\Big/\sqrt{\frac{8\Delta_\phi^2(1+4\Delta_\phi)}{1+2\Delta_\phi}}\,. \tag{A.11}$$

It is straightforward to compute the four-point functions by Wick contractions. Using the conventions of equation (4.16), we find the correlators

$$f(z) = 1 + z^{4\Delta_\phi} + \left(\frac{z}{1-z}\right)^{4\Delta_\phi} + 4\left(z^{2\Delta_\phi} + \left(\frac{z}{1-z}\right)^{2\Delta_\phi} + \frac{z^{4\Delta_\phi}}{(1-z)^{2\Delta_\phi}}\right), \qquad (A.12)$$

$$g(z) = -4\Delta_\phi\sqrt{\frac{1+2\Delta_\phi}{1+4\Delta_\phi}}\left(\left(\frac{z}{1-z}\right)^{2\Delta_\phi} + z^{2\Delta_\phi}(1-z)^2 + \frac{z^{4\Delta_\phi+2}}{(1-z)^{2\Delta_\phi}}\right), \qquad (A.13)$$

which satisfy the Ward Identity (4.15) with the ratio of coefficients

$$\beta/\alpha = -\frac{6\Delta_\phi\sqrt{1+2\Delta_\phi}\sqrt{1+4\Delta_\phi}}{\Delta_\phi+1}, \qquad (A.14)$$

which can easily be independently fixed by considering the normalized action of the higher-spin charge on $\phi^2$. Indeed, writing

$$N\phi\partial^3\phi - \partial^3(\mathcal{O}_0) - \frac{\beta}{\alpha}\partial(\mathcal{O}_2) = 0, \qquad (A.15)$$

and making use of the explicit form of the operators above, we can individually set to zero the coefficients of $\phi\partial^3\phi$ and $\partial\phi\partial^2\phi$, determining both $N$ and $\beta/\alpha$, precisely such that the Ward identity is satisfied.

## A.3   Action on multi-traces

In this appendix we examine how multi-particle operators $\phi^n$ in GFB are acted upon by the Higher-spin charges. The expectation is that the generic action (4.11) will be satisfied with all coefficients being non-vanishing. The reason $\gamma$ is expected to be non-vanishing for $n \geq 3$ but not for $n = 2$ is because there is exactly one primary of dimension $n\Delta_\phi + 3$ for $n \geq 3$, as can easily be counted with characters. For the sake of concreteness let us analyse the case of $\phi^3$. We have

$$\left[Q_3, \phi^3\right]_{\text{GFB}} = N\phi^2\partial^3\phi, \qquad (A.16)$$

for some normalization $N$. The derivatives act on a single constituent field by the same argument given in the main text for quadratic operators. We want to interpret this action as

$$\left[Q_3, \phi^3\right] = \alpha\partial^3(\phi^3) + \beta\partial(\mathcal{O}_{3,2}) + \gamma\mathcal{O}_{3,3}, \qquad (A.17)$$

where

$$\mathcal{O}_{3,2} = N_2\left(\phi(\partial\phi)^2 - \frac{2\Delta_\phi}{1+2\Delta_\phi}\phi^2\partial^2\phi\right), \qquad (A.18)$$

$$\mathcal{O}_{3,3} = N_3\left(\phi^2\partial^3\phi - \frac{3(1+\Delta_\phi)}{\Delta_\phi}\phi\partial\phi\partial^2\phi + \frac{(1+\Delta_\phi)(1+2\Delta_\phi)}{\Delta_\phi^2}(\partial\phi)^3\right), \qquad (A.19)$$

are the triple trace primaries. We can then match the two actions and $Q_3$, to find the coefficients

$$\frac{\alpha}{N} = \frac{(1+\Delta_\phi)(1+2\Delta_\phi)}{3(1+3\Delta_\phi)(1+6\Delta_\phi)}, \; \frac{\beta}{N} = -\frac{6(1+\Delta_\phi)(1+2\Delta_\phi)}{N_2(2+3\Delta_\phi)(1+6\Delta_\phi)}, \; \frac{\gamma}{N} = \frac{2\Delta_\phi^2}{N_3(1+3\Delta_\phi)(2+3\Delta_\phi)},$$
$$(A.20)$$

where we indeed confirm that $\gamma$ is non-vanishing. The form of low-lying multi-particle primaries is similar, so very similar formulas should hold for $\phi^n$.

## B    Details on the BOE

In this appendix we verify some of the statements made in Section 2.2 by using the BOE. We start by studying the constraints that conservation imposes on the spectrum of exchanged operators. We first consider the even $J$ case, and comment on the minor changes necessary to address the odd spin case at the end. Tensors in $\mathrm{AdS}_2$ can be decomposed into $O(2)$ spin components (the rotation that fixes a point in Euclidean AdS). This is most easily done working in complex Poincaré coordinates $\mathrm{z} = x + iy$ and $\bar{\mathrm{z}} = x - iy$, which make the $O(2)$ spin manifest:

$$T_{(\ell)\,\mathrm{z}...\mathrm{z}} \equiv g^{\mu_1\mu_2} \cdots g^{\mu_{J-\ell-1}\mu_{J-\ell}}\, T_{\mu_1...\mu_{J-\ell}\,\mathrm{z}...\mathrm{z}} \,, \tag{B.1}$$

and similarly for the bottom component, where all z's are replaced by $\bar{\mathrm{z}}$'s. For a fully conserved current (2.6), the BOE coefficients of all the different $O(2)$ spin components $\ell = J, J - 2, \ldots, 2, 0$ are related by conservation. For instance, the conservation equation for the top $O(2)$ component,

$$\nabla_{\bar{\mathrm{z}}} T_{\mathrm{z}...\mathrm{z}} + \nabla_{\mathrm{z}} T_{\bar{\mathrm{z}}\mathrm{z}...\mathrm{z}} = 0 \,, \tag{B.2}$$

relates $T_{(J)}$ to $T_{(J-2)}$. Using the leading BOE behavior $T_{(\ell)} \sim b_{\ell,\Delta} y^{\Delta-\ell}$, one finds the constraint

$$4(\Delta - J)b_{J,\Delta} - (\Delta + J - 2)b_{J-2,\Delta} = 0 \,. \tag{B.3}$$

However, due to the full conservation condition (2.6), the $\ell = J - 2$ component is itself a conserved current, and must therefore satisfy the same equation! Iterating, we find that for all $\ell$

$$4(\Delta - \ell)b_{\ell,\Delta} - (\Delta + \ell - 2)b_{\ell-2,\Delta} = 0 \,. \tag{B.4}$$

This equation has very different consequences for the exchange of parity-even and parity-odd boundary operators. Indeed, consider a parity-even operator of generic dimension $\Delta_+$, which can couple to the spin-0 component, i.e. $b_{\ell=0,\Delta_+} \neq 0$. In this situation, all the higher $\ell$ coefficients are fixed in terms of the scalar one, and there is no constraint on $\Delta_+$. Let us consider instead a parity-odd operator of dimension $\Delta_-$. This operator can never couple to the scalar mode. Therefore, by considering equation (B.4) with $\ell = 2$, we conclude that either the operator does not couple to the spin-2 mode, or it must have the special scaling dimension $\Delta_- = 2$. Proceeding by analyzing the same equation for larger values of $\ell$, we conclude that parity-odd operators can only appear with the special scaling dimension $\Delta_- = 2, 4, \ldots, J$, and must therefore be tuned away to ensure the charge exists. The same equation also allows us to conclude that parity-even operators with the same special dimensions $\Delta_+ = 2, 4, \ldots, J$ only couple to the modes with $\ell \geq \Delta_+$. This is in fact familiar from CFT in AdS (BCFT) where the traceless currents only couple to exactly one parity-even operator of dimension $\Delta_+ = J$, the so-called higher-spin displacement operators. For the odd-spin case, the relation (B.4) between BOE coefficients remains the same, but the

roles of exchanged parity-even and odd operators remain the same. parity-odd operators have an unconstrained dimension $\Delta_-$, while parity-even operators can only take the values $\Delta_+ = 1, 3, \ldots, J$.

We now move on to verifying that the flux across the boundary of an even (odd) higher-spin current contracted with a Killing tensor—and projected onto even (odd) parity boundary operators—is a total derivative, i.e. equations (2.16) and (2.19). We once again focus solely on the primary operators—but we expect the same conclusion to hold for descendants (this is shown for $J = 1, 2, 3, 4$ in Section 2.2). We specialize to the even $J$ case, as the odd case follows with trivial substitutions. Since Killing tensors are linear combinations of products of Killing vectors (see (2.9)), it is important to note the near-boundary limit of the Killing vectors (2.4): they are regular along the $x$ direction ($p^x = 1$, $d^x = x$, $k^x \sim x^2$) and suppressed by $y$ along the $y$ direction ($p^y = 0$, $d^y = y$, $k^y = 2xy$). Now, when exchanging parity-even boundary operators, the BOE of the conserved currents is

$$
\begin{aligned}
(\text{even} \# \text{of } x\text{'s}): \quad & T_{x\ldots xy\ldots y}(x, y) \sim y^{\Delta - J} O(x) \,, \\
(\text{odd} \# \text{of } x\text{'s}): \quad & T_{x\ldots xy\ldots y}(x, y) \sim y^{\Delta - J + 1} \partial_x O(x) \,,
\end{aligned}
\tag{B.5}
$$

with the same proportionality coefficient. Since there is at least one $y$ index due to computing the flux across a surface parallel to the boundary, the leading power in the BOE is $y^{\Delta - J + 1}$. Terms with this behavior can only occur in two ways: we can either take all indices of the Killing tensor along the $x$ direction (where there is no suppression by powers of $y$), or take exactly one index along the $y$ direction (where the suppression by one power of $y$ is compensated by the enhancement in the BOE). The flux then reads

$$
\zeta^{\mu_1 \ldots \mu_{J-1}}(x, y) \, T_{\mu_1 \ldots \mu_{J-1} y}(x, y)\big|_{O_+} \sim \zeta^{x\ldots x} y^{\Delta - J + 1} \partial_x O + \zeta^{x\ldots xy} y^{\Delta - J} O \,,
\tag{B.6}
$$

and terms where more indices along $y$ are further suppressed. Now, let us start by a Killing tensor of the form

$$
\zeta = p^{n_p} d^{n_d} k^{n_k} \,, \qquad n_p + n_d + n_k = J - 1 \,,
\tag{B.7}
$$

it is easy to see that the asymptotic behavior of the Killing tensor components is

$$
\zeta^{x\ldots x} \sim x^{n_d + 2n_k} \,, \qquad \zeta^{x\ldots xy} \sim y(n_d \, x^{n_d + 2n_k - 1} + 2n_k x^{n_d + 2n_k - 1}) \,,
\tag{B.8}
$$

from which we conclude that

$$
\zeta^{\mu_1 \ldots \mu_{J-1}}(x, y) \, T_{\mu_1 \ldots \mu_{J-1} y}(x, y)\big|_{O_+} \sim \partial_x \left( y^{\Delta - J + 1} x^{n_d + 2n_k} O(x) \right) \,,
\tag{B.9}
$$

as required. The generic Killing tensor is a linear combination of the monomials (B.7), hence the conclusion follows by linearity.

The analysis of the odd spin case proceeds analogously, with the role of odd and even number of $x$ indices being exchanged in (B.5).

## C  Killing tensors in AdS$_2$

This appendix is devoted to some representation theoretical statements about Killing tensors in AdS$_2$. First, we establish an isomomorphism between symmetric tensors in embedding space [55, 56] and Killing tensors. Then we take advantage of the link between the former and spherical harmonics to compute in closed form the Killing tensors appearing in the basis in Section 3.1, equation (3.12). We will explicitly construct a one-to-one correspondence between the Killing tensors and constant symmetric tensors at a point in AdS. This provides an alternative proof of our counting of Killing tensors in Section 3.

### C.1  Killing tensors and tensors in embedding space

The embedding space formalism consists in realizing Eulidean AdS$_d$ as a hypersurface embedded in $(d+1)-$dimensional Minkowski spacetime:

$$X^A \eta_{AB} X^B = -L^2 \ , \qquad X^0 > 0 \ , \tag{C.1}$$

where $X^A$ are cartesian coordinates in Minkowski, and $A = 0, \ldots, d$. If we pick a parametrization of AdS $X^A(x^\mu)$, $\mu = 1, \ldots, d$, the following completeness relation is useful:

$$\delta^A_B = \partial_\mu X^A \partial^\mu X_B - \frac{X^A X_B}{L^2} \ , \tag{C.2}$$

where greek (latin) indices are lowered with the AdS (Minkowski) metric. Lorentz transformations in embedding space act as isometries in AdS, and the corresponding Killing vectors are tangent to the hypersurface. Indeed, by eq. (C.2), the Lorentz generators are written as

$$J_{AB} = X_A \partial_B - X_B \partial_A = \left( X_A \frac{\partial X_B}{\partial x_\mu} - X_B \frac{\partial X_A}{\partial x_\mu} \right) \partial_\mu \ . \tag{C.3}$$

In the case of AdS$_2$, we can further exploit the epsilon tensor to reorganize the Killing vectors:

$$\xi^\mu_A \equiv \epsilon_{ABC} J^{BC} = L \sqrt{g} \, \partial_\alpha X_A \epsilon^{\alpha\mu} \ . \tag{C.4}$$

In our conventions, $\epsilon_{012} = \epsilon_{12} = 1$, and the appropriate metric tensor is used to raise the indices of the Levi-Civita symbols. Evaluating eq. (C.4) in Poincaré coordinates, one finds the following correspondence with the Killing vectors defined in eq (2.4):

$$\xi^{0,\mu} + \xi^{1,\mu} = p^\mu \ , \quad \xi^{0,\mu} - \xi^{1,\mu} = k^\mu \ , \quad \xi^{2,\mu} = d^\mu \ . \tag{C.5}$$

Organizing the Killing vectors in a triplet of $so(2,1) \simeq sl(2,\mathbb{R})$, as in eq. (C.4), is useful in the classification of the Killing tensors. In spaces of constant curvature, all Killing tensors are symmetrized products of Killing vectors [50]. One can therefore write the general Killing tensor in terms of a constant symmetric tensor in embedding space, as in eq. (2.9):

$$\zeta^{\mu_1 \cdots \mu_\ell}(x) = \alpha^{A_1 \ldots A_\ell} \xi^{\mu_1}_{A_1}(x) \ldots \xi^{\mu_\ell}_{A_\ell}(x) \ . \tag{C.6}$$

This correspondence between symmetric tensors $\alpha$ in embedding space and Killing tensors $\zeta^{\mu_1 \cdots \mu_\ell}(x)$ is one-to-one, as we will now show. Suppose one is given a Killing tensor $\zeta$: we

will construct a unique symmetric tensor $\alpha$ corresponding to it. First, the action of Lie derivatives $\mathcal{L}_A$ on $\zeta$ along the Killing vectors $\xi_A$ corresponds to infinitesimal rotations of the indices of the tensor $\alpha$. In other words, both sides of the correspondence transform in a reducible representation of $so(2,1)$ (that of symmetric tensors). Then, one can use the pair of differential operators $(\mathcal{L}_A\mathcal{L}^A, \mathcal{L}_B)$ to project $\alpha$ onto each eigenvalue of the Casimir and of a component chosen as the generator of the Cartan subalgebra. Since this projection identifies a unique tensor in embedding space, one can reconstruct all the components of $\alpha$ uniquely, thus proving the one-to-one correspondence. The isomorphism between the two spaces is confirmed by the counting of independent Killing tensors provided in [50], which for AdS$_2$ reads

$$\frac{(\ell + 2)!}{2\ell!}, \tag{C.7}$$

Which agrees with the number of independent components of a symmetric tensor of rank $\ell$ in three dimensions. For completeness, we note that in the general AdS$_d$ case, the Killing tensors transform as a rank-2 antisymmetric tensor in embedding space—see (C.3). Correspondingly, the tensor $\alpha$ in (C.6) decomposes in $so(d,1)$ representations whose Young tableaux has two rows of equal length [51–53].

## C.2 The basis (3.12) from spherical harmonics

We saw above that Killing tensors in AdS$_2$ transform in reducible representations of $so(2,1)$, which decompose into irreducible ones. We will now take advantage of the isomorphism with tensors in embedding space to explicitly construct each irreducible component.

The relevant irreducible representations of $so(2,1)$ consist of symmetric traceless tensors in embedding space. Therefore, the spin-$\ell$ component of the Killing tensor (C.6) is obtained by subtracting all the traces from $\alpha$. We note in passing that the AdS Killing tensor corresponding to the Minkowski metric in embedding space is simply the AdS metric, i.e. the quadratic Casimir:

$$\eta^{AB}\xi_A^\mu \xi_B^\nu = L^2 g^{\mu\nu} . \tag{C.8}$$

This is of course consistent with the fact that the metric is invariant under isometries, and so a rank-$\ell$ Killing tensor proportional to $g_{\mu\nu}$ transforms in a representation with spin strictly lower than $\ell$.

The elements of the basis (3.12) are the Killing tensors obtained from traceless symmetric embedding space tensors that diagonalize $J_{01}$, i.e., have a definite weight under dilatations. Our aim is to compute this basis. We begin by contracting each index with a polarization vector $\eta^\mu$. This is tantamount to replacing the Killing vectors on the RHS of (C.6) with

$$\eta^A \equiv \eta^\mu \xi_\mu^A . \tag{C.9}$$

This maps $\zeta$ to a polynomial in three variables, which, abusing notation, we will call $\alpha$:

$$\zeta_{\mu_1 \ldots \mu_\ell}(x)\, \eta^{\mu_1} \ldots \eta^{\mu_\ell} \equiv \alpha(\eta^0, \eta^1, \eta^2) . \tag{C.10}$$

Homogeneous polynomials obtained from traceless symmetric tensors are closely related to spherical harmonics. Indeed, after a Wick rotation, $\alpha(i\eta^3, \eta^1, \eta^2)$ is harmonic in $\eta^1$, $\eta^2$, $\eta^3$.

The connection to spherical harmonics is obtained by going to spherical coordinates in the three-dimensional space spanned by $(i\eta^3, \eta^1, \eta^2)$:

$$(\eta^1, \, \eta^2, \, \eta^3) = r(\sin\theta\sin\phi, \cos\theta, \sin\theta\cos\phi) \ . \tag{C.11}$$

Then, any harmonic polynomial $\alpha$ is in the span of the set

$$\alpha_\ell^m(i\eta^3, \eta^1, \eta^2) = r^\ell Y_\ell^m(\theta, \phi) \ , \tag{C.12}$$

where $Y_\ell^m$ is the spherical harmonic of degree $\ell$ and order $m$. Upon replacing the $\eta$'s with $p$, $k$, $d$ via (C.5,C.9), each polynomial in (C.12) is, up to a prefactor, one element of the basis (3.12). This was the reason for orienting the spherical coordinates as in (C.11), so that dilatations are singled out as the Cartan.

For completeness, let us make these polynomials fully explicit. We start from the definition

$$Y_\ell^m(\theta, \phi) = N(\ell, m)e^{im\phi}P_\ell^m(\cos\theta) \ , \tag{C.13}$$

where $N(\ell, m)$ will be chosen at the end and the associated Legendre functions read

$$P_\ell^m(x) = (-1)^m \, 2^\ell \left(1 - x^2\right)^{m/2} \sum_{n=m}^{\ell} \frac{n!}{(n-m)!} \binom{\ell}{n} \binom{\frac{\ell+n-1}{2}}{\ell} x^{n-m} \ , \quad m \geq 0 \ , \tag{C.14}$$

$$P_\ell^m(x) = (-1)^m \frac{(\ell+m)!}{(\ell-m)!} P_\ell^{-m}(x) \ , \quad m < 0 \ . \tag{C.15}$$

When $m$ is odd, the associated Legendre funcions are not polynomials, but $\alpha_\ell^m$ in (C.12) still is, thanks to the fact that summand in (C.14) vanishes when $\ell - n$ is odd.[29] It is useful to make the latter fact explicit, by changing variable $n \to \ell - 2n$, after which, focusing on $m \geq 0$,

$$P_\ell^m(x) = (-1)^m \, 2^\ell \left(1 - x^2\right)^{m/2} \sum_{n=0}^{\mathrm{Floor}\left(\frac{\ell-m}{2}\right)} \frac{(\ell-2n)!}{(\ell-m-2n)!} \binom{\ell}{\ell-2n} \binom{\ell-n-\frac{1}{2}}{\ell} x^{\ell-m-2n} \ . \tag{C.16}$$

Plugging this expression in (C.12), one gets

$$\alpha_\ell^m(p, k, d)$$

$$= 2^\ell N(\ell, m)(ik)^m \sum_{n=0}^{\mathrm{Floor}\left(\frac{\ell-m}{2}\right)} \frac{(\ell-2n)!}{(\ell-m-2n)!} \binom{\ell}{\ell-2n} \binom{\ell-n-\frac{1}{2}}{\ell} \left(d^2 - pk\right)^n d^{\ell-m-2n}$$

$$= (ik)^m \sum_{a=0}^{\mathrm{Floor}\left(\frac{\ell-m}{2}\right)} \tilde{N}(\ell, m, a)(-pk)^a d^{\ell-m-2a} \ , \tag{C.17}$$

---

[29]Here and in the following, we always set $1/x! = 0$ for $x$ a negative integer, or equivalently we use $x! = \Gamma(x+1)$.

where we abused notation by denoting $p = p^\mu \eta_\mu$ and similarly for $k$ and $d$ in the arguments of $\alpha_\ell^m$. In the last line of (C.17), we defined

$$\tilde{N}(\ell, m, a) = 2^\ell N(\ell, m) \sum_{n=0}^{\text{Floor}\left(\frac{\ell-m}{2}\right)} \frac{(\ell-2n)!}{(\ell-m-2n)!} \binom{\ell}{\ell-2n} \binom{\ell-n-\frac{1}{2}}{\ell} \binom{n}{a} . \quad \text{(C.18)}$$

Notice that $\tilde{N}(\ell, m, a) = 0$ when $a > n$, which justifies the summation range for $a$ in (C.17). Furthermore, one can sum (C.18) up to $n = \infty$, since the summand vanishes for $n > \text{Floor}(\ell - m)/2$. In this way, one obtains[30]

$$\tilde{N}(\ell, m, a) = 2^\ell N(\ell, m) \frac{\sqrt{\pi}(-1)^\ell}{a!(\ell-m)!} {}_3F_2^{\text{reg}} \left(1, \frac{m-\ell}{2}, \frac{1+m-\ell}{2}; 1-a, \frac{1}{2} - \ell; 1\right)$$

$$= N(\ell, m) 2^{-m-2a}(-1)^a \frac{(m+\ell)!}{(m+a)!a!(\ell-m-2a)!} . \quad \text{(C.20)}$$

The last equality follows from first applying the identity

$$_3F_2^{\text{reg}}(a_1, a_2, a_3; b_2, b_3; 1) = \frac{\Gamma(r)}{\Gamma(b_2)} {}_3F_2^{\text{reg}}(b_1 - a_3, b_2 - a_3, r; r + a_1, r + a_2; 1) , \quad \text{(C.21)}$$

where $r = b_1 + b_2 - a_1 - a_2 - a_3$ and then using

$$_3F_2(a_1, a_2, -n; b_1, a_1 + a_2 - b_1 - n + 1; 1) = \frac{(b_1 - a_1)_n (b_1 - a_2)_n}{(b_1)_n (b_1 - a_1 - a_2)_n} , \quad \text{(C.22)}$$

which holds for $n$ a non-negative integer. Plugging this result in (C.17) we find the final formula

$$\alpha_\ell^m(p, k, d) = N(\ell, m) \left(\frac{ik}{2}\right)^m \sum_{a=0}^{\text{Floor}\left(\frac{\ell-m}{2}\right)} \frac{(\ell+m)!}{a!(m+a)!(\ell-m-2a)!} \left(\frac{pk}{4}\right)^a d^{\ell-m-2a} , \tag{C.23}$$

While this expression was obtained for $m > 0$, we see that (C.23) gives a polynomial for all $-\ell \le m \le \ell$, because the summand vanishes for $a < -m$. In fact, repeating the derivation for $m < 0$ starting from the definition (C.15), one gets

$$\alpha_\ell^m(p, k, d) = N(\ell, m) \left(-\frac{ip}{2}\right)^{-m} \sum_{a=0}^{\text{Floor}\left(\frac{\ell+m}{2}\right)} \frac{(\ell+m)!}{a!(-m+a)!(\ell+m-2a)!} \left(\frac{pk}{4}\right)^a d^{\ell+m-2a} , \tag{C.24}$$

and it is easy to see that (C.23) and (C.24) are equal, and so they both define $\alpha_\ell^m$ for $-\ell \le m \le \ell$.

---

[30]We use the following definition for the regularized hypergeometric function:

$$_pF_q^{\text{reg}}(a_1, \ldots, a_p; b_1, \ldots, b_q; x) = \left(\prod_{i=1}^q \frac{1}{\Gamma(b_i)}\right) {}_pF_q(a_1, \ldots, a_p; b_1, \ldots, b_q; x) . \quad \text{(C.19)}$$

The prefactor $N(\ell, m)$ is unimportant, but can be fixed for instance so that the polynomials match the one obtained starting from $p^\ell$ and applying lowering operators, as in (3.12). By explicit computation, one easily guesses

$$N(\ell, m) = 2^\ell i^{-m} \frac{\ell!(\ell + m)!}{(\ell - m)!} . \tag{C.25}$$

For completeness, we also notice that the sum in (C.23) is hypergeometric, and, with the choice (C.25) can be resummed as follows:

$$\alpha_\ell^m(p, k, d) = 2^{\ell-m} \frac{\ell!}{m!} d^{\ell-m} k^m \, {}_2F_1 \left( \frac{m-\ell}{2}, \frac{m-\ell+1}{2}, m+1, \frac{pk}{d^2} \right) . \tag{C.26}$$

The basis (C.23) allows us to give a direct proof of the final step in the theorem of Section 3.1: when evaluated to a specific point, the Killing tensors span the space of all the symmetric tensors. As discussed in the main text, proving this statement is equivalent to checking that, upon imposing $k + p = 0$, (C.23) provides a basis for all the polynomials in two variables $k, d$. This is easy to see. Consider the $a = 0$ term in the sum, with $m \geq 0$. Its coefficient never vanishes, and it produces the monomial $k^m d^{\ell-m}$. By varying $m$, one gets all the monomials, and therefore the set $\alpha_\ell^m(k, -k, d)$, for $m = 0, \ldots \ell$, is a basis for the polynomials in two variables.

# D   Details of form-factor sum rules

In this appendix we will fill in some of the technical gaps left in the derivation of the form-factor sum rules in Section 4.5.

## D.1   Spin-2 sum rule

To obtain the correlator of a symmetric-traceless spin-2 tensor (4.58), we simply need to project to physical space after obtaining the result in the AdS embedding formalism of [56]. The traceless spin-2 component can be written as

$$W^M W^N \langle \mathcal{O}(P_1) \mathcal{O}(P_2) T_{MN}(X) \rangle = \frac{t}{(-2P_1 \cdot P_2)^{\Delta_\mathcal{O}}} \sum_\Delta c_{\mathcal{O}\mathcal{O}\Delta} b_{\ell=2,\Delta} h_\Delta(\chi) , \tag{D.1}$$

where $W^M$ is a null polarization vector, $P$ denotes the embedding coordinate of a boundary point and $X$ the embedding coordinate of a bulk point. We then use the tensor structure

$$t = -\frac{4(W \cdot P_1)(W \cdot P_2)}{(X \cdot P_1)(X \cdot P_2)} , \tag{D.2}$$

the cross-ratio

$$\chi = \frac{-(P_1 \cdot P_2)}{2(X \cdot P_1)(X \cdot P_2)} , \tag{D.3}$$

and the conformal block

$$h_\Delta(\chi) = \chi^{\Delta/2} \, {}_2F_1 \left( \frac{\Delta}{2} + 1, \frac{\Delta}{2} - 1, \Delta + \frac{1}{2}, \chi \right) . \tag{D.4}$$

We also have the scalar trace part

$$\langle \mathcal{O}(P_1)\mathcal{O}(P_2)\Theta(X)\rangle = \frac{1}{(-2P_1 \cdot P_2)^{\Delta_\mathcal{O}}} \sum_\Delta c_{\mathcal{O}\mathcal{O}\Delta} b_{\Theta\Delta} g_\Delta(\chi) \, , \tag{D.5}$$

with the conformal block

$$g_\Delta(\chi) = \chi^{\Delta/2} \, {}_2F_1\left(\frac{\Delta}{2}, \frac{\Delta}{2}, \Delta + \frac{1}{2}, \chi\right) . \tag{D.6}$$

As described in the main text, the sum rule is then derived by combining the traceless and trace contributions, relation their BOE coefficients through (4.61), and performing the integral in (4.57). To write this integral, we find it convenient to work in the frame $x_1 = -1$, $x_2 = 1$ and $x = 0$, where the cross-ratio takes the value $\chi = 4y^2/(1+y^2)^2$. This finally gives the sum rule

$$-2 \int_0^\infty \frac{dy}{y^2} \sum_\Delta c_{\mathcal{O}\mathcal{O}\Delta} b_{\ell=2,\Delta} \left(h_\Delta(\chi) + \frac{\Delta-2}{\Delta} g_\Delta(\chi)\right) = \Delta_\mathcal{O} \, . \tag{D.7}$$

Swapping the integral with the sum in a cavalier fashion is what leads to the naive sum rule (4.62).

## D.2  Spin-4 sum rule

Moving on to the spin-4 sum rule, we once again derive the form factor (4.67) by projecting to physical space the embedding space correlator

$$W^A W^B W^C W^D \langle \mathcal{O}(P_1)\mathcal{O}(P_2)T_{ABCD}(X)\rangle = \frac{t^2}{(-2P_1 \cdot P_2)^{\Delta_\mathcal{O}}} \sum_\Delta c_{\mathcal{O}\mathcal{O}\Delta} b_{\ell=4,\Delta} f_\Delta(\chi) \, , \tag{D.8}$$

where we used that there is a unique tensor structure, and the spin 4 conformal block is given by

$$f_\Delta(\chi) = \chi^{\Delta/2} \, {}_2F_1\left(\frac{\Delta}{2}+2, \frac{\Delta}{2}-2, \Delta+\frac{1}{2}, \chi\right) , \tag{D.9}$$

which is easily obtained by solving the Casimir differential equation with the prefactor above. Working in the convenient frame defined above, making use of the BOE relation (4.54), and combining the contributions from the spin-4, 2 and 0 components, one finds

$$2^{2\Delta_\mathcal{O}+3} \int_0^\infty dz \langle \mathcal{O}T_{xxxx}\mathcal{O}\rangle = \tag{D.10}$$
$$+ 4\sum_i \mu_i^2 \int_0^\infty \frac{dz}{z^4} \sum_\Delta c_{\mathcal{O}\mathcal{O}\Delta} b_{\ell=2,\Delta}^{(i)} \left(\frac{\Delta+2}{\Delta-4} f_\Delta(\chi) + 4h_\Delta(\chi) + 3\frac{\Delta-2}{\Delta} g_\Delta(\chi)\right) .$$

If we once again proceed by performing the integral term by term, we obtain the naive sum rule (4.68).

We can now test this sum rule, as well as the local version (4.70), where the kernel is explicitly given by

$$\kappa_4(\Delta,\alpha) - \kappa_4(\Delta) = \tag{D.11}$$

$$-\frac{\pi 2^{-2\alpha-3}\Gamma(2\alpha+3)\Gamma\left(\Delta+\frac{1}{2}\right)\left(\frac{8(\alpha+2)(2\alpha+3)\,{}_5F_4\left(1,\alpha-1,\alpha-\frac{1}{2},\alpha+\frac{5}{2},\alpha+3;\alpha+1,\alpha+\frac{3}{2},\alpha-\frac{\Delta}{2}+2,\alpha+\frac{\Delta}{2}+\frac{3}{2};1\right)}{\alpha(2\alpha+1)(2\alpha-\Delta+2)(2\alpha+\Delta+1)} + \frac{4}{2\alpha^2-7\alpha+6}\right)}{(\alpha-1)(2\alpha-1)\Gamma\left(\frac{\Delta}{2}-1\right)\Gamma\left(\frac{\Delta}{2}+1\right)\Gamma\left(\alpha-\frac{\Delta}{2}+1\right)\Gamma\left(\alpha+\frac{\Delta}{2}+\frac{1}{2}\right)}$$

$$+\frac{3\pi 2^{-2\alpha-1}(\Delta-2)\Delta\Gamma(2\alpha-1)\Gamma\left(\Delta+\frac{1}{2}\right)(3-2\alpha)\alpha\,{}_3F_2^{\mathrm{reg}}\left(2,\alpha-\frac{1}{2},\alpha+1;\alpha-\frac{\Delta}{2}+2,\alpha+\frac{\Delta}{2}+\frac{3}{2};1\right)}{(2\alpha-3)\Gamma\left(\frac{\Delta}{2}+1\right)^2}$$

$$-\frac{3\pi 2^{-2\alpha-1}(\Delta-2)\Delta\Gamma(2\alpha-1)\Gamma\left(\Delta+\frac{1}{2}\right)(2\alpha+1)\,{}_3F_2^{\mathrm{reg}}\left(1,\alpha-\frac{3}{2},\alpha;\alpha-\frac{\Delta}{2}+1,\alpha+\frac{\Delta}{2}+\frac{1}{2};1\right)}{(2\alpha-3)\Gamma\left(\frac{\Delta}{2}+1\right)^2}$$

$$-\frac{\pi^{3/2}4^{2-2\alpha}\Gamma(2\alpha-3)\Gamma(2\alpha+2)\Gamma\left(\Delta+\frac{1}{2}\right)\,{}_5F_4^{\mathrm{reg}}\left(1,\alpha-\frac{3}{2},\alpha-1,\alpha+1,\alpha+\frac{3}{2};\alpha,\alpha+\frac{1}{2},\alpha-\frac{\Delta}{2}+1,\alpha+\frac{\Delta}{2}+\frac{1}{2};1\right)}{\Gamma\left(\frac{\Delta}{2}-1\right)\Gamma\left(\frac{\Delta}{2}+1\right)},$$

by considering the GFB correlator. To do this, we take the trace of the spin-4 current (2.26) which yields

$$g^{\rho\sigma}T_{\mu\nu\rho\sigma} = -16(3+\Delta_\phi(\Delta_\phi-1))T_{\mu\nu} + 8\Delta_\phi(\Delta_\phi-1)\Delta T_{\mu\nu} + 9g^{\rho\sigma}\Delta T^{(0)}_{\mu\nu\rho\sigma}. \tag{D.12}$$

The last term, which is the trace of a spin 4 improvement, integrates to zero and does not contribute to the sum rule. However the spin 2 improvement does contribute. Therefore, $\sum_i \mu_i b^{(i)}_{\ell=2,\Delta}$ in the sum rule should be read as $\mu_0 b_{T(\ell=2),\Delta} + \mu_1 b_{\Delta T(\ell=2),\Delta}$, where $\mu_0$ and $\mu_1$ can be read-off from (D.12). It is straightforward to compute the form factors, expand in blocks, and read off the BOE coefficients. The spectrum is simply $\Delta = 2\Delta_\phi + 2n$, and the coefficients for the canonical stress tensor read:

$$c_{\phi\phi n}b_{\Theta n} = \frac{2(-1)^{n+1}\Delta_\phi(\Delta_\phi-1)\Gamma(n+\Delta_\phi)^2}{\sqrt{\pi}n!(2\Delta_\phi-1)\Gamma\left(\Delta_\phi-\frac{1}{2}\right)\Gamma(\Delta_\phi)\left(n+2\Delta_\phi-\frac{1}{2}\right)_n}, \tag{D.13}$$

and for the improvement

$$c_{\phi\phi n}b_{\Delta\Theta n} = \frac{4(-1)^{n+1}(\Delta_\phi+n-1)(2\Delta_\phi+2n+1)\Gamma(n+\Delta_\phi)^2\Gamma\left(n+2\Delta_\phi-\frac{1}{2}\right)}{\sqrt{\pi}(2\Delta_\phi-1)\Gamma(n+1)\Gamma\left(\Delta_\phi-\frac{1}{2}\right)\Gamma(\Delta_\phi)\Gamma\left(2n+2\Delta_\phi-\frac{1}{2}\right)}, \tag{D.14}$$

with the relation (4.61) between $b_{\Theta\Delta}$ and $b_{\ell=2,\Delta}$.

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
