# Peer review of "Demystifying integrable QFTs in AdS: No-go theorems for higher-spin charges"

_SciPost Physics_

## Round 2 · Referee Report · Anonymous (Referee 1) · 2025-8-13

Report

In the paper under review the authors investigate the space of QFTs on a fixed AdS$_2$ background that admit higher-spin conserved charges. This is motivated by the goal of formulating integrability criteria for QFTs on curved 2d spaces. One way to define integrability in 2d flat space is the existence of an infinite tower of higher-spin conserved charges in involution, and it is this approach that is paralleled in the paper. The authors find that the space of QFTs on AdS$_2$ that admit local higher-spin conserved currents is highly constrained and formulate a number of no-go theorems. In particular, starting from either the massive free field or a Virasoro CFT on AdS$_2$, with suitable caveats, it is not possible to deform or perturb the theory and preserve higher-spin charges.

The paper is largely well-written and the referencing of the literature is comprehensive. There are a number of technical appendices containing additional details and ensuring reproducibility. There is also a clear conclusion and a detailed abstract and introduction summarising the results objectively.

The paper addresses an interesting question and contains important results providing key insights into the role of integrability on AdS$_2$ and curved spaces more generally. Understanding this could help to solve such QFTs that appear across theoretical physics. I am happy to recommend the paper for publication in SciPost Physics, however, there are a number of points that I would ask the authors to consider first:

  1. In the paper, the authors consider QFTs on AdS$_2$ and demand that the isometries of AdS$_2$ are preserved by any deformation or perturbation. However, integrable field theories on higher-dimensional Euclidean space, such as WZW$_4$ in 4d, typically break space-time symmetries. Do the authors think that deformations or perturbations that partially break the AdS$_2$ isometries could preserve higher-spin charges?

  2. Throughout the paper the authors consider Euclidean QFTs . Euclidean AdS$_2$ has a different structure to Lorentzian AdS$_2$ and its universal cover. Do the authors foresee any issues analytically continuing between the two?

  3. The phrase "boundary CFT" is used in two ways, it either refers to the CFT$_1$ living on the conformal boundary of AdS$_2$ or a CFT on AdS space, with the latter mapped via a conformal transformation to a CFT in the presence of a boundary. It might be helpful to use different phrases to distinguish these two setups.

  4. In footnote 9, page 13 is there a reference that can be provided for this Mathematica package?

  5. I am confused by the ranges of coordinates in eq. (3.27). This metric can be recovered by setting $X_0 \pm X_1 = \cosh\rho e^{\pm \tau}$ and $X_2 = \sinh\rho$. If both $\tau$ and $\rho$ cover the whole real line then this parametrisation covers one of the two disconnected halves, i.e. the one defined by $X_0 > 0$, of Euclidean AdS$_2$.

6.Is it more possible to better explain why taking the commutation relations in eq. (4.1) is "perfectly meaningful ... from the boundary's perspective" on page 24?

  1. On page 50, the authors suggest the possibility of considering other non-trivial geometries. Is there a clear proposal for what would be the meaning of spin in such setups?

  2. What do the indices $M, N, ...$ in eqs.(A.6) and (D.1) refer to?

I also spotted a few possible typographical errors that I list below for the authors' convenience: 1. Page 7, line 4, should $e^\mu_x$ be $\delta^\mu_x$? 2. Page 9, line 11, "see also earlier discussion" should be "see also an earlier discussion". 3. Page 12, line 15, delete the comma after "spin-3". 4. Capitalisation is not always consistent, e.g. "lie" appears on page 17 and "casimir" on page 52. 5. Page 17, line 13, "accordance to" should be "accordance with". 5. Page 17, line 30, "field" should be "fields". 6. Page 20, footnote 12, "the" is repeated. 7. Page 22, line 28, "tensors" should be "tensor". 8. Page 23, in and around eqs. (3.27) and (3.28), $\rho$ is both used as an index and a coordinate on AdS$_2$. 9. Page 34, line 4, "AdS2" should be "AdS$_2$". 10. Page 47, footnote 31, should "cross a close codimension-one" should be "cross a closed codimension-one". 11. Page 51, the sentence containing eq. (A.1) is not clear, in particular the final part starting "and could potentially". 12. Page 55, line 16, "parity" should be "Parity". 13. Page 57, line 21, "Which" should be "which". 14. Page 61, line 9, "relation" should be "relating".

Recommendation

Ask for minor revision

---

## Round 2 · Referee Report · Marco Serone (Referee 2) · 2025-9-12

Strengths

  1. Topic of great interest
  2. The results found are relevant
  3. Concrete cases worked out
  4. Comprehensive bibliography

Weaknesses

  1. Hypotheses not always spelled out clearly
  2. Connection with BCFT literature

Report

I recommend publication in Scipost after minor revision.

Requested changes

The authors investigate the possibility of having integrable field theories on AdS${}_2$ by studying the structure of local higher spin currents and their consequences. Among other results, it is shown that i) in contrast to flat space, higher spin currents in AdS${}_2$ can only be fully conserved, ii) massive free scalar or fermion theories admit no deformations which preserve higher-spin charges, and iii) no deformation of a bulk CFT by a Virasoro primary can preserve any spin-4 charge.

The topic is of great interest, the paper is sound, and the results found are very relevant. It is also appreciated that the authors work out in detail some examples. I think the paper deserves publication in Scipost. However, before being accepted, I would like the authors to clarify to me this point:

  • In section 5 the authors correctly point out that a CFT in AdS can be mapped via a Weyl transformation to a BCFT on the upper half plane, but they do not seem to further comment about the relation between their results and the known ones in BCFT, e.g. in [84]. For example, it is known that in flat space, in presence of a boundary, a bulk $\phi_{1,3}$ deformation in general breaks integrability, unless a localized $\phi_{1,3}$ interaction is also added. The authors should expand on the connection between the known results in BCFT and theirs in AdS, and clarify why (if any) the ``no-go theorem for deformations of CFTs in AdS" in page 43 is really a new result.

In addition, the authors might want to clarify some more or less minor points, and possibly spell out better the hypotheses of their theorems, as required for a no-go theorem like paper. More specifically:

  • In the introduction, page 2, the authors presents their 3 main results without a proper explanation of the assumptions behind. For example, whether the 2d theory is local, the energy-momentum tensor is unique, the nature of the deformations (local vs non-local) mentioned in item 3, or the nature of the boundary conditions in item 3 (b). Some of these assumptions can be deduced by the reader going through the paper, but I think it is important to state them clearly once and for all.

  • The BOE is extensively used in section 2.2 but it is only defined in section 4.5. The authors might want to consider, e.g., to move eq.(4.49) in section 2.2, or adding a few words of explanation of what a BOE is, for further clarity and to avoid to the reader to jump ahead to section 4.5.

  • Page 10, right before eq.(2.17): is [60], and not [9], the correct reference to cite?

  • The analysis in page 10 is not easy to follow. In particular, it seems to me that the existence of an identically conserved tensor is not necessary. Rather, it seems a useful way to get the result without the need to work out the detailed form of the tensor structures in the two-point function between the higher-spin current and the operator O. Close to the boundary, I do not see why it is crucial for a tensor to be identically conserved, as in a local analysis we cannot distinguish closed vs exact forms. Indeed, the property should then apply to the higher-spin current $T$ and not only to its improvement term $\Delta T$. It would be good if the authors could clarify.

  • Page 29: It would be useful if the authors could expand a bit on how strong is the assumption of continuous deformation entering in the ``no-go theorem on the boundary".

  • Page 29: In the no-go theorem in AdS" it should be specified that the deformation islocal". I wonder if the word ``interaction" is meant to exclude the deformation $m^2\to m^2+\delta m^2$. If so, it might be useful to be more explicit. Is it obvious that any local deformation in the bulk leads to a continuous deformation on the boundary?

  • Page 33: it is not clear why the authors comment about the equality between the EOM of $\phi^4$ and sine-Gordon theories at order $\lambda$, given that the $\phi^4$ theory is no longer mentioned after (4.39).

In addition to the typos identified by referee 1, end of paragraph containing eq.(2.17): $\sqrt{\gamma} \xi^\mu n^\mu \Delta T_{\mu\nu}\to \sqrt{\gamma} \xi^\mu n^\nu \Delta T_{\mu\nu}$.

Recommendation

Ask for minor revision

---

## Editorial Decision

resubmitted